# Effective Policy Learning for Multi-Agent Online Coordination Beyond Submodular Objectives

Qixin Zhang[1]*    Yan Sun[2]*    Can Jin[4]    Xikun Zhang[1]
Yao Shu[3]    Puning Zhao[5]    Li Shen[5]    Dacheng Tao[1]†

[1]College of Computing and Data Science, Nanyang Technological University, Singapore
[2]The University of Sydney    [3]Hong Kong University of Science and Technology(Guangzhou)
[4]Rutgers University    [5]Sun Yat-sen University
{qixin.zhang, xikun.zhang, dacheng.tao}@ntu.edu.sg; sun9899@uni.sydney.edu.au
yaoshu@hkust-gz.edu.cn; {zhaopn,shenli6}@mail.sysu.edu.cn

## Abstract

In this paper, we present two effective policy learning algorithms for multi-agent online coordination(MA-OC) problem. The first one, `MA-SPL`, not only can achieve the optimal $(1 - \frac{c}{e})$-approximation guarantee for the MA-OC problem with submodular objectives but also can handle the unexplored $\alpha$-weakly DR-submodular and $(\gamma, \beta)$-weakly submodular scenarios, where $c$ is the curvature of the investigated submodular functions, $\alpha$ denotes the diminishing-return(DR) ratio and the tuple $(\gamma, \beta)$ represents the submodularity ratios. Subsequently, in order to reduce the reliance on the unknown parameters $\alpha, \gamma, \beta$ inherent in the `MA-SPL` algorithm, we further introduce the second online algorithm named `MA-MPL`. This `MA-MPL` algorithm is entirely *parameter-free* and simultaneously can maintain the same approximation ratio as the first `MA-SPL` algorithm. The core of our `MA-SPL` and `MA-MPL` algorithms is a novel continuous-relaxation technique termed as *policy-based continuous extension*. Compared with the well-established *multi-linear extension*, a notable advantage of this new *policy-based continuous extension* is its ability to provide a lossless rounding scheme for any set function, thereby enabling us to tackle the challenging weakly submodular objectives. Finally, extensive simulations are conducted to validate the effectiveness of our proposed algorithms.

## 1 Introduction

Coordinating multiple autonomous agents to cooperatively complete complex tasks in time-varying environments is a significant challenge with extensive applications in machine learning, robotics and control, including target tracking [27, 129, 130, 132], area monitoring [55, 73, 97], multi-path planning [100, 101, 102], mobile sensor placement [65, 67, 95], environmental mapping [6, 75] and task assignment [92, 5]. Critically, with the rapid advancement of self-supervised learning and the development of large Transformers with billions of parameters, large language models (LLMs) are now capable of generating factual and coherent responses to human queries [11, 25, 107, 51]. However, the mixed quality of training data can lead to the generation of undesired responses, presenting a significant challenge. Therefore, how to select a limited number of high-quality input-output examples from multi-source and massive datasets for fine-tuning LLMs remains a critical challenge—a problem that can be formulated as multi-agent collaborative data selection [69, 70, 80, 93]. Moreover, the emergence of foundation models further highlights the necessity of effective

---

*These authors contributed equally to this work.
†Corresponding author: Dacheng Tao.

39th Conference on Neural Information Processing Systems (NeurIPS 2025).

cooperation architectures to enable multi-agent AI systems to mutually learn and adapt in dynamic environments [4, 23, 28, 60]. Motivated by these diverse scenarios, thus this paper delves into the Multi-Agent Online Coordination(MA-OC) problem, establishing a theoretical paradigm for real-world embodied systems, LLMs fine-tuning, and intelligent AI agent collaboration.

Prior to this, numerous studies have demonstrated that the utility functions associated with a wide range of multi-agent coordination scenarios often exhibit a *diminishing-return*(DR) property. Specifically, as the number of agents increases, the marginal gain in benefit will tend to decrease. For instance, in area monitoring using a fleet of unmanned aerial vehicles(UAVs), due to the overlaps of sensing ranges, the increment of the total monitored area from adding an additional UAV typically becomes less and less as the team of UAVs expands. Note that the *diminishing-return* property is also known as *submodularity* in mathematics [7, 42]. Consequently, a vast majority of research regarding multi-agent coordination focus on *submodular* objectives [45, 50, 64, 66, 78, 92, 94, 95, 114, 115, 126, 129, 131].

However, recent works [55, 56, 59, 62, 117] observed that there also exist many multi-agent coordination scenarios inducing utility functions that are *close-to-submodular*, but *not strictly submodular*. A notable example is the employment of a swarm of UAVs to track multiple moving objects. In this scenario, each UAV needs to periodically determine its moving direction and speed. Particularly, under the Kalman filter framework, [55, 56, 62] pointed out that the aforementioned trajectory selection problem of UAVs can be formulated as a multi-agent variant of the general online *weakly* submodular maximization problems. It is worth noting that the currently well-known algorithms for weakly submodular maximization are highly dependent on the discrete local search [20, 43, 53, 76, 106]. So far, however, how to extend this local search into online settings still remains an uncharted territory. Thus, almost all existing studies regarding multi-agent coordination with weakly submodular objectives choose to neglect the changing environment and focus on simple offline scenarios [55, 56, 62, 117]. In view of all this, a natural question arises:

**Q1:** *Is it possible to design an effective online algorithm for MA-OC problem with weakly submodular objectives?*

In addition, we also find that when the objective function is exactly *submodular*, the state-of-the-art algorithms [94, 126] for the MA-OC problem only can guarantee a sub-optimal $(\frac{1-e^{-c}}{c})$-approximation, which mismatches the best possible $(1 - \frac{c}{e})$-approximation established in the works [105, 54] for *single-agent* submodular maximization. Here, $c$ denotes the curvature of the investigated submodular functions. Given this drawback, another question comes to our mind, that is,

**Q2:** *Is it possible to achieve the optimal $(1 - \frac{c}{e})$-approximation ratio for MA-OC problem with submodular objectives?*

In the subsequent sections of this paper, we will provide an affirmative answer to these two questions by presenting an effective online algorithm named `MA-SPL`, which not only can achieve the optimal $(1 - \frac{c}{e})$-approximation for the MA-OC problem with submodular objectives but also can address the previously unexplored weakly submodular scenarios. The core of our proposed `MA-SPL` algorithm is a novel continuous-relaxation framework named *policy-based continuous extension*, which can efficiently transform the discrete set function maximization problem into a solvable continuous optimization task. Furthermore, compared to the well-established *multi-linear extension* [15], a notable advantage of our proposed policy-based continuous extension is that it can provide a lossless rounding scheme for any set objective function. In contrast, all known lossless rounding schemes for the *multi-linear extension* require the set objective function to be *submodular* [15, 18, 19]. Moreover, to eliminate the dependence of both DR ratio and submodularity ratio inherent in our proposed `MA-SPL` algorithm, we further present a *parameter-free* online algorithm termed as `MA-MPL` for the MA-OC problem with general weakly submodular objectives.

In summary, we make the following contributions:

- This paper introduces an innovative continuous-relaxation technique named *policy-based continuous extension* for the general multi-agent coordination problem. Furthermore, we conduct an in-depth exploration of the differentiability, monotonicity and submodularity of our proposed policy-based continuous extension. More importantly, when the investigated set objective function is submodular or weakly submodular, we design three different surrogate functions for our policy-based continuous extension. The stationary points of these three surrogate functions can yield a better approximation guarantee than those of the original policy-based continuous extension itself.

Table 1: Comparison of the different algorithms for $T$-round MA-OC problem. Note that 'Approx.' denotes the obtained approximation ratio, '#Com.' represents the number of communication, '#Queries' denotes the number of queries to the set objective functions, 'D-Regret' denotes the dynamic regret bound, 'Proj-free' indicates whether the method does not require projection, 'Para-free' indicates whether the method does not require prior knowledge of curvature $c$ and parameters $\alpha, \gamma, \beta$, $\mathcal{P}_T$ is the deviation of maximizer sequence, $\tau$ is the spectral gap of the weight matrix, $d(G)$ is the diameter of the graph $G$, $\phi(\gamma, \beta) \triangleq \beta(1-\gamma) + \gamma^2$ and $\kappa \triangleq \sum_{i=1}^n \kappa_i$.

| Method | Utility | Para-free | Proj-free | Graph | Approx. | #Com. | #Queries | D-Regret |
|---|---|---|---|---|---|---|---|---|
| OSG [115, 50] | Submodular | ✔ | ✔ | **Complete** | $\left(\frac{1}{1+c}\right)$ | $\mathcal{O}(T)$ | $\mathcal{O}(\kappa T)$ | $\tilde{\mathcal{O}}(\sqrt{\mathcal{P}_T T})$ |
| MA-OSMA [126] | Submodular | ✗ | ✗ | Connected | $\left(\frac{1-e^{-c}}{c}\right)$ | $\mathcal{O}(T)$ | $\mathcal{O}(\kappa T)$ | $\mathcal{O}\left(\sqrt{\frac{\mathcal{P}_T T}{1-\tau}}\right)$ |
| MA-OSEA [126] | Submodular | ✗ | ✔ | Connected | $\left(\frac{1-e^{-c}}{e}\right)$ | $\mathcal{O}(T)$ | $\mathcal{O}(\kappa T)$ | $\tilde{\mathcal{O}}\left(\sqrt{\frac{\mathcal{P}_T T}{1-\tau}}\right)$ |
| MA-SPL (Algorithm 1) | Submodular | ✔ | ✗ | Connected | $\left(1-\frac{c}{e}\right)$ | $\mathcal{O}(T)$ | $\mathcal{O}(\kappa T)$ | $\mathcal{O}\left(\sqrt{\frac{\mathcal{P}_T T}{1-\tau}}\right)$ |
|  | $\alpha$-weakly DR-Sub | ✗ | ✗ | Connected | $(1-e^{-\alpha})$ | $\mathcal{O}(T)$ | $\mathcal{O}(\kappa T)$ | $\mathcal{O}\left(\sqrt{\frac{\mathcal{P}_T T}{1-\tau}}\right)$ |
|  | $(\gamma,\beta)$-weakly Sub | ✗ | ✗ | Connected | $\left(\frac{\gamma^2(1-e^{-\phi(\gamma,\beta)})}{\phi(\gamma,\beta)}\right)$ | $\mathcal{O}(T)$ | $\mathcal{O}(\kappa T)$ | $\mathcal{O}\left(\sqrt{\frac{\mathcal{P}_T T}{1-\tau}}\right)$ |
| MA-MPL (Algorithm 2) | $\alpha$-weakly DR-Sub | ✔ | ✔ | Connected | $(1-e^{-\alpha})$ | $\mathcal{O}(T^{3/2})$ | $\mathcal{O}(\kappa T^{5/2})$ | $\mathcal{O}\left(d(G)\sqrt{\mathcal{P}_T T}\right)$ |
|  | $(\gamma,\beta)$-weakly Sub | ✔ | ✔ | Connected | $\left(\frac{\gamma^2(1-e^{-\phi(\gamma,\beta)})}{\phi(\gamma,\beta)}\right)$ | $\mathcal{O}(T^{3/2})$ | $\mathcal{O}(\kappa T^{5/2})$ | $\mathcal{O}\left(d(G)\sqrt{\mathcal{P}_T T}\right)$ |

- Building on these surrogate functions, we then propose a novel online algorithm named MA-SPL for the concerned MA-OC problem. Moreover, we also verify that, when the set objective function is monotone submodular with curvature $c$, $\alpha$-weakly DR-submodular or $(\gamma, \beta)$-weakly submodular, our proposed MA-SPL can achieve an approximation ratio of $(1-\frac{c}{e})$, $(1-e^{-\alpha})$ or $\left(\frac{\gamma^2(1-e^{-(\beta(1-\gamma)+\gamma^2)})}{\beta(1-\gamma)+\gamma^2}\right)$ with a dynamic regret bound of $\mathcal{O}\left(\sqrt{\frac{\mathcal{P}_T T}{1-\tau}}\right)$ to the best comparator in hindsight, respectively. Here, $\mathcal{P}_T$ is the deviation of maximizer sequence, $\tau$ is the spectral gap of the network, $\alpha$ denotes the DR ratio, $\beta$ represents the upper submodularity ratio and $\gamma$ is the lower submodularity ratio.

- To eliminate the dependence of both DR ratio and submodularity ratio of our MA-SPL algorithm, we next present a *parameter-free* online algorithm named MA-MPL for the MA-OC problem. The cornerstone of this MA-MPL algorithm is a novel inequality between our proposed policy-based continuous extension and the original set objective function. Moreover, when the objective function is monotone $\alpha$-weakly DR-submodular or $(\gamma, \beta)$-weakly submodular, our proposed MA-MPL algorithm also can enjoy the same approximation ratio with a regret bound of $\mathcal{O}\left(d(G)\sqrt{\mathcal{P}_T T}\right)$ to the best comparator in hindsight, where $d(G)$ is the diameter of the corresponding communication graph $G$.

- We conduct numerical experiments to verify the effectiveness of our proposed algorithms.

**Related Work.** Due to space limitations, the comprehensive literature review is placed in Appendix A. In particular, we present a detailed comparison of our proposed MA-SPL algorithm and MA-MPL algorithm with existing studies on multi-agent online coordination in Table 1.

## 2 Problem Setup

This section will provide a detailed introduction to multi-agent online coordination(MA-OC) problem.

In MA-OC problem, we generally consider a collection of $n$ distinct agents, indexed by the set $\mathcal{N} \triangleq \{1, \ldots, n\}$ and interconnected through an undirected network $G(\mathcal{N}, \mathcal{E})$. Here, $\mathcal{E} \subseteq \mathcal{N} \times \mathcal{N}$ represents the possible communication links among agents. Additionally, each agent $i \in \mathcal{N}$ is endowed with a unique set of actions $\mathcal{V}_i \triangleq \{v_{i,1}, \ldots, v_{i,\kappa_i}\}$, meaning that these action sets are mutually disjoint, i.e., $\mathcal{V}_i \cap \mathcal{V}_j = \emptyset$ for any $i, j \in \mathcal{N}$. In the process of multi-agent online coordination, at every time spot $t \in [T]$, each agent $i \in \mathcal{N}$ will separately select one action $a_i(t)$ from its own action set $\mathcal{V}_i$. After committing to these choices, the environment will reveal a utility set function $f_t$ defined over the aggregated action space $\mathcal{V} \triangleq \cup_{i \in \mathcal{N}} \mathcal{V}_i$. Then, the agents receive the utility $f_t(\cup_{i \in \mathcal{N}} \{a_i(t)\})$. As a result, the goal of agents is to maximize their cumulative reward as much as possible. Specifically, at each time step $t \in [T]$, we need to address the following set function maximization problem in a multi-agent collaborative manner:

$$\max f_t(S), \quad \text{s.t. } S \subseteq \mathcal{V} \text{ and } |S \cap \mathcal{V}_i| \leq 1, \forall i \in \mathcal{N}. \tag{1}$$

Furthermore, in numerous practical applications regarding MA-OC problem, each agent usually has a limited perceptual range, allowing it to sense only the environmental changes in its immediate

surroundings. For example, in target tracking scenarios involving a swarm of unmanned aerial vehicles(UAVs), each UAV only can perceive the targets within its sensing radius, leaving those outside this range undetected. To model this limitation, several studies on the MA-OC problem [126, 115, 114, 95, 94] adopt a local feedback model. More specifically, after $f_t$ is revealed, each agent $i \in \mathcal{N}$ is only permitted to query a local marginal oracle $\mathcal{Q}_t^i : \mathcal{V}_i \times 2^{\mathcal{V}} \to \mathbb{R}_+$ defined as $\mathcal{Q}_t^i(a, S) \triangleq f_t(a|S) \triangleq \big(f_t(S \cup \{a\}) - f_t(S)\big)$ for any $a \in \mathcal{V}_i$ and $S \subseteq \mathcal{V}$. This implies that, at each time $t \in [T]$, agents only can receive the marginal evaluations about the actions within their individual action set, rather than the full information of $f_t$. We also impose this local-feedback constraint in this paper.

Before going into the details, it is crucial to emphasize that, in many real-world scenarios, there exists such an (approximate) local marginal feedback oracle $\mathcal{Q}_t^i$ for each agent $i \in \mathcal{N}$ after $f_t$ is revealed. Typically, in addition to the decision-making process, agents often utilize various off-the-shelf learning algorithms to estimate the marginal contributions of their available actions based on the observed and collected information (See [27]). Moreover, the local information available to one agent is often insufficient for precisely assessing the actions of other agents who are not in close vicinity. Given this fact, confining each agent to the marginal estimations of the actions within its own action set also can further reduce the accumulation of learning errors.

**Dynamic $\rho$-Regret:** Generally speaking, the set function maximization problem (1) is **NP**-hard [83, 35], indicating that no polynomial-time algorithms can solve it optimally. Thus, this paper employs the *dynamic $\rho$-regret* [21, 61, 103, 115, 126, 134] to measure the performance of our proposed algorithms for MA-OC problem. In *dynamic $\rho$-regret*, the algorithm is compared against a sequence of local maximizers with scale parameter $\rho \in [0, 1]$, i.e., $R_\rho^*(T) \triangleq \rho \sum_{t=1}^{T} f_t(\mathcal{A}_t^*) - \sum_{t=1}^{T} f_t\big(\cup_{i \in \mathcal{N}} \{a_i(t)\}\big)$, where $\mathcal{A}_t^*$ is the optimal solution of problem (1) and $a_i(t)$ is the action chosen by agent $i$ at time $t$.

# 3   Preliminaries

In this section, we introduce some basic concepts and the frequently used notations.

**Notations.** For any positive integer $n$, the symbol $[n]$ denotes the set $\{1, \ldots, n\}$. $\mathbf{0}_p$ and $\mathbf{1}_p$ denote the $p$-dimensional vector whose all components are 0 and 1, respectively. Moreover, $\|\cdot\|_1$ and $\|\cdot\|_2$ stand for the $L_1$ norm and $L_2$ norm for vectors, respectively. We also use $\Delta_m$ to represent the standard $m$-dimensional simplex, that is, $\Delta_m \triangleq \{(x_1, \ldots, x_m)| \sum_{i=1}^{m} x_i \leq 1, \ x_i \geq 0, \forall i \in [m]\}$.

**Submodularity and Curvature.** Let $\mathcal{V}$ be a finite ground set and $f : 2^{\mathcal{V}} \to \mathbb{R}_+$ be a set function mapping any subset of $\mathcal{V}$ to a non-negative real number. Then, for any two subsets $S, T \subseteq \mathcal{V}$, we denote by $f(T|S)$ the marginal contribution of adding the elements of $T$ to $S$, i.e., $f(T|S) \triangleq f(T \cup S) - f(S)$. In particular, when $T$ is a singleton set $\{v\}$, we also use $f(v|S)$ to represent $f(\{v\}|S)$. Therefore, we say a set function $f$ is *submodular* if and only if it satisfies the *diminishing-return property* [85, 42, 41], that is, $f(v|S) \geq f(v|T)$ for any $S \subseteq T \subseteq \mathcal{V}$ and $v \in \mathcal{V} \setminus T$. To precisely characterize the diminishing-return property, [26, 36, 105, 109] introduced the concept of *curvature* for submodular functions, which is defined as $c \triangleq 1 - \min_{S \subseteq \mathcal{V}, v \notin S} \frac{f(S \cup \{v\}) - f(S)}{f(\{v\}) - f(\emptyset)}$.

**Monotonicity.** A set function $f : 2^{\mathcal{V}} \to \mathbb{R}_+$ is *monotone* if and only if $f(S) \leq f(T)$ for any $S \subseteq T \subseteq \mathcal{V}$. Moreover, in this paper, we suppose the set function $f$ is *normalized*, that is, $f(\emptyset) = 0$.

**Weak Submodularity.** A set function $f : 2^{\mathcal{V}} \to \mathbb{R}_+$ is said to be $\gamma$-*weakly submodular from below* for some $\gamma \in (0, 1]$ if and only if $\sum_{v \in T \setminus S} f(v|S) \geq \gamma\big(f(T) - f(S)\big)$ for any two subsets $S \subseteq T \subseteq \mathcal{V}$, where $\gamma$ is called as the *lower submodularity ratio* [29, 30, 20]. Similarly, we also can define the *weak submodularity from above*, that is, a set function $f : 2^{\mathcal{V}} \to \mathbb{R}_+$ is $\beta$-*weakly submodular from above* for some $\beta \geq 1$ if and only if $\sum_{v \in T \setminus S} f(v|T - \{v\}) \leq \beta\big(f(T) - f(S)\big), \forall S \subseteq T \subseteq \mathcal{V}$, where $\beta$ is the *upper submodularity ratio*. When a set function $f$ is both $\gamma$-*weakly submodular from below* and $\beta$-*weakly submodular from above*, we say it is $(\gamma, \beta)$-*weakly submodular* [106].

**Weak DR-submodularity.** A set function $f : 2^{\mathcal{V}} \to \mathbb{R}_+$ is $\alpha$-*weakly DR-submodular* for some $\alpha \in (0, 1]$ if and only if $f(v|S) \geq \alpha f(v|T)$ for any two subsets $S \subseteq T \subseteq \mathcal{V}$ and $v \in \mathcal{V} \setminus T$. In particular, $\alpha$ is often called as the diminishing-return(DR) ratio [9, 68, 43, 48, 76]. It is worth noting that, from the previous definition of weak submodularity, we can infer that an $\alpha$-weakly DR-submodular function automatically satisfies the conditions for being $(\alpha, \frac{1}{\alpha})$-weakly submodular.

# 4 Policy-based Continuous-Relaxation Framework

Before presenting our proposed algorithms for the MA-OC problem, we firstly explore the offline set function maximization problem (1). In recent years, compared to discrete optimization, the field of continuous optimization has made significant advancements, yielding a broad spectrum of effective algorithmic frameworks and theoretical tools. Consequently, one promising strategy to addressing the set function maximization problem (1) is to convert it into a solvable continuous optimization problem throughout *continuous-relaxation* techniques.

A well-known continuous-relaxation framework is the *multi-linear extension* [15], which was introduced for maximizing submodular set functions. Regrettably, this relaxation framework cannot be directly applied to the general set function maximization problem (1), as most existing lossless rounding schemes for *multi-linear extension*—such as pipage rounding [1], swap rounding [18], and contention resolution [19]—rely heavily on the *submodular* assumption. Note that lossless rounding schemes refer to methods that convert the obtained continuous solution into a feasible discrete solution without any loss in terms of the objective function value. To date, how to losslessly round the *multi-linear extension* of non-submodular set functions, e.g. $(\gamma, \beta)$-weakly submodular and $\alpha$-weakly DR-submodular functions, still remains an open question [106]. To overcome this hurdle, we will introduce an innovative continuous-relaxation technique in the subsequent part of this section.

## 4.1 Policy-based Continuous Extension

From the previous description about the MA-OC problem provided in Section 2, we can view the set function maximization problem (1) as a variant of multi-agent cooperative game [2, 98, 112, 120]. Inspired by this viewpoint, we naturally consider whether each agent $i \in \mathcal{N}$ can learn a policy $\pi_i \triangleq (\pi_{i,1}, \ldots, \pi_{i,\kappa_i})$ over its individual action space $\mathcal{V}_i \triangleq \{v_{i,1}, \ldots, v_{i,\kappa_i}\}$ and then utilizes this policy $\pi_i$ to make decision, where each $\pi_{i,m}$ represents the probability of agent $i$ taking the action $v_{i,m}, \forall m \in [\kappa_i]$. Based on this idea, if letting $a_i \in \mathcal{V}_i \cup \{\emptyset\}$ denote the random action chosen by each policy $\pi_i, \forall i \in \mathcal{N}$, then we can obtain the following policy-based continuous extension, namely,

**Definition 1.** *If $\pi_i \triangleq (\pi_{i,1}, \ldots, \pi_{i,\kappa_i}) \in \Delta_{\kappa_i}$ for any $i \in \mathcal{N}$, then the policy-based continuous extension $F_t : \prod_{i=1}^{n} \Delta_{\kappa_i} \to \mathbb{R}_+$ for the set function maximization problem (1) can be defined as:*

$$F_t(\pi_1, \ldots, \pi_n) \triangleq \sum_{a_i \in \mathcal{V}_i \cup \{\emptyset\}, \forall i \in \mathcal{N}} \left( f_t\left( \cup_{i=1}^{n} \{a_i\} \right) \prod_{i=1}^{n} p(a_i | \pi_i) \right), \tag{2}$$

*where $p(\cdot | \pi_i)$ is a probability distribution over the set $\mathcal{V}_i \cup \{\emptyset\}$, that is, $p(v_{i,m} | \pi_i) = \pi_{i,m}, \forall i \in [n], \forall m \in [\kappa_i]$ and $p(\emptyset | \pi_i) = 1 - \sum_{m=1}^{K_i} \pi_{i,m}, \forall i \in \mathcal{N}$.*

**Remark 1.** *It is noteworthy that, in Eq.(2), with the probability $1 - \sum_{m=1}^{\kappa_i} \pi_{i,m}$, the policy $\pi_i$ will not pick any action from $\mathcal{V}_i$, which means there is a possibility that no action will be chosen, i.e., $\emptyset$.*

**Remark 2.** *The definition in Eq.(2) highlights a notable advantage of our proposed policy-based continuous extension: it does not assign probabilities to any subset that violates the constraint of problem (1). As a result, for any set function $f_t$ and any $(\pi_1, \ldots, \pi_n) \in \prod_{i=1}^{n} \Delta_{\kappa_i}$, throughout the Definition 1, we can easily generate a subset, i.e., $\cup_{i=1}^{n}\{a_i\}$, that adheres to the constraints of problem (1) while ensuring $\mathbb{E}\left( f_t\left( \cup_{i=1}^{n}\{a_i\} \right) \right) = F_t(\pi_1, \ldots, \pi_n)$. In contrast, all known lossless rounding schemes of the multi-linear extension require the function $f_t$ to be submodular [15, 18, 19].*

**Remark 3.** *Notably, we observe that the works [96, 128] have introduced two relaxation techniques for lattice submodular and $k$-submodular functions, both of which are analogous to our policy-based continuous extension $F_t$. However, it is crucial to emphasize that there exist notable differences between our work and [96, 128]: a) The lattice formulation typically requires an 'order' relationship among different actions of the same agent. However, in our multi-agent coordination problem, we do not impose any specific order on the decisions. Consequently, the results and algorithms from [96, 128] are not directly applicable to our scenario. b) In addition to submodularity, our paper also considers weak submodularity and allows for varying sizes of action sets among agents.*

With the policy-based continuous extension $F_t$ defined in Eq.(2), the set function maximization problem (1) can be naturally relaxed into a continuous maximization task, i.e.,

$$\max F_t(\pi_1, \ldots, \pi_n), \quad \text{s.t. } \|\pi_i\|_1 \le 1, \pi_i \in [0,1]^{\kappa_i}, \forall i \in \mathcal{N}. \tag{3}$$

In order to effectively tackle the policy optimization problem (3), we next investigate the properties of our proposed policy-based continuous extension $F_t$.

## 4.2 Properties of Policy-based Continuous Extension

This subsection will focus on characterizing the differentiability, monotonicity and submodularity of our proposed policy-based continuous extension. Specifically, we have the following theorem:

**Theorem 1** (Proof in Appendix D). *The policy-based continuous extension $F_t : \prod_{i=1}^n \Delta_{\kappa_i} \to \mathbb{R}_+$ defined in Eq.(2) satisfies the following properties:*

*1): For any point $(\boldsymbol{\pi}_1, \ldots, \boldsymbol{\pi}_n) \in \prod_{i=1}^n \Delta_{\kappa_i}$, the first-order derivative of $F_t$ at variable $\pi_{i,m}, \forall i \in \mathcal{N}, \forall m \in [\kappa_i]$, can be expressed as follows:*

$$\frac{\partial F_t}{\partial \pi_{i,m}}(\boldsymbol{\pi}_1, \ldots, \boldsymbol{\pi}_n) \triangleq \mathbb{E}_{a_j \sim \boldsymbol{\pi}_j, \forall j \in \mathcal{N}} \Big( f_t\big(v_{i,m} \big| \cup_{j \neq i, j \in \mathcal{N}} \{a_j\}\big)\Big),$$

*where $a_j \sim \boldsymbol{\pi}_j$ indicates that action $a_j$ is randomly selected from $\mathcal{V}_j \cup \{\emptyset\}$ based on the policy $\boldsymbol{\pi}_j$;*

*2): If the set function $f_t$ is monotone, then $\frac{\partial F_t}{\partial \pi_{i,m}}(\boldsymbol{\pi}_1, \ldots, \boldsymbol{\pi}_n) \geq 0$ for any point $(\boldsymbol{\pi}_1, \ldots, \boldsymbol{\pi}_n) \in \prod_{i=1}^n \Delta_{\kappa_i}, i \in \mathcal{N}$ and $m \in [\kappa_i]$, which means the monotonicity of $f_t$ can be inherited by $F_t$;*

*3): If $f_t$ is $\alpha$-weakly DR-submodular, then $F_t$ is $\alpha$-weakly continuous DR-submodular [58, 88] over $\prod_{i=1}^n \Delta_{\kappa_i}$, that is, for any two point $(\boldsymbol{\pi}_1^a, \ldots, \boldsymbol{\pi}_n^a) \in \prod_{i=1}^n \Delta_{\kappa_i}$ and $(\boldsymbol{\pi}_1^b, \ldots, \boldsymbol{\pi}_n^b) \in \prod_{i=1}^n \Delta_{\kappa_i}$, if $\boldsymbol{\pi}_i^a \leq \boldsymbol{\pi}_i^b \; \forall i \in \mathcal{N}$, we have that $\nabla F_t(\boldsymbol{\pi}_1^a, \ldots, \boldsymbol{\pi}_n^a) \geq \alpha \nabla F_t(\boldsymbol{\pi}_1^b, \ldots, \boldsymbol{\pi}_n^b)$;*

*4): For any subset $S$ within the constraint of problem (1) and any point $(\boldsymbol{\pi}_1, \ldots, \boldsymbol{\pi}_n) \in \prod_{i=1}^n \Delta_{\kappa_i}$, when $f_t$ is monotone $\alpha$-weakly DR-submodular, the following inequality holds:*

$$\alpha\Big(f_t(S) - F_t(\boldsymbol{\pi}_1, \ldots, \boldsymbol{\pi}_n)\Big) \leq \sum_{(i,m):v_{i,m} \in S} \frac{\partial F_t}{\partial \pi_{i,m}}(\boldsymbol{\pi}_1, \ldots, \boldsymbol{\pi}_n), \tag{4}$$

*where $\{(i,m) : v_{i,m} \in S\}$ denotes the set of all indices $(i,m)$ such that $v_{i,m} \in S$. Similarly, when $f_t$ is monotone $(\gamma, \beta)$-weakly submodular, we can show that*

$$\Big(\gamma^2 f_t(S) - (\beta(1-\gamma) + \gamma^2)F_t(\boldsymbol{\pi}_1, \ldots, \boldsymbol{\pi}_n)\Big) \leq \sum_{(i,m):v_{i,m} \in S} \frac{\partial F_t}{\partial \pi_{i,m}}(\boldsymbol{\pi}_1, \ldots, \boldsymbol{\pi}_n). \tag{5}$$

**Remark 4.** *Part 3) indicates that when $f_t$ is submodular, namely $\alpha = 1$, our policy-based continuous extension $F_t$ belongs to the well-studied continuous DR-submodular functions. However, it is important to highlight that these existing results for continuous DR-submodular maximization [57, 58, 89, 90, 111, 122] cannot be directly extended to our policy-based continuous extension $F_t$. This is because all of them heavily rely on the inequality $\langle \mathbf{y} - \mathbf{x}, \nabla G(\mathbf{x}) \rangle \geq G(\mathbf{y} \vee \mathbf{x}) - G(\mathbf{x})$ where $G$ is a monotone continuous DR-submodular function and $\vee$ denotes coordinate-wise maximum operation. Note that the previous inequality requires that the domain of $G$ must be closed under the maximum operation $\vee$. However, the domain $\prod_{i=1}^n \Delta_{\kappa_i}$ of our proposed $F_t$ does not meet this requirement.*

## 5 Multi-Agent Policy Learning

This section explores how to utilize the policy-based continuous extension introduced in Section 4 to address our concerned MA-OC problem. Broadly speaking, a notable advantage of continuous-relaxation techniques is that they enable the use of gradient-based methods, such as gradient ascent [8, 72, 86] and Frank-Wolfe method [10, 71], to tackle discrete optimization problems. Moreover, as is well established in the literature [3, 31, 44, 71], under mild conditions, a wide range of gradient-based algorithms can converge to the stationary points of their target objectives. Motivated by these findings, we next investigate the stationary points of our proposed policy-based continuous extension $F_t$.

### 5.1 Stationary Points and Surrogate Functions

At first, we recall the definition of stationary points for maximization problem, that is,

**Definition 2.** *Given a differentiable function $G : \mathcal{K} \to \mathbb{R}$ and a domain $\mathcal{C} \subseteq \mathcal{K}$, a point $\mathbf{x} \in \mathcal{C}$ is called as a stationary point for the function $G$ over the domain $\mathcal{C}$ if and only if $\max_{\mathbf{y} \in \mathcal{C}} \langle \mathbf{y} - \mathbf{x}, \nabla G(\mathbf{x}) \rangle \leq 0$.*

Next, we examine the performance of the stationary points of our proposed policy-based continuous extension $F_t$ relative to the maximum value of problem (1). Specifically, we have that

**Theorem 2** (Proof in Appendix E). *Given a set function $f_t : 2^{\mathcal{V}} \to \mathbb{R}_+$, if $(\pi_1^s, \dots, \pi_n^s)$ is a stationary point of its policy-based continuous extension $F_t$ over the domain $\prod_{i=1}^n \Delta_{\kappa_i}$ and $S^*$ denotes the optimal solution of the corresponding maximization problem* (1), *then the following inequalities hold:*

*1): When $f_t$ is monotone submodular with curvature $c \in [0, 1]$, $F_t(\pi_1^s, \dots, \pi_n^s) \geq \left(\frac{1}{1+c}\right) f_t(S^*)$;*

*2): When $f_t$ is monotone $\alpha$-weakly DR-submodular, $F_t(\pi_1^s, \dots, \pi_n^s) \geq \left(\frac{\alpha^2}{1+\alpha^2}\right) f_t(S^*)$;*

*3): When $f_t$ is monotone $(\gamma, \beta)$-weakly submodular, $F_t(\pi_1^s, \dots, \pi_n^s) \geq \left(\frac{\gamma^2}{\beta+\beta(1-\gamma)+\gamma^2}\right) f_t(S^*)$.*

**Remark 5.** *It is worth noting that, in a certain sense, the approximation guarantees established in Theorem 2 is **tight**. In Appendix E.3, we will present a simple instance of a submodular function $f_t$, i.e., $c = \alpha = \gamma = \beta = 1$, whose policy-based continuous extension $F_t$ can attain a $\frac{1}{2}$-approximation guarantee at a stationary point that is also a local maximum.*

Theorem 2 indicates that when the original set function $f_t$ is monotone submodular with curvature $c$, $\alpha$-weakly DR-submodular or $(\gamma, \beta)$-weakly submodular, the direct application of gradient-based methods on our proposed policy-based continuous extension $F_t$ can ensure an approximation ratio of $(\frac{1}{1+c})$, $(\frac{\alpha^2}{1+\alpha^2})$ or $(\frac{\gamma^2}{\beta+\beta(1-\gamma)+\gamma^2})$ to the problem (1), respectively. However, as shown in [105, 36], the optimal approximation ratio for maximizing a monotone submodular function with curvature $c$ is $(1 - \frac{c}{e})$, which significantly exceeds the previous $(\frac{1}{1+c})$-approximation provided by stationary points of $F_t$. Similarly, [53] pointed out that the optimal approximation for $\alpha$-weakly DR-submodular maximization is $(1 - e^{-\alpha})$, which is also greater than the $(\frac{\alpha^2}{1+\alpha^2})$-approximation established in Theorem 2. Motivated by these findings, we naturally wonder *whether it is possible to enhance the approximation guarantees of the stationary points of our proposed policy-based continuous extension.*

Previously, the *multi-linear extension* of submodular functions also encountered a similar issue, namely, the stationary points of the *multi-linear extension* only can guarantee a *sub-optimal* $\frac{1}{2}$-approximation [38, 58]. To overcome this drawback, prior studies [39, 40, 122, 126] constructed a novel surrogate function for the *multi-linear extension* and proved that the stationary points of this surrogate function can achieve the optimal $(1 - \frac{1}{e})$-approximation. Inspired by this idea, we also hope to develop a surrogate function for our proposed policy-based continuous extension $F_t$ such that it can improve the approximation ratios of the stationary points of $F_t$. Specifically, we have that

**Theorem 3** (Proof in Appendix F). *Similar to [122, 125, 126], for any given policy-based continuous extension $F_t$ introduced in Definition 1, we consider a surrogate function $F_t^s : \prod_{i=1}^n \Delta_{K_i} \to \mathbb{R}_+$ whose gradient at each point $\mathbf{x} \in \prod_{i=1}^n \Delta_{\kappa_i}$ is a weighted average of the gradient $\nabla F_t(z * \mathbf{x})$, i.e., $\nabla F_t^s(\mathbf{x}) = \int_0^1 w(z) \nabla F_t(z * \boldsymbol{x}) \mathrm{d}z$ where $w(z)$ is a positive weight function over $[0, 1]$. After elaborately designing the weight function $w(z)$, we can show that:*

*1): When $f_t$ is $\alpha$-weakly DR-submodular and $w(z) = e^{\alpha(z-1)}$, for any stationary point $(\pi_1^s, \dots, \pi_n^s)$ of the surrogate objective $F_t^s$ over $\prod_{i=1}^n \Delta_{\kappa_i}$, then we have $F_t(\pi_1^s, \dots, \pi_n^s) \geq \left(1 - e^{-\alpha}\right) f_t(S^*)$;*

*2): When $f_t$ is $(\gamma, \beta)$-weakly submodular and $w(z) = e^{\phi(\gamma, \beta)(z-1)}$ where $\phi(\gamma, \beta) = \beta(1 - \gamma) + \gamma^2$, for any stationary point $(\pi_1^s, \dots, \pi_n^s)$ of the surrogate objective $F_t^s$ over the domain $\prod_{i=1}^n \Delta_{\kappa_i}$, then we can show that $F_t(\pi_1^s, \dots, \pi_n^s) \geq \left(\frac{\gamma^2(1-e^{-\phi(\gamma,\beta)})}{\phi(\gamma,\beta)}\right) f_t(S^*)$;*

*3): When $f_t$ is submodular with curvature $c$ and $w(z) = e^{z-1}$, for any stationary point $(\pi_1^s, \dots, \pi_n^s)$ of the objective $(F_t^s + \frac{G_t}{e})$ over $\prod_{i=1}^n \Delta_{\kappa_i}$, then we have $F_t(\pi_1^s, \dots, \pi_n^s) \geq \left(1 - \frac{c}{e}\right) f_t(S^*)$, where $S^*$ is the optimal subset of problem (1) and $G_t(\pi_1, \dots, \pi_n) \triangleq \sum_{i=1}^n \sum_{m=1}^{\kappa_i} \left(f_t(v_{i,m}|\mathcal{V}-\{v_{i,m}\})\right)\pi_{i,m}$.*

**Remark 6.** *Note that part 3) of Theorem 3 considers the objective $(F_t^s + \frac{G_t}{e})$ instead of the surrogate $F_t^s$ alone. Here, $G_t$ is a special linear linear function related to the minimum margin gains of the corresponding submodular objective $f_t$, that is, $f_t(a|\mathcal{V}-\{a\}), \forall a \in \mathcal{V}$. Furthermore, it is important to emphasize that the proof of Theorem 3 is not a parallel copy of the articles [122, 126] regarding the multi-linear extension. This is because the works [122, 126] also utilized the same inequality in Remark 4, which requires our domain to be closed under the operation $\vee$. However, the domain of our $F_t$ does not satisfy this condition. **As a result, new techniques are required to verify Theorem 3.***

---

**Algorithm 1:** Multi-Agent Surrogate Policy Learning(`MA-SPL`)

---

**Input:** Time horizon $T$, weight function $w(z)$, action set $\mathcal{V}_i \triangleq \{v_{i,1}, \ldots, v_{i,\kappa_i}\}$, weight matrix
$\quad$ $\mathbf{W} \triangleq [w_{ij}]_{n \times n}$, parameters $(\alpha, \gamma, \beta)$ and step size $\eta_t, \forall t \in [T]$

`// Policy Initialization (Lines 1-2)`

1   Initialize a policy vector $(\boldsymbol{\pi}_{i,1}(1), \ldots, \boldsymbol{\pi}_{i,n}(1))$ for each agent $i \in \mathcal{N}$;

2   Set the policy $\boldsymbol{\pi}_{i,i}(1) \triangleq \frac{1}{\kappa_i}\mathbf{1}_{\kappa_i}$ and $\boldsymbol{\pi}_{i,j}(1) \triangleq \mathbf{0}_{\kappa_j}$ when $i \neq j$ for any $i \in \mathcal{N}$;

3   **for** $t = 1, \ldots, T$ **do**

4      **for** *each agent $i \in \mathcal{N}$* **do**

         `// Actions Sampling and Information Exchange (Lines 5-8)`

5          Compute the normalized policy $\mathbf{p}_i(t) \triangleq \frac{\boldsymbol{\pi}_{i,i}(t)}{\|\boldsymbol{\pi}_{i,i}(t)\|_1}$;

6          Utilize the normalized policy $\mathbf{p}_i(t)$ to sample an action $a_i(t)$ from $\mathcal{V}_i$;

7          Agent $i$ executes the sampled action $a_i(t)$;

8          Exchange policy vector $(\boldsymbol{\pi}_{i,1}(t), \ldots, \boldsymbol{\pi}_{i,n}(t))$ with the neighboring agent $j \in \mathcal{N}_i$;

         `// Surrogate Gradient Estimation (Lines 9-16)`

9          Generate a random number $z_i(t)$ from r.v. $\mathcal{Z}$ where $\Pr(\mathcal{Z} \leq z) \triangleq \frac{\int_0^z w(a)\mathrm{d}a}{\int_0^1 w(a)\mathrm{d}a}, \forall z \in [0, 1]$;

10         **for** $j = 1, \ldots, n$ **do**

11             $\lfloor$ Utilize the weighted policy $z_i(t) * \boldsymbol{\pi}_{i,j}$ to sample an action $\widetilde{a}_j(t) \in \mathcal{V}_i \cup \{\emptyset\}$;

12         Compute the random subset of actions $S_i(t) := \cup_{j \neq i, j \in \mathcal{N}}\{\widetilde{a}_j(t)\}$;

13         Estimate the derivatives of the surrogate function based on Theorem 1 and 3, i.e.,

$$\widehat{\frac{\partial F_t^s}{\partial \pi_{i,m}}}(\boldsymbol{\pi}_{i,1}(t), \ldots, \boldsymbol{\pi}_{i,n}(t)) \triangleq d_{i,m}(t) \triangleq \left(\int_{z=0}^1 w(z)\mathrm{d}z\right)f_t(v_{i,m}|S_i(t)), \forall m \in [\kappa_i];$$

14         **if** *$f_t$ is submodular* **then**

15             $\lfloor$ Update $d_{i,m}(t) = \left(d_{i,m}(t) + e^{-1}f_t(v_{i,m}|\mathcal{V} - \{v_{i,m}\})\right), \forall m \in [\kappa_i]$;

16         Aggregate the surrogate gradient estimations $(d_{i,1}(t), \ldots, d_{i,\kappa_i}(t))$ as vector $\mathbf{d}_i(t)$;

         `// Policy Update (Lines 17-19)`

17         Update $\boldsymbol{\pi}_{i,j}(t+1) \triangleq \sum_{k \in \mathcal{N}_i \cup \{i\}} w_{ik}\boldsymbol{\pi}_{k,j}(t)$ for any $j \neq i$ and $j \in \mathcal{N}$;

18         Compute $\mathbf{y}_{i,i}(t+1) \triangleq \sum_{j \in \mathcal{N}_i \cup \{i\}} w_{ij}\boldsymbol{\pi}_{j,i}(t) + \eta_t\mathbf{d}_i(t)$;

19         Update $\boldsymbol{\pi}_{i,i}(t+1) \triangleq \arg\min_{\mathbf{b} \in \Delta_{\kappa_i}} \|\mathbf{b} - \mathbf{y}_{i,i}(t+1)\|_2$;

---

## 5.2   Multi-Agent Policy Learning via Surrogate Functions

Theorem 3 suggests that, when the original set function $f_t$ is monotone submodular with curvature $c$, $\alpha$-weakly DR-submodular or $(\gamma, \beta)$-weakly submodular, the direct application of gradient-based methods targeting stationary points to the surrogate function $F_t^s$ or its variant $(F_t^s + \frac{G_t}{e})$ can achieve a tight approximation ratio of $(1 - \frac{c}{e})$, $(1 - e^{-\alpha})$ or $\left(\frac{\gamma^2(1 - e^{-(\beta(1-\gamma)+\gamma^2)})}{\beta(1-\gamma)+\gamma^2}\right)$ to the set function maximization problem (1), respectively. Furthermore, recent study [126] developed an effective online algorithm for the multi-agent *submodular* coordination problem based on the well-studied consensus technique [84, 99, 118, 119] and the surrogate functions of *multi-linear extension* [122, 126]. Thus, in order to address the unexplored weakly submodular scenarios and simultaneously achieve the optimal $(1 - \frac{c}{e})$-approximation for submodular settings, we naturally consider replacing the surrogate functions of *multi-linear extension* in the algorithms of [126] with those of our proposed policy-based continuous extension $F_t$. Motivated by this idea, we then present a general online algorithm named `MA-SPL` for MA-OC problem, with details presented in Algorithm 1.

In Algorithm 1, at every time step $t \in [T]$, each agent $i \in \mathcal{N}$ will maintain a local policy vector $(\boldsymbol{\pi}_{i,1}(t), \ldots, \boldsymbol{\pi}_{i,n}(t))$. Here, $\boldsymbol{\pi}_{i,i}(t)$ represents the policy being executed by agent $i$, while $\boldsymbol{\pi}_{i,j}(t), j \neq i$ reflects agent $i$'s current estimate of the policy $\boldsymbol{\pi}_{j,j}(t)$ being taken by other agent $j$. After that, each agent $i$ selects an action $a_i(t)$ from $\mathcal{V}_i$ based on the normalized policy $\mathbf{p}_i(t) \triangleq \frac{\boldsymbol{\pi}_{i,i}(t)}{\|\boldsymbol{\pi}_{i,i}(t)\|_1}$ and shares its local policy vector with the neighboring agent $j \in \mathcal{N}_i$, where $\mathcal{N}_i$ denotes the neighbors

of agent $i$. Then, according to the results of Theorem 1 and Theorem 3, each agent $i$ estimates the first-order partial derivatives of our proposed surrogate function $F_t^s$ at every coordinate $\pi_{i,m}$. Specifically, agent $i$ initially samples a random number $z_i(t)$ from the random variable $\mathcal{Z}$ with distribution $\Pr(\mathcal{Z} \leq z) \triangleq \frac{\int_0^z w(a)\mathrm{d}a}{\int_0^1 w(a)\mathrm{d}a}, \forall z \in [0,1]$ and then approximates each $\frac{\partial F_t^s}{\partial \pi_{i,m}}(\pi_{i,1}(t), \ldots, \pi_{i,n}(t))$ by $d_{i,m}(t) \triangleq (\int_{z=0}^1 w(z)\mathrm{d}z)f_t(v_{i,m}|S_i(t))$ where $S_i(t)$ is a random set generated from the weighted policy vector $z_i(t) * (\pi_{i,1}(t), \ldots, \pi_{i,n}(t))$. Furthermore, when the set objective function $f_t$ is *submodular*, Algorithm 1 will further adjust each surrogate gradient estimation $d_{i,m}(t)$ with the minimum marginal contribution $f_t(v_{i,m}|\mathcal{V}-\{v_{i,m}\}), \forall m \in [\kappa_i]$ (See Lines 14-15). Finally, each agent $i$ updates its policy $\pi_{i,i}(t)$ throughout a projected ascent along the direction $\mathbf{d}_i(t) \triangleq (d_{i,1}(t), \ldots, d_{i,\kappa_i}(t))$.

In sharp contrast with the previous MA-OSMA and MA-OSEA algorithms in [126], the key innovation of Algorithm 1 lies in the use of our proposed policy-based continuous extension and its surrogate functions to update the policy vector in Lines 9-19, rather than relying on the well-studied *multi-linear extension* [15]. Moreover, our proposed policy-based continuous extension naturally aligns with the actions sampling process in Lines 5-7, ensuring that, for any set function $f_t$, the function value at the executed actions $\cup_{i\in\mathcal{N}}\{a_i(t)\}$ is at least as large as the expected function value $F_t(\pi_{1,1}, \ldots, \pi_{n,n})$, namely, $\mathbb{E}(f_t(\cup_{i\in\mathcal{N}}\{a_i(t)\})) \geq F_t(\pi_{1,1}, \ldots, \pi_{n,n})$. This is a significant advantage over the *multi-linear extension*, as the latter generally requires the *submodular* assumption to guarantee the losslessness of the action sampling of Lines 5-7 (See Lemma 13 in [126]).

Next, we provide the theoretical analysis for the proposed Algorithm 1. Before that, we introduce some standard assumptions about the communication graph $G(\mathcal{N}, \mathcal{E})$ and the weight matrix $\mathbf{W}$, i.e.,

**Assumption 1.** *The graph $G(\mathcal{N}, \mathcal{E})$ is connected. Furthermore, the weight matrix $\mathbf{W} \triangleq [w_{ij}]_{n \times n} \in \mathbb{R}_+^{n \times n}$ is symmetric and doubly stochastic, namely, $\mathbf{W}^T = \mathbf{W}$ and $\mathbf{W}\mathbf{1}_n = \mathbf{1}_n$. That is to say, $\tau < 1$ where $\tau = \max(|\lambda_2(\mathbf{W})|, |\lambda_n(\mathbf{W})|)$ is the second largest magnitude of the eigenvalues of $\mathbf{W}$ and $\lambda_i(\mathbf{W})$ denotes the $i$-th largest eigenvalue of matrix $\mathbf{W}$.*

With this Assumption 1, we then can get the following convergence results for our Algorithm 1, i.e.,

**Theorem 4** (Proof in Appendix G). *Under Assumption 1, when each set objective function $f_t$ is monotone submodular with curvature $c$, $\alpha$-weakly DR-submodular or $(\gamma, \beta)$-weakly submodular, if we set the weight function $w(z)$ according to Theorem 3 and choose $\eta_t = \mathcal{O}\big(\sqrt{\frac{(1-\tau)P_T}{T}}\big)$ where $P_T \triangleq \sum_{t=2}^T |\mathcal{A}_t^* \triangle \mathcal{A}_{t-1}^*|$ is the deviation of maximizer sequence and the symbol $\triangle$ denotes the symmetric difference, namely, $S \triangle T = (S \setminus T) \cup (T \setminus S)$, then our proposed Algorithm 1 can achieve a dynamic $\rho$-regret bound of $\mathcal{O}\left(\sqrt{\frac{P_T T}{1-\tau}}\right)$, that is, $\mathbb{E}\left(R_\rho^*(T)\right) \leq \mathcal{O}\left(\sqrt{\frac{P_T T}{1-\tau}}\right)$, where $\rho = (1 - \frac{c}{e})$, $\rho = (1 - e^{-\alpha})$ or $\rho = \big(\frac{\gamma^2(1-e^{-(\beta(1-\gamma)+\gamma^2)})}{\beta(1-\gamma)+\gamma^2}\big)$, respectively.*

**Remark 7.** *To the best of our knowledge, this is the first result that achieves the tight $(1 - \frac{c}{e})$-approximation for the MA-OC problem with submodular objectives and simultaneously can tackle the previously unexplored $(\gamma, \beta)$-weakly submodular and $\alpha$-weakly DR-submodular scenarios.*

**Remark 8.** *Note that when considering the weakly submodular objectives, Line 9 of Algorithm 1 requires prior knowledge of the ratios $\alpha, \gamma, \beta$ to set the weight function $w(z)$. However, in general, accurately computing these parameters will incur exponential computations. To overcome this drawback, we further present a **parameter-free** online algorithm named MA-MPL in Appendix B.*

## 6 Numerical Experiments

In this section, we validate the effectiveness of our proposed MA-SPL and MA-MPL algorithms via two different multi-target tracking scenarios. Especially in Figure 1(a),1(b),1(c), we consider a *submodular* facility-location objective function [115, 126] with different proportions of 'Random', 'Adversarial' and 'Polyline' targets. According to the results in Figure 1(a)-1(c), we can find that the average utility of our MA-SPL can significantly exceed the state-of-the-art MA-OSMA and MA-OSEA algorithms in [126], which is consistent with our Theorem 4. Note that the suffixes in Figure 1(a)-1(d) represent two different choices for communication graphs, where 'c' stands for a complete graph and 'r' denotes an Erdos-Renyi random graph with average degree 4. In contrast, Figure 1(d)-1(f) adopt a bayesian A-optimal criterion for the target tracking task, which will lead to a $\alpha$-weakly

DR-submodular utility function [53, 55, 106]. Given the unknown DR ratio $\alpha$, in Figure 1(e)-1(f), we perform a brute-force 0.1-network search to find the optimal parameter settings for MA-SPL algorithm. Subsequently, we report the best $\alpha = 0.1$ case and the worse $\alpha = 1$ scenario in Figure 1(d). Similarly, from Figure 1(d), we also find that our proposed MA-MPL and MA-SPL can substantially outperform the 'RANDOM' baseline, which is in accord with our theoretical findings. Particularly in 'RANDOM' baseline, we let each agent $i$ randomly execute an action from its own action set $\mathcal{V}_i$. Due to space limitations, more discussions about experiment setups and results are presented in Appendix C.

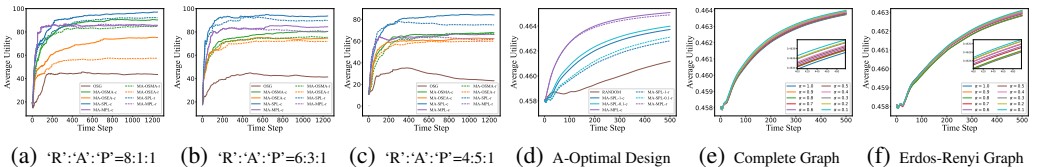

(a) 'R':'A':'P'=8:1:1   (b) 'R':'A':'P'=6:3:1   (c) 'R':'A':'P'=4:5:1   (d) A-Optimal Design   (e) Complete Graph   (f) Erdos-Renyi Graph

Figure 1: A comparison of the running average utility across two distinct target tracking simulations.

# 7    Conclusion

In this paper, we primarily introduce an effective online policy learning algorithm named MA-SPL for the concerned MA-OC problem. Compared with the state-of-art MA-OSMA and MA-OSEA algorithms [126] , our proposed MA-SPL not only can guarantee a tight $(1 - \frac{c}{e})$-approximation for MA-OC problem with submodular objectives but also can address the unexplored $(\gamma, \beta)$-weakly submodular and $\alpha$-weakly DR-submodular scenarios. Subsequently, to eliminate the dependence on the unknown DR ratio and submodularity ratio in our MA-SPL algorithm, we further present a *parameter-free* online algorithm named MA-MPL for the MA-OC problem. The key cornerstone of our MA-SPL and MA-MPL algorithms is a novel continuous relaxation termed as *policy-based continuous extension*. In sharp contrast with the well-studied *multi-linear extension* [15], a notable advantage of this new *policy-based continuous extension* is its ability to provide a lossless rounding scheme for any set function, thereby enabling us to tackle the challenging weakly submodular objective functions.

# Acknowledgment

This project is supported by the National Research Foundation, Singapore, under its NRF Professorship Award No. NRF-P2024-001.

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

**The checklist answers are an integral part of your paper submission.** They are visible to the reviewers, area chairs, senior area chairs, and ethics reviewers. You will be asked to also include it (after eventual revisions) with the final version of your paper, and its final version will be published with the paper.

The reviewers of your paper will be asked to use the checklist as one of the factors in their evaluation. While "[Yes] " is generally preferable to "[No] ", it is perfectly acceptable to answer "[No] " provided a proper justification is given (e.g., "error bars are not reported because it would be too computationally expensive" or "we were unable to find the license for the dataset we used"). In general, answering "[No] " or "[NA] " is not grounds for rejection. While the questions are phrased in a binary way, we acknowledge that the true answer is often more nuanced, so please just use your best judgment and write a justification to elaborate. All supporting evidence can appear either in the main paper or the supplemental material, provided in appendix. If you answer [Yes] to a question, in the justification please point to the section(s) where related material for the question can be found.

IMPORTANT, please:

- **Delete this instruction block, but keep the section heading "NeurIPS Paper Checklist"**,
- **Keep the checklist subsection headings, questions/answers and guidelines below.**
- **Do not modify the questions and only use the provided macros for your answers**.


# Appendices for "Effective Multi-Agent Online Coordination Beyond Submodular Objectives"

## Table of Contents

# A  Literature Review

In this section, we aim to provide a comprehensive review of the related literature.

## A.1  Submodular Maximization

A set function $f : 2^{\mathcal{V}} \to \mathbb{R}_+$ is called submodular if and only if it satisfies the diminishing-return property, namely, for any two subsets $A \subseteq B \subseteq \mathcal{V}$ and $v \in \mathcal{V} \setminus B$, $f(A \cup \{v\}) - f(A) \geq f(B \cup \{v\}) - f(B)$. It is widely recognized that the maximization of submodular functions is **NP**-hard, implying that no polynomial-time algorithms can solve it optimally. To overcome this challenge, [85] introduced a greedy algorithm for solving the monotone submodular maximization problem under a cardinality constraint and demonstrated that this greedy algorithm can achieve an approximation ratio of $(1 - e^{-1})$. Subsequently, [35] showed that this $(1 - e^{-1})$-approximation guarantee is tight for monotone submodular maximization under reasonable complexity-theoretic assumptions. After that, [41] extended the greedy algorithm to the general matroid constraint. However, [41] also pointed out that under such a matroid constraint, the approximation ratio achievable by the greedy algorithm diminishes from $(1 - e^{-1})$ to $1/2$. To achieve the tight $(1 - e^{-1})$-approximation under matroid constraint, then [15] proposed a continuous greedy algorithm for submodular functions. A key innovation of this continuous greedy algorithm is a novel continuous-relaxation technique termed as the *multi-linear extension*. Furthermore, there has been extensive research dedicated to the non-monotone submodular maximization [12, 13, 19, 22, 24, 37, 52, 82, 87, 123]

## A.2  Non-Submodular Maximization

Recently, numerous studies have found that there exist various real-world applications inducing utility functions that are close to submodular, but not strictly submodular. Examples include variable selection [30, 34], data summarization [46, 79, 113], neural network pruning [33, 104], target tracking [55, 56] and sparse optimal transport [77].

**Weakly Submodular Maximization.** A important class of close-to-submodular functions is known as $\gamma$-weakly submodular functions. Specifically, for a set function $f : 2^{\mathcal{V}} \to \mathbb{R}_+$, it is called $\gamma$-weakly submodular if and only if, for any two subsets $A \subseteq B \subseteq \mathcal{V}$, the following inequality holds: $\sum_{v \in B \setminus A} f(v|A) \geq \gamma\big(f(B) - f(A)\big)$. The $\gamma$-weakly submodular functions were originally introduced by the work [29]. Furthermore, when considering the simple cardinality constraint, [29] also show that the standard greedy algorithm can achieve an approximation ratio of $(1 - e^{-\gamma})$ for the maximization of a $\gamma$-weakly submodular function. Subsequently, [53] proved that, for any $\epsilon > 0$, no polynomial-time algorithm can achieve a $(1 - e^{-\gamma} + \epsilon)$-approximation for the problem of maximizing a $\gamma$-weakly submodular function subject to a cardinality constraint. As for more complicated matroid constraints, [20] showed that the residual random greedy method of [14] can achieve an approximation ratio of $\frac{\gamma^2}{(1+\gamma)^2}$ for the problem of maximizing a monotone $\gamma$-weakly submodular functions. After that, [63] examined the approximation performance of the standard greedy algorithm on the $\gamma$-weakly submodular maximization problem over a matroid constraint, which indicated that the standard greedy algorithm only can offer an approximation ratio of $\frac{0.4\gamma^2}{\sqrt{r\gamma}+1}$ where $r$ is the rank of the matroid. It is important to note that this approximation ratio $\frac{0.4\gamma^2}{\sqrt{r\gamma}+1}$ is not a constant guarantee and highly depends on the matroid rank $r$. In order to improve the approximation performance of these greedy-based algorithms over matroid constraints, [106] introduced the notion of upper submodularity ratio $\beta$ and developed a more powerful distorted local-search algorithm for $(\gamma, \beta)$-weakly submodular maximization problem. More importantly, this distorted local search can guarantee a $\frac{\gamma^2(1-e^{-(\beta(1-\gamma)+\gamma^2)})}{\beta(1-\gamma)+\gamma^2}$-approximation for the problem of maximizing a monotone $(\gamma, \beta)$-weakly submodular functions subject to a matroid constraint. Note that, when the $(\gamma, \beta)$-weakly submodular function is closer to being submodular, i.e., $\gamma, \beta \to 1$, the approximation ratio $\frac{\gamma^2(1-e^{-(\beta(1-\gamma)+\gamma^2)})}{\beta(1-\gamma)+\gamma^2}$ will approach the tight $(1-1/e)$. Conversely, when $\gamma, \beta \to 1$, the approximation ratio $\frac{\gamma^2}{(1+\gamma)^2}$ of the residual random greedy method [20] will trend to the sub-optimal $1/4$.

**Weakly DR-Submodular Maximization.** The other important class of non-submodular functions is known as $\alpha$-weakly DR-submodular functions, where $\alpha$ is variously referred to as the diminishing-

return(DR) ratio [68], the generalized curvature [9] or the generic submodularity ratio [48]. The work of [63] is the first to explore the problem of maximizing a $\alpha$-weakly DR-submodular maximization subject to general matroid constraints and proved that the standard greedy algorithm achieves approximation ratios of $\frac{\alpha}{1+\alpha}$ for the matroid-constrained $\alpha$-weakly DR-submodular maximization. Recently, [47] also showed that the continuous greedy combined with the contention resolution scheme [19] can obtain a sub-optimal approximation ratio of $(\alpha(1-1/e)(1-e^{-\alpha}))$ for the problem of maximizing a monotone $\alpha$-weakly DR-submodular functions subject to a matroid constraint. To achieve the tight $(1-e^{-\alpha})$-approximation guarantee, [76] presented a novel distorted local-search method for the problem of maximizing a $\alpha$-weakly DR-submodular maximization subject to al matroid constraint, which is motivated via the non-oblivious search [40, 39, 38].

### A.3 Multi-Agent Submodular Maximization

**Multi-Agent Offline Submodular Maximization.** Coordinating multiple agents to collaboratively maximize a submodular function is a critical task with numerous applications in machine learning, robot planning and control. A common solution for multi-agent submodular maximization problem heavily depends on the distributed implementation of the classic sequential greedy method [41], which can ensure a $(\frac{1}{1+c})$-approximation [26] when the submodular function possesses a curvature of $c \in [0,1]$. However, this distributed greedy algorithm requires each agent to have full access to the decisions of all previous agents, thereby forming a *complete* directed communication graph. Subsequently, several studies [50, 45, 78] have investigated how the topology of the communication network affects the performance of the distributed greedy method. Particularly, [50] pointed out that the worst-case performance of the distributed greedy algorithm will deteriorate in proportion to the size of the largest independent group of agents in the communication graph. In order to overcome these challenges, various studies [94, 95, 32] utilized the *multi-linear extension* [15] to design algorithms for solving mutli-agent submodular maximization problem. Specifically, [32] proposed a multi-agent variant of gradient ascent for mutli-agent submodular maximization problem and showed that this multi-agent variant of gradient ascent can attain $\frac{1}{2}OPT - \epsilon$ over any connected communication graph where $OPT$ is the optimal value. After that, to achieve the tight $(1-1/e)$-approximation, [95] developed a multi-agent variant of continuous greedy method [15, 19]. However, this multi-agent continuous greedy [95] requires the exact knowledge of the *multi-linear extension*, which will lead to the exponential number of function evaluations. To tackle this drawback, [94] also proposed a stochastic variant of continuous greedy method [15, 19], which can enjoy $(\frac{1-e^{-c}}{c})OPT - \epsilon$. Here, $c$ is the curvature of the investigated submodular objectives.

According to the hardness result in *centralized* submodular maximization [105], the optimal achievable approximation guarantee for maximizing a submodular function with curvature $c$ is $(1 - \frac{c}{e})$. However, as previously discussed, the state-of-the-art approximation guarantee for multi-agent offline submodular maximization problems is $(\frac{1-e^{-c}}{c})$, which highly mismatches the best possible guarantee of $(1 - \frac{c}{e})$ established in [105]. Thus, if in Algorithm 1 we treat any incoming objective function $f_t, \forall t \in [T]$ as a fixed submodular set objective $f$, our Algorithm 1 can be naturally transformed into an approximation algorithm with the tight $(1 - \frac{c}{e})$ guarantee for multi-agent offline submodular maximization problems. Similar to [126], we compare this offline version of our Algorithm 1 with existing algorithms for multi-agent offline submodular maximization problems in Table 2.

**Multi-Agent Online Submodular Maximization.** [115] is the first one to explore the multi-agent submodular maximization problems in time-varying environments. Moreover, [115] also proposed an *online sequence greedy*(OSG) algorithm for multi-agent online submodular maximization problems and proved this OSG algorithm can achieve a sub-optimal $(\frac{1}{1+c})$-approximation over a *complete* communication graph, where $c$ is the joint curvature of the investigated submodular objectives. Concurrently, [114] extended this OSG algorithm into bandit settings. In order to improve this sub-optimal $(\frac{1}{1+c})$-approximation guarantee and reduce the rigid requirement of a fully connected communication network of OSG algorithm, [126] utilized the non-oblivious auxiliary functions presented in [122] to design two multi-agent variants of online gradient ascent algorithm, namely, MA-OSMA algorithm and MA-OSEA algorithm, for the multi-agent online submodular maximization problem. Furthermore, [126] also showed that these two MA-OSMA and MA-OSEA algorithms can attain a regret bound of $\widetilde{O}(\sqrt{\frac{\mathcal{P}_T T}{1-\tau}})$ against a $(\frac{1-e^{-c}}{c})$-approximation to the best comparator in

Table 2: Comparison of the different algorithms for multi-agent offline submodular maximization problems. Note that 'Approx.' denotes the obtained approximation result, '#Com.' represents the number of communication, '#Queries' denotes the number of queries to the set objective functions, 'Proj-free' indicates whether the method does not require projection, 'Para-free' indicates whether the method does not require the knowledge of curvature $c$, $OPT$ denotes the optimal function value, $d(G)$ is the diameter of the graph $G$, $\tau$ is the spectral gap of the weight matrix, $\kappa \triangleq \sum_{i=1}^n \kappa_i$ and $\alpha(G) \geq 1$ is the number of nodes in the largest independent set in graph $G$.

| Method | Type | Para-free | Proj-free | Approx. | Graph($G$) | #Com. | #Queries | Reference |
|---|---|---|---|---|---|---|---|---|
| Greedy Method | det. | ✔ | ✔ | $(\frac{1}{1+c})OPT$ | **complete** | $\mathcal{O}(n)$ | $\mathcal{O}(\kappa n)$ | [26] |
| PGA | sto. | ✔ | ✗ | $\frac{1}{2}OPT-\epsilon$ | connected | $\mathcal{O}\big(\frac{1}{(1-\tau)\epsilon^2}\big)$ | $\mathcal{O}\big(\frac{\kappa}{(1-\tau)\epsilon^2}\big)$ | [32] |
| Greedy Method | det. | ✔ | ✔ | $(\frac{1}{1+\alpha(G)})OPT$ | connected | $\mathcal{O}(n)$ | $\mathcal{O}(\kappa n)$ | [50, 45] |
| CDCG | det. | ✔ | ✔ | $(1-\frac{1}{e})OPT-\epsilon$ | connected | $\mathcal{O}\big(\frac{1}{(1-\tau)\epsilon}\big)$ | $\mathcal{O}\big(\frac{\kappa 2^\kappa}{(1-\tau)\epsilon}\big)$ | [95] |
| Distributed-CG | sto. | ✔ | ✔ | $\big(\frac{1-e^{-c}}{c}\big)OPT-\epsilon$ | connected | $\mathcal{O}\big(\frac{d(G)}{\epsilon}\big)$ | $\widetilde{\mathcal{O}}\big(\frac{\kappa d^3(G)}{\epsilon^3}\big)$ | [94] |
| MA-OSMA | sto. | ✗ | ✗ | $\big(\frac{1-e^{-c}}{c}\big)OPT-\epsilon$ | connected | $\mathcal{O}\big(\frac{1}{(1-\tau)\epsilon^2}\big)$ | $\mathcal{O}\big(\frac{\kappa}{(1-\tau)\epsilon^2}\big)$ | [126] |
| MA-OSEA | sto. | ✗ | ✔ | $\big(\frac{1-e^{-c}}{c}\big)OPT-\epsilon$ | connected | $\mathcal{O}\big(\frac{1}{(1-\tau)\epsilon^2}\big)$ | $\mathcal{O}\big(\frac{\kappa\log(\frac{1}{\epsilon})}{(1-\tau)\epsilon^2}\big)$ | [126] |
| Algorithm 1 | sto. | ✔ | ✗ | $(1-\frac{c}{e})OPT-\epsilon$ | connected | $\mathcal{O}\big(\frac{1}{(1-\tau)\epsilon^2}\big)$ | $\mathcal{O}\big(\frac{\kappa}{(1-\tau)\epsilon^2}\big)$ | Theorem 4 |

hindsight, where $\mathcal{P}_T$ is the deviation of maximizer sequence, $\tau$ is the spectral gap of the network and $c$ is the joint curvature of submodular objectives.

## A.4 Multilinear Extension

As almost all state-of-the-art algorithms for multi-agent submodular coordination [94, 95, 126] rely on the *multi-linear extension* of [15], thus, in this subsection, we review the concept of *multi-linear extension* and compare it with our proposed policy-based continuous extension in Section 4.

At first, we define $\kappa \triangleq \sum_{i=1}^n \kappa_i$. Furthermore, from the definition of $\mathcal{V}$ in Section 2, we can know that $\kappa = |\mathcal{V}|$ such that we can re-define $\mathcal{V} \triangleq [\kappa] \triangleq \{1, \ldots, \kappa\}$. Then, we can show that

**Definition 3.** *For a set function $f : 2^\mathcal{V} \to \mathbb{R}_+$, we define its multi-linear extension as*

$$G(\mathbf{x}) = \sum_{\mathcal{A} \subseteq \mathcal{V}} \Big( f(\mathcal{A}) \prod_{a \in \mathcal{A}} x_a \prod_{a \notin \mathcal{A}} (1 - x_a) \Big) = \mathbb{E}_{\mathcal{R} \sim \mathbf{x}} \Big( f(\mathcal{R}) \Big), \qquad (6)$$

*where $\mathbf{x} = (x_1, \ldots, x_\kappa) \in [0,1]^\kappa$ and $\mathcal{R} \subseteq \mathcal{V}$ is a random set that contains each element $a \in \mathcal{V}$ independently with probability $x_a$ and excludes it with probability $1 - x_a$. We write $\mathcal{R} \sim \mathbf{x}$ to denote that $\mathcal{R} \subseteq \mathcal{V}$ is a random set sampled according to $\mathbf{x}$.*

With this *multi-linear extension*, we then can transfer the previous discrete subset selection problem (1) into a continuous maximization which aims at learning the optimal independent probability for each element $a \in \mathcal{V}$, that is,

$$\max_{\mathbf{x}} G(\mathbf{x}), \quad \text{s.t. } \mathbf{x} \in [0,1]^\kappa \text{ and } \sum_{a \in \mathcal{V}_i} x_a \leq 1, \forall i \in \mathcal{N}. \qquad (7)$$

It is important to note that, if we round any point $\mathbf{x}$ included into the constraint of problem (7) by the definition of multi-linear extension, namely, Eq.(6), there is a certain probability that the resulting subset will violate the partition constraint of the subset selection problem (1). Therefore, for *multi-linear extension*, we need to design a specific rounding methods based on the properties of the investigated set objective functions. However, current known lossless rounding schemes for multi-linear extension, such as pipage rounding [1], swap rounding [18] and contention resolution [19], are heavily dependent on the *submodular* assumption. Currently, how to losslessly round the multi-linear extension of *non-submodular* set functions, e.g. $(\gamma, \beta)$-weakly submodular and $\alpha$-weakly DR-submodular functions, still remains an open question [106]. Conversely, our proposed policy-based continuous extension in Section 4 does not assign probabilities to any subsets that are out of the partition constraint of problem (1), which means that, for any set function $f : 2^\mathcal{V} \to \mathbb{R}_+$ and any given policy vector $(\boldsymbol{\pi}_1, \ldots, \boldsymbol{\pi}_n) \in \prod_{i=1}^n \Delta_{\kappa_i}$, we can, through the Definition 1, easily produce a subset that conforms to the constraint of problem (1) without any loss in terms of the expected function value $F_t(\boldsymbol{\pi}_1, \ldots, \boldsymbol{\pi}_n)$.

# B  Parameter-free Multi-Agent Policy Learning

It is important to note that, when considering the weakly submodular scenarios, the Line 9 of Algorithm 1 requires the prior knowledge of the unknown submodularity ratio $(\gamma, \beta)$ or the diminishing-return(DR) ratio $\alpha$ to set the weight function $w(z)$. However, in general, accurately computing these parameters will incur exponential computations. To overcome this drawback, in this section, we explore how to design a *parameter-free* online algorithm for the MA-OC problem with $(\gamma, \beta)$-weakly submodular or $\alpha$-weakly DR-submodular objectives.

Note that, in part **4)** of Theorem 1, we establish a novel relationship between our proposed policy-based continuous extension $F_t$ and its original set function $f_t$. More specifically, when the $f_t$ is monotone $\alpha$-weakly DR-submodular or $(\gamma, \beta)$-weakly submodular, the weighted discrepancy between any $f_t(S)$ and its policy-based continuous extension $F_t(\boldsymbol{\pi}_1, \ldots, \boldsymbol{\pi}_n)$ can be bounded by some sum of first-order derivatives, i.e., $\sum_{(i,m):v_{i,m} \in S} \frac{\partial F_t}{\partial \pi_{i,m}}(\boldsymbol{\pi}_1, \ldots, \boldsymbol{\pi}_n)$. It is worth noting that the computation of this sum $\sum_{(i,m):v_{i,m} \in S} \frac{\partial F_t}{\partial \pi_{i,m}}(\boldsymbol{\pi}_1, \ldots, \boldsymbol{\pi}_n)$ does not rely on the knowledge of $\alpha, \gamma, \beta$. Therefore, we naturally consider whether it is possible to devise a *parameter-free* online algorithm for the MA-OC problem by controlling this aforementioned sum $\sum_{(i,m):v_{i,m} \in S} \frac{\partial F_t}{\partial \pi_{i,m}}(\boldsymbol{\pi}_1, \ldots, \boldsymbol{\pi}_n)$ to narrow the gap between $F_t(\boldsymbol{\pi}_1, \ldots, \boldsymbol{\pi}_n)$ and $f_t(S)$.

Before that, [21, 88, 121, 133] utilized the idea of meta-actions [103] to devise a Meta-Frank-Wolfe algorithm for online submodular maximization problems. The core of this Meta-Frank-Wolfe algorithm lies in iteratively optimizing an upper bound of the gap of the corresponding *multi-linear extension* $G_t$, i.e., $\langle \mathbf{y}, \nabla G_t(\mathbf{x}) \rangle \geq G_t(\mathbf{y}) - G_t(\mathbf{x})$, which is very similar to our previously discussed idea. Motivated by this finding, we then leverage the idea of meta-actions [103] and the insight from part **4)** of Theorem 1 to design a *parameter-free* `MA-MPL` algorithm for the MA-OC problem with $(\gamma, \beta)$-weakly submodular or $\alpha$-weakly DR-submodular objectives, as shown in Algorithm 2.

Like the Algorithm 1, the core of our Algorithm 2 is primarily composed of three interleaved components, namely, Policy Update and Information Exchange (Lines 5-10), Actions Sampling(Lines 11-13), Surrogate Gradient Estimation(Lines 9-16) and Batch Gradient Estimation and Linear Oracles Update (Lines 14-21). Firstly, at every time step $t \in [T]$, each agent $i \in \mathcal{N}$ employs $K$ online linear oracles $\{Q_i^{(1)}, \ldots, Q_i^{(K)}\}$ to mimic the process of decentralized Meta-Frank-Wolfe [81, 94] for maximizing our proposed policy-based continuous extension $F_t$. Specifically, during every inner iteration $k \in [K]$, each agent $i$ pushes the $i$-th local policy vector $\boldsymbol{\pi}_{i,i}^{(k-1)}$ along the direction $\mathbf{v}_i^{(k)} \in \Delta_{\kappa_i}$ provided by the oracle $Q_i^{(k)}$, while maintaining other policies $\boldsymbol{\pi}_{i,j}^{(k-1)}, j \neq i$ unchanged. Then, agent $i$ exchanges the updated policy vector $\left(\mathbf{y}_{i,1}^{(k)}(t), \ldots, \mathbf{y}_{i,n}^{(k)}(t)\right)$ with the neighboring agent $j \in \mathcal{N}_i$ and simultaneously utilizes these received information to initialize the next policy $\boldsymbol{\pi}_{i,m}^{(k)}$, i.e., $\boldsymbol{\pi}_{i,m}^{(k)} \triangleq \max_{j \in \mathcal{N}_i \cup \{i\}} \left(\mathbf{y}_{j,m}^{(k)}(t)\right), \forall m \in \mathcal{N}$, where $\mathbf{y}_{i,i}^{(k)}(t) \triangleq \boldsymbol{\pi}_{i,i}^{(k-1)}(t) + \frac{1}{K}\mathbf{v}_i^{(k)}(t)$ and $\mathbf{y}_{i,m}^{(k)}(t) \triangleq \boldsymbol{\pi}_{i,m}^{(k-1)}(t), \forall m \neq i$. After completing all $K$ iterations, each agent $i$ normalizes the final policy vector $\boldsymbol{\pi}_{i,i}^{(K)}(t)$ to select an action $a_i(t)$ from $\mathcal{V}_i$(See Lines 11-13). Next, in Lines 14-20, each agent $i$ uses a $L$-batch stochastic estimation $\mathbf{d}_i^{(k)}(t)$ to approximate the first-order partial derivatives of our proposed policy-based continuous extension $F_t$ at every policy vector $\left(\boldsymbol{\pi}_{i,1}^{(k)}(t), \ldots, \boldsymbol{\pi}_{i,n}^{(k)}(t)\right)$ and every coordinate $\pi_{i,m}, \forall m \in [\kappa_i]$. Finally, each agent $i$ feeds the obtained $L$-batch gradient estimation $\mathbf{d}_i^{(k)}(t)$ back to its corresponding linear oracle $Q_i^{(k)}$.

Note that in the process of Algorithm 2, each agent $i \in \mathcal{N}$ only needs to evaluate the margin contributions of the actions within its own action set $\mathcal{V}_i$(See Line 18).

It is important to highlight that, compared with the previous studies [21, 103, 126, 133], the innovations of our proposed `MA-MPL` algorithm are threefold: First, we utilize a policy-based continuous extension instead of the well-studied *multi-linear extension* to adjust the linear oracles in Lines 14-21. Second, rather than using a weight matrix to aggregate the received information, we employ the coordinate-wise *maximization operation* to update the policy vector(See Line 10). Third, to reduce communication complexity, we implement the batch gradient estimation in Lines 16-20.

Next, we provide the theoretical analysis for the proposed Algorithm 2. Before that, we introduce some standard assumptions about the linear maximization oracles $Q_i^{(k)}, \forall i \in \mathcal{N}, k \in [K]$, namely,

---

**Algorithm 2:** Multi-Agent Meta-Policy Learning(`MA-MPL`)

---

**Input:** Time horizon $T$, action set $\mathcal{V}_i \triangleq \{v_{i,1}, \ldots, v_{i,\kappa_i}\}$, Communication Graph $G(\mathcal{N}, \mathcal{E})$,
   number of online linear maximization oracles, namely, $K$ and batch size $L$

   `// Policy Vector and Online Linear Oracles Initialization (Lines 1-2)`

1  Initialize a policy vector $(\pi_{i,1}^{(0)}(t), \ldots, \pi_{i,n}^{(0)}(t)) \triangleq \mathbf{0}, \forall t \in [T]$ for each agent $i \in \mathcal{N}$;

2  Initialize $K$ online linear oracles over $\Delta_{\kappa_i}$, i.e., $\{Q_i^{(1)}, \ldots, Q_i^{(K)}\}$, for each agent $i \in \mathcal{N}$;

3  **for** $t = 1, \ldots, T$ **do**

4  $\quad$ **for** *each agent* $i \in \mathcal{N}$ **do**

$\qquad$ `// Policy Update and Information Exchange (Lines 5-10)`

5  $\qquad$ **for** $k = 1, \ldots, K$ **do**

6  $\qquad\quad$ Set the shared vector $\mathbf{y}_{i,m}^{(k)}(t) \triangleq \pi_{i,m}^{(k-1)}(t)$ for any $m \neq i$ and $m \in \mathcal{N}$;

7  $\qquad\quad$ Obtain the update direction $\mathbf{v}_i^{(k)}(t) \in \Delta_{\kappa_i}$ from oracle $Q_i^{(k)}$;

8  $\qquad\quad$ Compute $\mathbf{y}_{i,i}^{(k)}(t) \triangleq \pi_{i,i}^{(k-1)}(t) + \frac{1}{K}\mathbf{v}_i^{(k)}(t)$;

9  $\qquad\quad$ Exchange the vector $(\mathbf{y}_{i,1}^{(k)}(t), \ldots, \mathbf{y}_{i,n}^{(k)}(t))$ with the neighboring agent $j \in \mathcal{N}_i$;

10 $\qquad\quad$ Set $\pi_{i,m}^{(k)} \triangleq \max_{j \in \mathcal{N}_i \cup \{i\}} (\mathbf{y}_{j,m}^{(k)}(t)), \forall m \in \mathcal{N}$;

$\qquad$ `// Actions Sampling (Lines 11-13)`

11 $\qquad$ Compute the normalized policy $\mathbf{p}_i(t) \triangleq \pi_{i,i}^{(K)}(t) / \|\pi_{i,i}^{(K)}(t)\|_1$;

12 $\qquad$ Utilize the normalized policy $\mathbf{p}_i(t)$ to sample an action $a_i(t)$ from $\mathcal{V}_i$;

13 $\qquad$ Agent $i$ executes the sampled action $a_i(t)$;

$\qquad$ `// Batch Gradient Estimation and Linear Oracles Update(Lines 14-21)`

14 $\qquad$ **for** $k = 1, \ldots, K$ **do**

15 $\qquad\quad$ Set the gradient estimation $\mathbf{d}_i^{(k)}(t) := \mathbf{0}_{\kappa_i}$;

16 $\qquad\quad$ **for** $l = 1, \ldots, L$ **do**

17 $\qquad\qquad$ Utilize the policy $\pi_{i,j}^{(k)}(t)$ to sample an action $\widetilde{a}_j(t) \in \mathcal{V}_i \cup \{\emptyset\}$ for any $j \in \mathcal{N}$;

18 $\qquad\qquad$ Estimate the derivatives of $F_t$ based on Theorem 1, i.e.,

$$\widehat{\frac{\partial F_t}{\partial \pi_{i,m}}}(\pi_{i,1}^{(k)}(t), \ldots, \pi_{i,n}^{(k)}(t)) \triangleq g_{i,m}^{(k)}(t) \triangleq f_t(v_{i,m} \big| \cup_{j \neq i} \{\widetilde{a}_j(t)\}), \forall m \in [\kappa_i];$$

19 $\qquad\qquad$ Aggregate the gradient estimations $(g_{i,1}^{(k)}(t), \ldots, g_{i,\kappa_i}^{(k)}(t))$ as vector $\mathbf{g}_i^{(k)}(t)$;

20 $\qquad\qquad$ Update $\mathbf{d}_i^{(k)}(t) \triangleq \mathbf{d}_i^{(k)}(t) + \frac{1}{L}\mathbf{g}_i^{(k)}(t)$;

21 $\qquad\quad$ Feed back the batch gradient estimation $\mathbf{d}_i^{(k)}(t)$ to the linear oracle $Q_i^{(k)}$;

---

**Assumption 2.** *Each linear maximization oracle $Q_i^{(k)}$ can achieve a dynamic regret of $\mathcal{O}(\sqrt{V_T T})$ where $V_T$ is the variation of any feasible path $(\mathbf{u}_1, \ldots, \mathbf{u}_T) \in \prod_{t=1}^{T} \Delta_{\kappa_i}$, that is to say, $V_T \triangleq \sum_{t=2}^{T} \|\mathbf{u}_t - \mathbf{u}_{t-1}\|_2$ for any path $(\mathbf{u}_1, \ldots, \mathbf{u}_T) \in \prod_{t=1}^{T} \Delta_{\kappa_i}$*

**Remark 9.** *It is worth noting that there exist several effective and efficient algorithms that can achieve a regret bound of $\mathcal{O}(\sqrt{V_T T})$ for online linear maximization problem, for instance, online Frank-Wolfe [110] and online gradient ascent [116, 127, 134].*

**Theorem 5** (Proof provided in Appendix H). *Under Assumption 2, when the communication graph $G(\mathcal{N}, \mathcal{E})$ is connected and each set function $f_t$ is monotone $\alpha$-weakly DR-submodular or $(\gamma, \beta)$-weakly submodular, if we set $L = \mathcal{O}(T)$ and $K = \mathcal{O}(\sqrt{T})$, our proposed `MA-MPL` algorithm can achieve a dynamic $\rho$-regret bound of $\mathcal{O}\left(d(G)\sqrt{P_T T}\right)$, that is, $\mathbb{E}\left(R_\rho^*(T)\right) \leq \mathcal{O}\left(d(G)\sqrt{P_T T}\right)$, where $\rho = (1 - e^{-\alpha})$ or $\rho = \left(\frac{\gamma^2(1 - e^{-(\beta(1-\gamma)+\gamma^2)})}{\beta(1-\gamma)+\gamma^2}\right)$, respectively.*

**Remark 10.** *Note that, in Theorem 5, $d(G)$ represents the diameter of graph $G(\mathcal{N}, \mathcal{E})$, i.e., the length of the shortest path between the most distanced nodes. Moreover, $P_T \triangleq \sum_{t=2}^{T} |\mathcal{A}_t^* \triangle \mathcal{A}_{t-1}^*|$ is the deviation of maximizer sequence and $\triangle$ denotes the symmetric difference.*

# C Additional Experimental Details and Results

In this section, we test the effectiveness of our proposed Algorithm 1 and Algorithm 2 in two different multi-target tracking scenarios.

## C.1 Target Tracking with Facility-Location Objective Functions

In line with the prior studies [115, 126], we consider a 2-dimensional plane where 20 agents are deployed to track 30 moving targets over a duration of 25 seconds, subdivided into $T \triangleq 1250$ discrete iterations. At every iteration, agents must determine their movement direction from "up", "down", "left", "right", or "diagonally". Simultaneously, agents also need to adjust their speeds from a predefined set of 5, 10, or 15 units/s. As for targets, we categorize them into three different types: the unpredictable 'Random' and the structured 'Polyline' as well as the challenging 'Adversarial'. More specifically, at every iteration, 'Random' target will change its movement angle $\theta$ randomly from $[0, 2\pi]$ and steers at a random speed between 5 units/s and 10 units/s. In contrast, the 'Polyline' target will maintain its trajectory and only behaves like the 'Random' target at the specific $\{0, \lfloor \frac{T}{k} \rfloor, 2\lfloor \frac{T}{k} \rfloor \ldots, (k-1)\lfloor \frac{T}{k} \rfloor\}$-th iteration where $T$ is the predefined total iterations and $k$ is a random number from $\{1, 2, 4\}$. Regarding the 'Adversarial' target, it mimics the 'Random' targets when all agents are beyond a 20 units. Nevertheless, upon detecting an agent within the 20-unit range, the 'Adversarial' target will evade at a speed of 15 units/s for one second, pointing to the direction that maximizes the average distance of all agents.

Generally speaking, every motion of any agent can be characterized by three key parameters, namely, its movement angle $\theta$, speed $s$ and the unique identifier $i$. With all these three parameters, then the action set $\mathcal{V}_i$ available to each agent $i \in [20]$ can be mathematically represented as:

$$\mathcal{V}_i = \{(\theta, s, i) : s \in \{5, 10, 15\}\text{units/s}, \theta \in \{\frac{\pi}{4}, \frac{\pi}{2}, \frac{3\pi}{4}, \pi, \ldots, 2\pi\}\}, \forall i \in [20],$$

where each tuple $(\theta, s, i)$ encodes a specific action of agent $i$, that is, it will move at a speed of $s$ in the direction of $\theta$. Furthermore, at each time step $t \in [T]$, we denote the location of each target $j \in [30]$ as $o_j(t)$. Similarly, we also utilize the symbol $o_{(\theta,s,i)}(t)$ to represent the new position of agent $i$ after moving from its previous location at time $(t-1)$ with a movement angle $\theta$ and speed $s$.

To enhance the tracking quality, a common strategy is to minimize the distances between agents and targets. Inspired by this idea, many studies [27, 114, 115, 126] naturally consider the following facility-location objective function for agents at every iteration $t \in [T]$, that is,

$$f_t(S) \triangleq \sum_{j=1}^{30} \max_{(\theta,s,i) \in S} \frac{1}{\|o_{(\theta,s,i)}(t) - o_j(t)\|_2},$$

where $\|o_{(\theta,s,i)}(t) - o_j(t)\|_2$ represents the Euclidean distance between the location $o_j(t)$ of target $j$ and the new position $o_{(\theta,s,i)}(t)$ after agent $i$ executing the action $(\theta, s, i)$ and $S$ is a subset of the ground action set $\mathcal{V} \triangleq \cup_{i=1}^n \mathcal{V}_i$. It is important to note that the larger the value of $\frac{1}{\|o_{(\theta,s,i)}(t) - o_j(t)\|_2}$ becomes, the closer the action $(\theta, s, i)$ drives agent $i$ to the target $j$.

Considering this facility-location utility set function $f_t$ and the truth that each agent only can execute one decision from $\mathcal{V}_i$ at every time $t \in [T]$, then we can easily transform the aforementioned multi-target tracking problem as a multi-agent online coordination(MA-OC) problem introduced in Section 2. Particularly, at each time $t \in [T]$, we need to tackle the following facility-location utility set function maximization problem in a multi-agent manner, namely,

$$\max f_t(S), \quad \text{s.t. } S \subseteq \mathcal{V} \text{ and } |S \cap \mathcal{V}_i| = 1, \forall i \in \mathcal{N}. \tag{8}$$

Furthermore, numerous studies [91, 115, 126] have verified that this facility-location objective function $f_t$ is *monotone submodular*. As a result, the problem (8) can be equivalently reformulated as the problem (1) in Section 2, i.e.,

$$\max f_t(S), \quad \text{s.t. } S \subseteq \mathcal{V} \text{ and } |S \cap \mathcal{V}_i| \leq 1, \forall i \in \mathcal{N}.$$

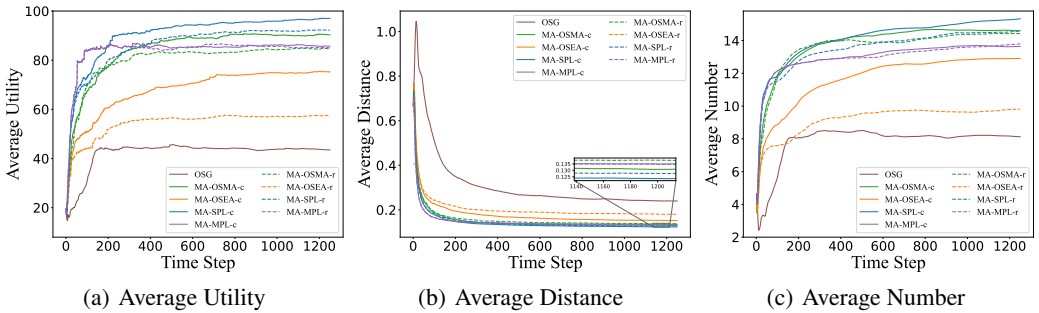

(a) Average Utility  (b) Average Distance  (c) Average Number

Figure 2: Comparison of the running average utility and the runing average number of targets within 1 unit as well as the running average distance of Top-5 nearest targets of OSG, MA-OSMA, MA-OSEA, MA-OSEA with our proposed `MA-SPL` and `MA-MPL` algorithms on the multi-target tracking scenario with 'Random':'Adversarial':'Polyline'=8:1:1.

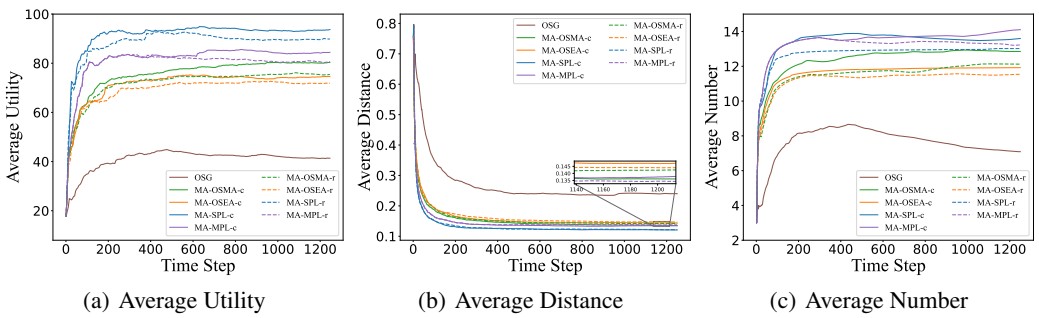

(a) Average Utility  (b) Average Distance  (c) Average Number

Figure 3: Comparison of the running average utility and the runing average number of targets within 1 unit as well as the running average distance of Top-5 nearest targets of OSG, MA-OSMA, MA-OSEA, MA-OSEA with our proposed `MA-SPL` and `MA-MPL` algorithms on the multi-target tracking scenario with 'Random':'Adversarial':'Polyline'=6:3:1.

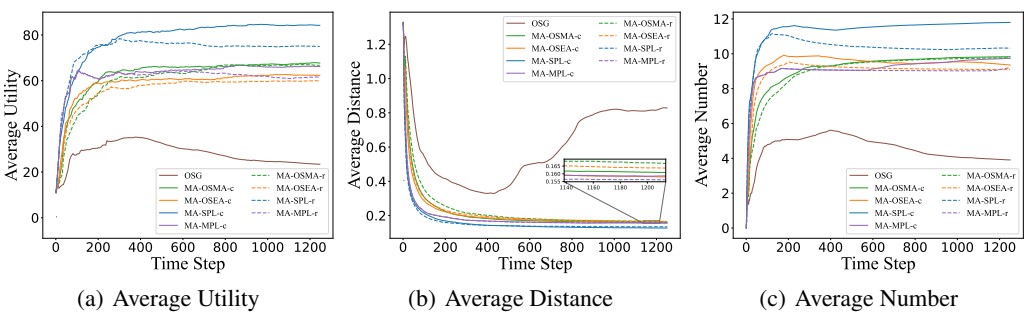

(a) Average Utility  (b) Average Distance  (c) Average Number

Figure 4: Comparison of the running average utility and the runing average number of targets within 1 unit as well as the running average distance of Top-5 nearest targets of OSG, MA-OSMA, MA-OSEA, MA-OSEA with our proposed `MA-SPL` and `MA-MPL` algorithms on the multi-target tracking scenario with 'Random':'Adversarial':'Polyline'=4:5:1.

In simulations, we initialize the starting positions of all agents and targets randomly within 20-unit radius circle centered at the origin. Furthermore, we consider different proportions of 'Random', 'Polyline', and 'Adversarial' targets. Specifically, we set the proportions of targets as 'Random':'Adversarial':'Polyline'=8:1:1 in Figure 2 and 6:3:1 in Figure 3 as well as 4:5:1 in Figure 4. Like [126], we also use the suffixes to represent two different choices for communication graphs, where 'c' stands for a complete graph and 'r' denotes an Erdos-Renyi random graph with average degree 4. According to the results in Figure 2(a), Figure 3(a) and Figure 4(a), we can find that the average utility of our proposed `MA-SPL` algorithm can significantly exceed the state-of-the-art MA-OSMA and MA-OSEA algorithms in [126], which is consistent with our Theorem 4, that is to say, our proposed `MA-SPL` algorithm can achieve a tight $(1 - \frac{c}{e})$-approximation guarantee

for submodular objectives, while the MA-OSMA and MA-OSEA in [126] only can guarantee a sub-optimal $\left(\frac{1-e^{-c}}{c}\right)$-approximation. Like [126], in Figure 2(b), Figure 3(b) and Figure 4(b), we also compare the average distance from agents to their closest five targets of our proposed MA-SPL and MA-MPL algorithms against OSG [115], MA-OSMA and MA-OSEA algorithms. Similarly, we observe that our proposed MA-SPL can effectively reduce the average distance between agents and targets. Furthermore, from Figure 2(c), Figure 3(c), and Figure 4(c), we also can infer that, compared with OSG, MA-OSMA, and MA-OSEA algorithms, our proposed MA-SPL can make more targets gather within 1 unit of agents. It is worth noting that as the proportion of 'Adversarial' targets increases, both the maximum runing average utility and the maximum running average number of targets within 1 unit exhibit a downward trend.

## C.2 Target Tracking under Extended Kalman-Filter Framework

In previous Appendix C.1, we considered a simplified multi-target tracking model, that is, we assume that the monitoring quality of each moving target $j$ only depends on its nearest agent. However, in many real-world scenarios, due to agents' different sensing capabilities and varying observation angles, relying *solely* on the information collected by the nearest agent for accurately tracking the state of each target is unrealistic. Instead, we generally need to aggregate the data from multiple agents to comprehensively reconstruct the targets' behaviors.

Recently, in order to obtain an accurate estimation of every target's location, [55, 56] employed an extend Kalman-filter(EKF) framework to process the observations of multiple distinct agents. Specifically, let us consider a target tracking task using a swarm of agents(UAVs) equipped with GPS and radar systems. In this scenario, at each time $t \in [T]$, each agent $i$ can measure the range to each target $j$ throughout its radar system and also can obtain its own position information from GPS. Furthermore, like the work [55], we also assume that the range measurements of the radar systems follow a quadratic model, namely,

$$r_{(\theta,s,i) \to j}(t) = \frac{1}{2}\|o_{(\theta,s,i)}(t) - o_j(t)\|_2^2 + \xi_{(\theta,s,i) \to j}(t), \tag{9}$$

where the symbol $r_{(\theta,s,i) \to j}(t)$ denotes the range measurement of agent $i$ to target $j$ after agent $i$ executes the action $(\theta, s, i)$, $o_j(t)$ is the location of target $j$, $\xi_{(\theta,s,i) \to j}(t)$ is the additive independent noise and the symbol $o_{(\theta,s,i)}(t)$ represents the new position of agent $i$ after moving from its previous location at time $t-1$ with a movement angle $\theta$ and speed $s$.

Note that when the new position $o_{(\theta,s,i)}(t)$ and the range measurement $r_{(\theta,s,i) \to j}(t)$ are known, we can view Equation (9) as a random quadratic experiment of the unknown location $o_j(t)$. In other words, different action $(\theta, s, i)$ can lead to distinct random quadratic observation of the unknown parameter $o_j(t)$. Inspired by this perspective, we can formulated the action selection problem in multi-target tracking task as an experimental design problem, which aims at selecting a feasible subset from the whole collection of experiments $\{r_{(\theta,s,i) \to j}(t) = \frac{1}{2}\|o_{(\theta,s,i)}(t) - o_j(t)\|_2^2 + \xi_{(\theta,s,i) \to j}(t)\big|(\theta, s, i) \in \mathcal{V}\}$ such that the measurements of these selected sub-experiments can accurately estimate the location of every target. Particularly, under the classical Van Trees' inequality [108], the work [55] established a lower bound for the covariance matrix associated with the EKF estimator of the location $o_j(t)$. Specifically, at time step $t \in [T]$, if we consider the sub-experiment $S \subseteq \mathcal{V}$, then the lower-bound matrix $\mathbf{B}_S^j(t)$ in the Van Trees' inequality for the EKF estimate of the unknown location $o_j(t)$ can be expressed as (Please refer to Theorem 2 in [55]):

$$\mathbf{B}_S^j(t) = \left( \sum_{(\theta,s,i) \in S} \frac{1}{\sigma_{(\theta,s,i) \to j}^2(t)} \left( \mathbf{P} + \mathbf{z}_{(\theta,s,i) \to j} \mathbf{z}_{(\theta,s,i) \to j}^T \right) + \mathbf{I}_j(t) \right)^{-1}, \tag{10}$$

where $\mathbf{z}_{(\theta,s,i) \to j} \triangleq o_{(\theta,s,i)}(t) - o_j(t-1) - \mathbb{E}\left(o_j(t) - o_j(t-1)\right)$, $\sigma_{(\theta,s,i) \to j}^2(t) \triangleq \mathrm{Var}\left(\xi_{(\theta,s,i) \to j}(t)\right)$, $\mathbf{P} \triangleq \mathrm{Cov}\left(o_j(t) - o_j(t-1)\right)$ and $\mathbf{I}_j(t)$ is the Fisher information matrix associated with the normalized random gap $\left(o_j(t) - o_j(t-1) - \mathbb{E}\left(o_j(t) - o_j(t-1)\right)\right)$. Note that in Equation (10), the tuple $(\theta, s, i)$ represents not only a selected action but also an observation experiment.

With this lower-bound covariance matrix $\mathbf{B}_S^j(t)$, we then can utilize various so-called alphabetical criteria [16, 17] to design a utility function for selecting a superior action set such that the resulting

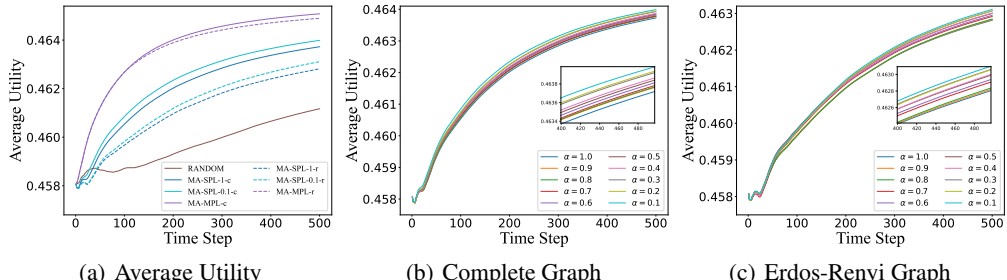

(a) Average Utility      (b) Complete Graph      (c) Erdos-Renyi Graph

Figure 5: In Figure 5(a), we compare the running average utility of our proposed `MA-SPL` and `MA-MPL` algorithms with that of the 'RANDOM' baseline in a multi-target tracking simulation with A-optimal objective functions. In 'RANDOM' baseline, each agent $i$ randomly selects an action from its own action set $\mathcal{V}_i$ at every iteration. As for Figure 5(b) and Figure 5(c), we illustrate the impact of different DR ratios $\alpha \in \{0.1, 0.2, \ldots, 1\}$ on `MA-SPL` algorithm. Particularly in Figure 5(b), we considers a complete communication graph among agents, whereas we employs an Erdos-Renyi graph with an average degree of 4 in Figure 5(c). Note that Figure 5(a) only shows the results for the best scenario ($\alpha = 0.1$) and the worst-case scenario ($\alpha = 1$) for our proposed `MA-SPL` algorithm.

EKF estimation can accurately approximate the location of each target. One commonly used strategy is to employ the A-optimality, i.e., we consider minimizing the trace of the lower-bound covariance matrix $\mathbf{B}_S^j(t)$ or equivalently maximize:

$$f_t(S) = \sum_{j=1}^{30} \left( \mathrm{Tr}\left(\mathbf{I}_j^{-1}(t)\right) - \mathrm{Tr}\left(\mathbf{B}_S^j(t)\right) \right) \tag{11}$$

where "Tr" is the trace of matrix and we also consider 30 moving targets in simulation.

Thus, under the A-optimality criterion, we can model the agents' action selection task as a special instance of the multi-agent online coordination problem with the set objective function $f_t$ defined in Equation (11). Furthermore, according to recent studies [53, 55, 106], we can show that the utility function $f_t$ in Equation (11) is monotone $\alpha$-weakly DR-submodular and $(\gamma, \beta)$-weakly submodular (See Theorem 6 in [55] and Theorem C.2 in [106]).

In our simulations, to simplify the computation of the lower-bound matrix $\mathbf{B}_S^j(t)$, we model the movement of each target as a two-dimensional Brownian motion. Specifically, we set $o_j(t) \triangleq o_j(t-1) + 0.02 * \mathcal{N}(\mathbf{0}_2, \mathbf{I}_2)$ where $\mathbf{I}_2$ is the 2-dimensional identity matrix. Moreover, we assume that the noise $\xi_{(\theta,s,i) \to j}$ follows an independent normal distribution, i.e., $\xi_{(\theta,s,i) \to j} \sim \mathcal{N}(0, 0.01)$. As for agents, at every iteration $t \in [T]$, we adjust their speeds from a set of 2, 7, or 12 units/s and simultaneously change their movement directions from "up", "down", "left", "right", or "diagonally". As a result, the action set $\mathcal{V}_i$ available to each agent $i \in [20]$ can be mathematically formulated as:

$$\mathcal{V}_i = \{(\theta, s, i) : s \in \{2, 7, 12\}\text{units/s}, \theta \in \{\frac{\pi}{4}, \frac{\pi}{2}, \frac{3\pi}{4}, \pi, \ldots, 2\pi\}\}, \forall i \in [20],$$

where $\theta$ denotes the movement angle , $s$ is the speed and $i$ represents the unique identifier.

Given the unknown diminishing-return(DR) ratio $\alpha$ of our investigated set function $f_t$, we test 10 different configurations from $\alpha = 0.1$ to $\alpha = 1$ for our proposed `MA-SPL` and then present the results in Figure 5(b) and Figure 5(c). Particularly in Figure 5(b), we considers a complete communication graph among agents, whereas we employs an Erdos-Renyi graph with an average degree of 4 in Figure 5(c). Subsequently, we compare the results of the best-case scenario ($\alpha = 0.1$) and the worst-case scenario ($\alpha = 1$) of our proposed `MA-SPL` algorithm with the *parameter-free* `MA-MPL` algorithm and 'RANDOM' baseline in Figure 5(a). According to the curves in Figure 5(a), we can find that the running average utility of our proposed `MA-MPL` and `MA-SPL` algorithms can significantly exceed the baseline 'RANDOM' algorithm, which is in accord with our Theorem 4 and Theorem 5. Moreover, we also find that the *parameter-free* `MA-MPL` algorithm can effectively outperform the `MA-SPL` algorithm associated with a 0.1-network search. It is worth noting that no previous works explore the MA-OC problem with weakly submodular objectives. Thus, we adopt the 'RANDOM' algorithm as a baseline in Figure 5(a). In 'RANDOM', each agent $i$ randomly selects an action from

its own action set $\mathcal{V}_i$. Like the previous Appendix C.1, we also use the suffixes to represent two different choices for communication graphs in Figure 5(a), where 'c' stands for a complete graph and 'r' denotes an Erdos-Renyi random graph with average degree 4.

## C.3  More Details on Experimental Setups

This subsection discusses some additional details about our experiments.

At first, we describe the parameter configurations about our proposed 'MA-SPL', 'MA-MPL', 'MA-OSMA' and 'MA-OSEA'. Specifically, we make the following setups:

- In 'MA-SPL', namely, the Algorithm 1, we set the step size $\eta_t = \frac{1}{\sqrt{T}}$ and employ a 10 batch of stochastic estimation to approximate the surrogate gradient of our proposed policy-based continuous extension in Section 5.1. As for the projection in Line 19 of Algorithm 1, we utilize the CVX optimization solver [49].

- In 'MA-MPL', namely, the Algorithm 2 of Appendix B, we set the batch size $L = 10$ and the number of oracles $K = 15$. As for the online linear maximization oracle, we utilize the online gradient ascent algorithm [116, 134] with step size $\eta = \mathcal{O}(\frac{1}{\sqrt{T}})$.

- In 'MA-OSMA', namely Algorithm 1 of [126], we consider the Euclidean distance with $\phi(\mathbf{x}) = \frac{\|\mathbf{x}\|_2^2}{2}$, set the step size $\eta_t = \frac{1}{\sqrt{T}}$ and implement a 10 batch of stochastic estimation to approximate the surrogate gradient of *multi-linear extension*. Similarly, we also use the CVX solver for projection operations.

- In 'MA-OSEA', namely Algorithm 2 of [126], we also set the step size $\eta_t = \frac{1}{\sqrt{T}}$ and consider the mixing parameter $\gamma = 1/T^{1.5}$.

As for the communication graph $G$, we consider two different setups:

- 'Complete graph' where we set the weight $w_{ij} = \frac{1}{n}, \forall i, j \in [n]$ where $n = |\mathcal{N}|$ is the number of agents.

- 'Erdos-Renyi random graph with average degree 4 where if the edge $(i, j)$ is an edge of the graph, we set $w_{ij} \triangleq 1/(1 + \max(d_i, d_j))$ where $d_i$ is the degree of agent $i \in \mathcal{N}$ and when $(i, j)$ is not an edge of the graph and $i \neq j$, we consider $w_{ij} = 0$. Finally, we set $w_{ii} \triangleq 1 - \sum_{j \in \mathcal{N}_i} w_{ij}$ where $\mathcal{N}_i$ is the neighboring nodes of agent $i$.

Furthermore, for all curves related to MA-SPL, MA-MPL, MA-OSMA, MA-OSEA and OSG, we repeat these algorithms **five runs** and then report the average result. Note that, in Figure 2(a), Figure 3(a), Figure 4(a) and Figure 5(a), the running average utility at any time $t$ is defined as $\left(\sum_{t_1 \in [t]} \frac{f_t(\cup_{i \in [n]} \{a_i(t_1)\})}{t}\right)$ where $a_i(t_1)$ is the action chosen by agent $i \in [n]$ at time $t_1$.

## D  Proof of Theorem 1

In this section, we prove the Theorem 1.

**1):** From Definition 1, we have that

$$
\frac{\partial F_t}{\partial \pi_{i,m}}(\boldsymbol{\pi}_1, \ldots, \boldsymbol{\pi}_n)
$$

$$
= \sum_{a_j \in \mathcal{V}_j \cup \{\emptyset\}, \forall j \in \mathcal{N}} \left( f_t\big( \cup_{j=1}^n \{a_j\} \big) \frac{\partial \left( \prod_{j=1}^n p(a_j|\boldsymbol{\pi}_j) \right)}{\partial \pi_{i,m}} \right) \tag{12}
$$

$$
= \sum_{a_j \in \mathcal{V}_j \cup \{\emptyset\}, \forall j \in \mathcal{N}} \left( f_t\big( \cup_{j=1}^n \{a_j\} \big) \frac{\partial p(a_i|\boldsymbol{\pi}_i)}{\partial \pi_{i,m}} \prod_{j \neq i, j \in \mathcal{N}} p(a_j|\boldsymbol{\pi}_j) \right).
$$

Note that $p(v_{i,m}|\boldsymbol{\pi}_i) = \pi_{i,m}, \forall i \in [n], \forall m \in [\kappa_i]$ and $p(\emptyset|\boldsymbol{\pi}_i) = 1 - \sum_{m=1}^{K_i} \pi_{i,m}, \forall i \in \mathcal{N}$. Therefore, we have $\frac{\partial p(a_i|\boldsymbol{\pi}_i)}{\partial \pi_{i,m}} = 1$ when $a_i = v_{i,m}$, $\frac{\partial p(a_i|\boldsymbol{\pi}_i)}{\partial \pi_{i,m}} = -1$ when $a_i = \emptyset$ and $\frac{\partial p(a_i|\boldsymbol{\pi}_i)}{\partial \pi_{i,m}} = 0$ when

$a_i \notin \{v_{i,m}, \emptyset\}$. Then, according to Eq.(12), we can get the following equality:

$$\frac{\partial F_t}{\partial \pi_{i,m}}(\pi_1, \ldots, \pi_n)$$

$$= \sum_{a_j \in \mathcal{V}_j \cup \{\emptyset\}, \forall j \in \mathcal{N}} \left( f_t\big( \cup_{j=1}^n \{a_j\} \big) \frac{\partial p(a_i|\pi_i)}{\partial \pi_{i,m}} \prod_{j \neq i, j \in \mathcal{N}} p(a_j|\pi_j) \right)$$

$$= \sum_{a_j \in \mathcal{V}_j \cup \{\emptyset\}, \forall j \neq i} \left( \sum_{a_i \in \mathcal{V}_i} \left( f_t\big( \cup_{j=1}^n \{a_j\} \big) \frac{\partial p(a_i|\pi_i)}{\partial \pi_{i,m}} \prod_{j \neq i, j \in \mathcal{N}} p(a_j|\pi_j) \right) \right)$$

$$= \sum_{a_j \in \mathcal{V}_j \cup \{\emptyset\}, \forall j \neq i} \left( \Big( f_t\big(\{v_{i,m}\} \bigcup (\cup_{j \neq i}\{a_j\}) \big) - f_t\big( \cup_{j \neq i, j \in \mathcal{N}} \{a_j\} \big) \Big) \prod_{j \neq i, j \in \mathcal{N}} p(a_j|\pi_j) \right)$$

$$= \sum_{a_j \in \mathcal{V}_j \cup \{\emptyset\}, \forall j \neq i} \left( f_t\big(v_{i,m} \big| \cup_{j \neq i, j \in \mathcal{N}} \{a_j\} \big) \prod_{j \neq i, j \in \mathcal{N}} p(a_j|\pi_j) \right)$$

$$= \mathbb{E}_{a_j \sim \pi_j, \forall j \in \mathcal{N}} \Big( f_t\big(v_{i,m} \big| \cup_{j \neq i, j \in \mathcal{N}} \{a_j\} \big) \Big),$$

(13)

where $a_j \sim \pi_j$ indicates that action $a_j$ is randomly selected from $\mathcal{V}_j \cup \{\emptyset\}$ based on the policy $\pi_j$.

**2):** When $f_t$ is monotone, we can know that $f_t\big(v_{i,m} \big| \cup_{j \neq i, j \in \mathcal{N}} \{a_j\}\big) \geq 0$ such that $\frac{\partial F_t}{\partial \pi_{i,m}}(\pi_1, \ldots, \pi_n) \geq 0$. In other words, for any $(\pi_1, \ldots, \pi_n) \in \prod_{i=1}^n \Delta_{\kappa_i}$, we have $\nabla F_t(\pi_1, \ldots, \pi_n) \geq \mathbf{0}$, namely, for any two point $(\pi_1^a, \ldots, \pi_n^a) \in \prod_{i=1}^n \Delta_{\kappa_i}$ and $(\pi_1^b, \ldots, \pi_n^b) \in \prod_{i=1}^n \Delta_{\kappa_i}$, if $\pi_i^a \leq \pi_i^b$ for any $i \in \mathcal{N}$, we have $F_t(\pi_1^a, \ldots, \pi_n^a) \leq F_t(\pi_1^b, \ldots, \pi_n^b)$. So $F_t$ is monotone.

**3):** For any fixed two policy vector $(\pi_1^a, \ldots, \pi_n^a) \in \prod_{i=1}^n \Delta_{\kappa_i}$ and $(\pi_1^b, \ldots, \pi_n^b) \in \prod_{i=1}^n \Delta_{\kappa_i}$ where $\pi_i^a \leq \pi_i^b, \forall i \in \mathcal{N}$, we consider an unified sampling strategy to generate actions. Before that, we set each $\pi_i^a = (\pi_{i,1}^a, \ldots, \pi_{i,\kappa_i}^a)$ and $\pi_i^b = (\pi_{i,1}^b, \ldots, \pi_{i,\kappa_i}^b), \forall i \in \mathcal{N}$. At first, we transfer the sampling process according to policy $\pi_i^a$ to a uniform random variable $X_i \in [0, 1]$, namely,

$$a(X_i, \pi_i^a) \triangleq \begin{cases} v_{i,1} & \text{If } X_i \in [0, \pi_{i,1}^a) \\ v_{i,m} & \text{If } X_i \in \big[ \sum_{k=1}^{m-1} \pi_{i,k}^a, \sum_{k=1}^{m} \pi_{i,k}^a \big) \text{ for any integer } m \in [2, \kappa_i] \\ \emptyset & \text{If } X_i \geq \sum_{k=1}^{\kappa_i} \pi_{i,k}^a \end{cases}$$

(14)

When $X_i$ is an uniform random variable over range $[0, 1]$, it is easy to check that the $a(X_i, \pi_i^a)$ follows the same law as the policy $\pi_i^a$, that is, $\Pr(a(X_i, \pi_i^a) = v_{i,m}) = \pi_{i,m}^a$ and $\Pr(a(X_i, \pi_i^a) = \emptyset) = 1 - \sum_{m=1}^{\kappa_i} \pi_{i,m}^a$ where the symbol 'Pr' denotes the probability.

Similarly, we also can transfer the sampling process according to policy $\pi_i^b$ to two independent uniform random variables $X_i, Y \in [0, 1]$, namely,

$$a(X_i, Y, \pi_i^a, \pi_i^b) \triangleq \begin{cases} a(X_i, \pi_i^a) & \text{If } X_i < \sum_{k=1}^{\kappa_i} \pi_{i,k}^a \\ v_{i,1} & \text{If } X_i \geq \sum_{k=1}^{\kappa_i} \pi_{i,k}^a \text{ and } Y \in \big[0, \frac{\pi_{i,1}^b - \pi_{i,1}^a}{1 - \sum_{k=1}^{\kappa_i} \pi_{i,k}^a} \big) \\ v_{i,m} & \text{If } X_i \geq \sum_{k=1}^{\kappa_i} \pi_{i,k}^a \text{ and } Y \in \big[ \frac{\sum_{k=1}^{m-1}(\pi_{i,k}^b - \pi_{i,k}^a)}{1 - \sum_{k=1}^{\kappa_i} \pi_{i,k}^a}, \frac{\sum_{k=1}^{m}(\pi_{i,k}^b - \pi_{i,k}^a)}{1 - \sum_{k=1}^{\kappa_i} \pi_{i,k}^a} \big), \forall m \in [2, \kappa_i] \\ \emptyset & \text{If } X_i \geq \sum_{k=1}^{\kappa_i} \pi_{i,k}^a \text{ and } Y \geq \sum_{k=1}^{\kappa_i}(\pi_{i,k}^b - \pi_{i,k}^a) \end{cases}$$

(15)

From part **1)** and fixed a sequence of independent uniform variables $X_i, Y_i, \forall i \in \mathcal{N}$, we can have that

$$\frac{\partial F_t}{\partial \pi_{i,m}}(\pi_1^a, \ldots, \pi_n^a) = \mathbb{E}_{X_i}\Big(f_t\big(v_{i,m}\big|\cup_{j \neq i, j \in \mathcal{N}} \{a(X_i, \pi_i^a)\}\big)\Big),$$

and

$$\frac{\partial F_t}{\partial \pi_{i,m}}(\pi_1^b, \ldots, \pi_n^b) = \mathbb{E}_{(X_i, Y_i)}\Big(f_t\big(v_{i,m}\big|\cup_{j \neq i, j \in \mathcal{N}} \{a(X_i, Y_i, \pi_i^a, \pi_i^b)\}\big)\Big).$$

From Eq.(14) and Eq.(15), we can know that $\{a(X_i, \pi_i^a)\} \subseteq \{a(X_i, Y_i, \pi_i^a, \pi_i^b)\}$ such that $\cup_{j \neq i, j \in \mathcal{N}}\{a(X_i, \pi_i^a)\} \subseteq \cup_{j \neq i, j \in \mathcal{N}}\{a(X_i, Y_i, \pi_i^a, \pi_i^b)\}$. Then, from the definition of $\alpha$-DR submodularity, we have that $f_t\big(v_{i,m}\big|\cup_{j \neq i, j \in \mathcal{N}} \{a(X_i, \pi_i^a)\}\big) \geq \alpha f_t\big(v_{i,m}\big|\cup_{j \neq i, j \in \mathcal{N}} \{a(X_i, Y_i, \pi_i^a, \pi_i^b)\}\big)$, so $\nabla F_t(\pi_1^a, \ldots, \pi_n^a) \geq \alpha \nabla F_t(\pi_1^b, \ldots, \pi_n^b)$.

**4):** For any subset $S$ within the constraint of problem (1) and any point $(\pi_1, \ldots, \pi_n) \in \prod_{i=1}^n \Delta_{\kappa_i}$, when $f_t$ is monotone $(\gamma, \beta)$-weakly monotone submodular, we firstly can show that for any two actions $a_1, a_2 \in \mathcal{V}$ and the subset $B \subseteq \mathcal{V}$, we have that

$$\gamma\big(f_t(a_1|B \cup \{a_2\}) + f_t(a_2|B)\big) = \gamma\big(f_t(B \cup \{a_1, a_2\}) - f(B)\big) \leq f_t(a_1|B) + f_t(a_2|B),$$

such that

$$f_t(a_1|B) \geq \gamma f_t(a_1|B \cup \{a_2\}) - (1-\gamma)f_t(a_2|B). \tag{16}$$

Therefore, we can show that

$$\sum_{(i,m): v_{i,m} \in S} \frac{\partial F_t}{\partial \pi_{i,m}}(\pi_1, \ldots, \pi_n)$$

$$= \sum_{(i,m): v_{i,m} \in S} \mathbb{E}_{a_j \sim \pi_j, \forall j \in \mathcal{N}}\Big(f_t\big(v_{i,m}\big|\cup_{j \neq i, j \in \mathcal{N}} \{a_j\}\big)\Big) \tag{17}$$

$$\geq \sum_{(i,m): v_{i,m} \in S} \mathbb{E}_{a_j \sim \pi_j, \forall j \in \mathcal{N}}\Big(\gamma f_t\big(v_{i,m}\big|\cup_{j \in \mathcal{N}} \{a_j\}\big) - (1-\gamma)f_t\big(a_i\big|\cup_{j \neq i, j \in \mathcal{N}} \{a_j\}\big)\Big),$$

where the first equality follows from part **1)** of Theorem 1 and the second inequality comes from the Eq.(16). Then, from the $\gamma$-weakly submodularity,

$$\sum_{(i,m): v_{i,m} \in S} \mathbb{E}_{a_j \sim \pi_j, \forall j \in \mathcal{N}}\Big(f_t\big(v_{i,m}\big|\cup_{j \in \mathcal{N}} \{a_j\}\big)\Big)$$

$$\geq \gamma \mathbb{E}_{a_j \sim \pi_j, \forall j \in \mathcal{N}}\Big(\gamma f_t\big(S\big|\cup_{j \in \mathcal{N}} \{a_j\}\big)\Big) \tag{18}$$

$$\geq \gamma\Big(f_t(S) - F_t(\pi_1, \ldots, \pi_n)\Big),$$

where the final inequality follows from the monotonicity.

Furthermore, from the $\beta$-weakly upper submodularity, we also have,

$$\sum_{(i,m): v_{i,m} \in S} \mathbb{E}_{a_j \sim \pi_j, \forall j \in \mathcal{N}}\Big(f_t\big(a_i\big|\cup_{j \neq i, j \in \mathcal{N}} \{a_j\}\big)\Big)$$

$$\leq \sum_{i \in \mathcal{N}} \mathbb{E}_{a_j \sim \pi_j, \forall j \in \mathcal{N}}\Big(f_t\big(a_i\big|\cup_{j \neq i, j \in \mathcal{N}} \{a_j\}\big)\Big) \tag{19}$$

$$\leq \beta \mathbb{E}_{a_j \sim \pi_j, \forall j \in \mathcal{N}}\Big(f_t\big(\cup_{j \in \mathcal{N}} \{a_j\}\big) - f_t(\emptyset)\Big)$$

$$= \beta F_t(\pi_1, \ldots, \pi_n),$$

where the second inequality follows from the definition of $\beta$-upper submodularity.

Merging Eq.(19) and Eq.(18) into Eq.(17), we have that

$$\sum_{(i,m): v_{i,m} \in S} \frac{\partial F_t}{\partial \pi_{i,m}}(\pi_1, \ldots, \pi_n)$$

$$\geq \sum_{(i,m): v_{i,m} \in S} \mathbb{E}_{a_j \sim \pi_j, \forall j \in \mathcal{N}}\Big(\gamma f_t\big(v_{i,m}\big|\cup_{j \in \mathcal{N}} \{a_j\}\big) - (1-\gamma)f_t\big(a_i\big|\cup_{j \neq i, j \in \mathcal{N}} \{a_j\}\big)\Big)$$

$$\geq \big(\gamma^2\big(f_t(S) - F_t(\pi_1, \ldots, \pi_n)\big) - (1-\gamma)\beta F_t(\pi_1, \ldots, \pi_n)\big)$$

$$= \gamma^2 f_t(S) - \big((1-\gamma)\beta + \gamma^2\big)F_t(\pi_1, \ldots, \pi_n)).$$

As for $f_t$ is monotone $\alpha$-weakly DR-submodular, we also can show that

$$\sum_{(i,m):v_{i,m}\in S} \frac{\partial F_t}{\partial \pi_{i,m}}(\boldsymbol{\pi}_1,\ldots,\boldsymbol{\pi}_n) = \mathbb{E}_{a_j\sim\pi_j,\forall j\in\mathcal{N}}\Big(\sum_{(i,m):v_{i,m}\in S} f_t\big(v_{i,m}\big|\cup_{j\neq i,j\in\mathcal{N}}\{a_j\}\big)\Big)$$

$$\geq \alpha\mathbb{E}_{a_j\sim\pi_j,\forall j\in\mathcal{N}}\Big(f_t\left(S\big|\cup_{j\in\mathcal{N}}\{a_j\}\right)\Big)$$

$$\geq \alpha\left(f_t(S) - F_t(\boldsymbol{\pi}_1,\ldots,\boldsymbol{\pi}_n)\right),$$

where the first inequality follows from the $\alpha$-DR submodularity and the final inequality comes from the monotonicity of $f_t$.

# E  Proof of Theorem 2

In this section, we verify the Theorem 2.

Before that, we firstly suppose that the symbol $\mathbf{1}_S$ represents the indicator function over the subset $S \subseteq \mathcal{V}$, namely, for any action $v_{i,m} \in S$, the vector $\mathbf{1}_S$ sets the corresponding probability for this action $v_{i,m}$ as 1. Note that, when $S$ satisfies the constraints of problem (1), namely, $|S\cap\mathcal{V}_i|\leq 1$, we can infer that $\mathbf{1}_S \in \prod_{i=1}^{n}\Delta_{\kappa_i}$.

## E.1  Proof of Part 2) and Part 3) in Theorem 2

With the previously defined symbol $\mathbf{1}_S$, we can rewrite the **4)** of Theorem 1 as:

**i)**: when $f_t$ is monotone $\alpha$-weakly DR-submodular, the following inequality holds:

$$\alpha\Big(f_t(S) - F_t(\boldsymbol{\pi}_1,\ldots,\boldsymbol{\pi}_n)\Big) \leq \langle\mathbf{1}_S,\nabla F_t(\boldsymbol{\pi}_1,\ldots,\boldsymbol{\pi}_n)\rangle;$$

**ii)**: when $f_t$ is monotone $(\gamma,\beta)$-weakly submodular, we can show that

$$\Big(\gamma^2 f_t(S) - (\beta(1-\gamma)+\gamma^2)F_t(\boldsymbol{\pi}_1,\ldots,\boldsymbol{\pi}_n)\Big) \leq \langle\mathbf{1}_S,\nabla F_t(\boldsymbol{\pi}_1,\ldots,\boldsymbol{\pi}_n)\rangle.$$

In order to prove Theorem 2, we next show the relationship between $\langle(\boldsymbol{\pi}_1,\ldots,\boldsymbol{\pi}_n),\nabla F_t(\boldsymbol{\pi}_1,\ldots,\boldsymbol{\pi}_n)\rangle$ and $F_t(\boldsymbol{\pi}_1,\ldots,\boldsymbol{\pi}_n)$.

From Eq.(13), if $\boldsymbol{\pi}_i = (\pi_{i,1},\ldots,\pi_{i,\kappa_i}), \forall i\in\mathcal{N}$, we can show

$$\langle(\boldsymbol{\pi}_1,\ldots,\boldsymbol{\pi}_n),\nabla F_t(\boldsymbol{\pi}_1,\ldots,\boldsymbol{\pi}_n)\rangle$$

$$= \sum_{i=1}^{n}\sum_{m=1}^{\kappa_i}\pi_{i,m}\frac{\partial F_t}{\partial\pi_{i,m}}(\boldsymbol{\pi}_1,\ldots,\boldsymbol{\pi}_n)$$

$$= \sum_{i=1}^{n}\sum_{m=1}^{\kappa_i}\pi_{i,m}\mathbb{E}_{a_j\sim\pi_j,\forall j\in\mathcal{N}}\Big(f_t\big(v_{i,m}\big|\cup_{j\neq i,j\in\mathcal{N}}\{a_j\}\big)\Big)$$

$$= \sum_{i=1}^{n}\sum_{m=1}^{\kappa_i}\left(\pi_{i,m}\sum_{a_j\in\mathcal{V}_j\cup\{\emptyset\},\forall j\neq i}\left(f_t\big(v_{i,m}\big|\cup_{j\neq i,j\in\mathcal{N}}\{a_j\}\big)\prod_{j\neq i,j\in\mathcal{N}}p(a_j|\boldsymbol{\pi}_j)\right)\right)$$

$$= \sum_{i=1}^{n}\sum_{m=1}^{\kappa_i}\left(p(v_{i,m}|\boldsymbol{\pi}_i)\sum_{a_j\in\mathcal{V}_j\cup\{\emptyset\},\forall j\neq i}\left(f_t\big(v_{i,m}\big|\cup_{j\neq i,j\in\mathcal{N}}\{a_j\}\big)\prod_{j\neq i,j\in\mathcal{N}}p(a_j|\boldsymbol{\pi}_j)\right)\right) \quad (20)$$

$$= \sum_{i=1}^{n}\sum_{m=1}^{\kappa_i}\left(\sum_{a_j\in\mathcal{V}_j\cup\{\emptyset\},\forall j\neq i}\left(f_t\big(v_{i,m}\big|\cup_{j\neq i,j\in\mathcal{N}}\{a_j\}\big)p(v_{i,m}|\boldsymbol{\pi}_i)*\prod_{j\neq i,j\in\mathcal{N}}p(a_j|\boldsymbol{\pi}_j)\right)\right)$$

$$= \sum_{i=1}^{n}\sum_{a_i\in\mathcal{V}_i\cup\{\emptyset\}}\left(\sum_{a_j\in\mathcal{V}_j\cup\{\emptyset\},\forall j\neq i}\left(f_t\big(a_i\big|\cup_{j\neq i,j\in\mathcal{N}}\{a_j\}\big)\prod_{j\in\mathcal{N}}p(a_j|\boldsymbol{\pi}_j)\right)\right)$$

$$= \sum_{i=1}^{n}\mathbb{E}_{a_j\sim\pi_j,\forall j\in\mathcal{N}}\Big(f_t\big(a_i\big|\cup_{j\neq i,j\in\mathcal{N}}\{a_j\}\big)\Big),$$

where the second equality follows from the **1)** of Theorem 1, the third equality comes from the final equality of the Eq.(13), the fourth equality follows from $p(v_{i,m}|\pi_i) = \pi_{i,m}$ and the final equality from the fact that the random element $a_i$ is drawn from the policy $\pi_i$.

Therefore, if the original set function $f_t$ is $\beta$-weakly submodular from above, we can show that

$$
\begin{aligned}
&\langle (\pi_1, \ldots, \pi_n), \nabla F_t(\pi_1, \ldots, \pi_n) \rangle \\
&= \sum_{i=1}^{n} \mathbb{E}_{a_j \sim \pi_j, \forall j \in \mathcal{N}} \Big( f_t\big( a_i \big| \cup_{j \neq i, j \in \mathcal{N}} \{a_j\} \big) \Big) \\
&= \mathbb{E}_{a_j \sim \pi_j, \forall j \in \mathcal{N}} \Big( \sum_{i=1}^{n} f_t\big( a_i \big| \cup_{j \neq i, j \in \mathcal{N}} \{a_j\} \big) \Big) \qquad (21) \\
&\leq \beta \mathbb{E}_{a_j \sim \pi_j, \forall j \in \mathcal{N}} \Big( f_t(\cup_{j \in \mathcal{N}} \{a_j\}) - f_t(\emptyset) \Big) \\
&= \beta F_t(\pi_1, \ldots, \pi_n).
\end{aligned}
$$

Therefore, by merging Eq.(21) and the **4)** of Theorem 1, when the set function $f_t$ is monotone and $(\gamma, \beta)$-weakly submodular, for any subset $S$ within the constraint of problem (1) and $(\pi_1, \ldots, \pi_n) \in \prod_{i=1}^{n} \Delta_{\kappa_i}$, we have that

$$
\langle \mathbf{1}_S - (\pi_1, \ldots, \pi_n), \nabla F_t(\pi_1, \ldots, \pi_n) \rangle \geq \gamma^2 f_t(S) - (\beta + \beta(1-\gamma) + \gamma^2) F_t(\pi_1, \ldots, \pi_n).
$$

Similarly, from the definition of $\alpha$-weakly DR-submodular function, we know that $\alpha$-weakly DR-submodular function automatically satisfies the conditions for being $\frac{1}{\alpha}$-weakly submodular from above, we also have that, when $f_t$ is monotone $\alpha$-weakly DR-submodular,

$$
\langle \mathbf{1}_S - (\pi_1, \ldots, \pi_n), \nabla F_t(\pi_1, \ldots, \pi_n) \rangle \geq \alpha f_t(S) - (\alpha + \frac{1}{\alpha}) F_t(\pi_1, \ldots, \pi_n).
$$

Therefore, when $(\pi_1^s, \ldots, \pi_n^s)$ is a stationary point of $F_t$ over the domain $\prod_{i=1}^{n} \Delta_{\kappa_i}$, we have

**i):** $f_t$ is monotone $(\gamma, \beta)$-weakly submodular, for any $S$ within the constraint of problem (1),

$$
\gamma^2 f_t(S) - (\beta + \beta(1-\gamma) + \gamma^2) F_t(\pi_1^s, \ldots, \pi_n^s) \leq \langle \mathbf{1}_S - (\pi_1^s, \ldots, \pi_n^s), \nabla F_t(\pi_1^s, \ldots, \pi_n^s) \rangle \leq 0.
$$

In other words,

$$
F_t(\pi_1^s, \ldots, \pi_n^s) \geq \frac{\gamma^2}{\beta + \beta(1-\gamma) + \gamma^2} f_t(S^*),
$$

where $S^*$ is the optimal subset of problem (1).

We get the **3)** in Theorem 2.

Similarly, **ii):** when $f_t$ is monotone $\alpha$-weakly DR-submodular, for any $S$ within the constraint of problem (1),

$$
\alpha f_t(S) - (\alpha + \frac{1}{\alpha}) F_t(\pi_1^s, \ldots, \pi_n^s) \leq \langle \mathbf{1}_S - (\pi_1^s, \ldots, \pi_n^s), \nabla F_t(\pi_1^s, \ldots, \pi_n^s) \rangle \leq 0.
$$

In other words,

$$
F_t(\pi_1^s, \ldots, \pi_n^s) \geq \frac{\alpha^2}{1 + \alpha} f_t(S^*).
$$

where $S^*$ is the optimal subset of problem (1).

We get the **2)** in Theorem 2.

### E.2 Proof of Part 1) in Theorem 2

In this subsection, we prove the part **1)** in Theorem 2.

From Eq.(17), we have

$$
\langle \mathbf{1}_S, \nabla F_t(\pi_1, \ldots, \pi_n) \rangle = \sum_{(i,m): v_{i,m} \in S} \mathbb{E}_{a_j \sim \pi_j, \forall j \in \mathcal{N}} \Big( f_t\big( v_{i,m} \big| \cup_{j \neq i, j \in \mathcal{N}} \{a_j\} \big) \Big).
$$

From Eq.(20), we have

$$\langle (\boldsymbol{\pi}_1, \ldots, \boldsymbol{\pi}_n), \nabla F_t(\boldsymbol{\pi}_1, \ldots, \boldsymbol{\pi}_n) \rangle = \sum_{i=1}^{n} \mathbb{E}_{a_j \sim \pi_j, \forall j \in \mathcal{N}} \Big( f_t \big( a_i \big| \cup_{j \neq i, j \in \mathcal{N}} \{a_j\} \big) \Big).$$

Therefore, we have that

$$\langle \mathbf{1}_S - (\boldsymbol{\pi}_1, \ldots, \boldsymbol{\pi}_n), \nabla F_t(\boldsymbol{\pi}_1, \ldots, \boldsymbol{\pi}_n) \rangle$$

$$= \sum_{(i,m): v_{i,m} \in S} \mathbb{E}_{a_j \sim \pi_j, \forall j \in \mathcal{N}} \Big( f_t \big( v_{i,m} \big| \cup_{j \neq i, j \in \mathcal{N}} \{a_j\} \big) \Big) - \sum_{i=1}^{n} \mathbb{E}_{a_j \sim \pi_j, \forall j \in \mathcal{N}} \Big( f_t \big( a_i \big| \cup_{j \neq i, j \in \mathcal{N}} \{a_j\} \big) \Big)$$

$$= \mathbb{E}_{a_j \sim \pi_j, \forall j \in \mathcal{N}} \Big( \sum_{(i,m): v_{i,m} \in S} f_t \big( v_{i,m} \big| \cup_{j \neq i, j \in \mathcal{N}} \{a_j\} \big) - \sum_{i=1}^{n} f_t \big( a_i \big| \cup_{j \neq i, j \in \mathcal{N}} \{a_j\} \big) \Big)$$

$$= \mathbb{E}_{a_j \sim \pi_j, \forall j \in \mathcal{N}} \Big( \sum_{v_{i,m} \in (S \setminus \cup_{j \in \mathcal{N}} \{a_j\})} f_t \big( v_{i,m} \big| \cup_{j \neq i, j \in \mathcal{N}} \{a_j\} \big) - \sum_{a_i \in (\cup_{j \in \mathcal{N}} \{a_j\} \setminus S)} f_t \big( a_i \big| \cup_{j \neq i, j \in \mathcal{N}} \{a_j\} \big) \Big).$$
(22)

When $f_t$ is monotone submodular, we have

$$\sum_{v_{i,m} \in (S \setminus \cup_{j \in \mathcal{N}} \{a_j\})} f_t \big( v_{i,m} \big| \cup_{j \neq i, j \in \mathcal{N}} \{a_j\} \big) \geq f_t \big( \cup_{j \in \mathcal{N}} \{a_j\} \big) \cup S \big) - f_t(\cup_{j \in \mathcal{N}} \{a_j\}). \quad (23)$$

Furthermore, from the definition of curvature $c$, we also have

$$f_t \big( \cup_{j \in \mathcal{N}} \{a_j\} \big) \cup S \big) - f_t(S) \geq (1 - c) \Big( f_t(\cup_{j \in \mathcal{N}} \{a_j\}) - f_t(\cup_{j \in \mathcal{N}} \{a_j\} \cap S) \Big), \quad (24)$$

where this inequality follows from $\big( \cup_{j \in \mathcal{N}} \{a_j\}) \cup S \big) \setminus S = \big( \cup_{j \in \mathcal{N}} \{a_j\} \big) \setminus \big( \cup_{j \in \mathcal{N}} \{a_j\} \cap S \big)$.
Similarly, from the submodularity, we have

$$\sum_{a_i \in (\cup_{j \in \mathcal{N}} \{a_j\} \setminus S)} f_t \big( a_i \big| \cup_{j \neq i, j \in \mathcal{N}} \{a_j\} \big) \leq \Big( f_t(\cup_{j \in \mathcal{N}} \{a_j\}) - f_t(\cup_{j \in \mathcal{N}} \{a_j\} \cap S) \Big), \quad (25)$$

where the inequality comes from that $\big( \cup_{j \in \mathcal{N}} \{a_j\} \cap S \big) \subseteq \big( \cup_{j \neq i, j \in \mathcal{N}} \{a_j\} \big)$ if $a_i \in \cup_{j \in \mathcal{N}} \{a_j\} \setminus S$.
Merging Eq.(23),Eq.(24) and Eq.(25) into Eq.(22), we can show that

$$\langle \mathbf{1}_S - (\boldsymbol{\pi}_1, \ldots, \boldsymbol{\pi}_n), \nabla F_t(\boldsymbol{\pi}_1, \ldots, \boldsymbol{\pi}_n) \rangle$$

$$= \mathbb{E}_{a_j \sim \pi_j, \forall j \in \mathcal{N}} \Big( \sum_{v_{i,m} \in S \setminus \cup_{j \in \mathcal{N}} \{a_j\}} f_t \big( v_{i,m} \big| \cup_{j \neq i, j \in \mathcal{N}} \{a_j\} \big) - \sum_{a_i \in \cup_{j \in \mathcal{N}} \{a_j\} \setminus S} f_t \big( a_i \big| \cup_{j \neq i, j \in \mathcal{N}} \{a_j\} \big) \Big)$$

$$\geq \mathbb{E}_{a_j \sim \pi_j, \forall j \in \mathcal{N}} \Big( f_t(S) - f_t(\cup_{j \in \mathcal{N}} \{a_j\}) - c \Big( f_t(\cup_{j \in \mathcal{N}} \{a_j\}) - f_t(\cup_{j \in \mathcal{N}} \{a_j\} \cap S) \Big) \Big)$$

$$\geq \mathbb{E}_{a_j \sim \pi_j, \forall j \in \mathcal{N}} \Big( f_t(S) - (1 + c) f_t(\cup_{j \in \mathcal{N}} \{a_j\}) \Big)$$

$$= f_t(S) - (1 + c) F_t(\boldsymbol{\pi}_1, \ldots, \boldsymbol{\pi}_n).$$

Thus, if $(\boldsymbol{\pi}_1^s, \ldots, \boldsymbol{\pi}_n^s)$ is a stationary point of $F_t$ over the domain $\prod_{i=1}^{n} \Delta_{\kappa_i}$, when $f_t$ is monotone submodular with curvature $c$, we have, for any $S$ in the constraint of problem (1), we have

$$f_t(S) - (1 + c) F_t(\boldsymbol{\pi}_1^s, \ldots, \boldsymbol{\pi}_n^s) \leq \langle \mathbf{1}_S - (\boldsymbol{\pi}_1^s, \ldots, \boldsymbol{\pi}_n^s), \nabla F_t(\boldsymbol{\pi}_1^s, \ldots, \boldsymbol{\pi}_n^s) \rangle \leq 0.$$

So,

$$F_t(\boldsymbol{\pi}_1^s, \ldots, \boldsymbol{\pi}_n^s) \geq \frac{1}{1 + c} f_t(S^*),$$

where $S^*$ is the optimal subset of problem (1).

### E.3 A Policy-based Continuous Extension with $1/2$-Approximation Stationary Point

In this subsection we consider a special set function.

At first, let the universe set $U$ consist of $n-1$ elements $\{x_1, \ldots, x_{n-1}\}$ and $n-k$ elements $\{y_1, \ldots, y_{n-k}\}$, all of weight 1, and $n-1$ elements $\{\epsilon_1, \ldots, \epsilon_{n-1}\}$ of arbitrarily small weight $\epsilon > 0$. Then, we define two different types of sets namely, $A_i$ and $A_{i+n}$ for any $i \in [n]$, that is to say ,

$$A_i \triangleq \{\epsilon_i\} \text{ for } 1 \leq i \leq n-1, \qquad A_n \triangleq \{x_1, \ldots, x_{n-1}\},$$
$$A_{n+i} \triangleq \{x_i\} \text{ for } 1 \leq i \leq n-1, \qquad A_{2n} \triangleq \{y_1, \ldots, y_{n-k}\}.$$

After that, we define a coverage set function $f : 2^{\mathcal{V}} \to \mathbb{R}_+$ over these $2n$ distinct set $\{A_1, \ldots, A_n, A_{n+1}, \ldots, A_{2n}\}$ where $\mathcal{V} = [2n]$. Specifically,for any subset $S \subseteq \mathcal{V}$,

$$f(S) \triangleq \sum_{v \in \bigcup_{i \in S} A_i} w(v), \tag{26}$$

where $w(v)$ is the weight of element $v$.

Moreover, we consider a partition constraint that contains at most one of $\{A_i, A_{n+i}\}$ for any $i \in [n]$. If we set $\mathcal{V}_i \triangleq \{i, i+n\}$ and $\mathcal{V} \triangleq \bigcup_{i \in [n]} \mathcal{V}_i \triangleq [2n]$, we naturally obtain the following coverage maximization problem:

$$\max_{S \subseteq \mathcal{V}} f(S) \text{ s.t. } |S \cap \mathcal{V}_i| \leq 1 \ \forall i \in [n]. \tag{27}$$

Note that this problem (27) is a special case of the concern problem (1). From the result of [38], we know that the coverage function $f$ in Equation (26) is a submodular set function, namely, $\alpha = \beta = \gamma = 1$.

A key feature of the coverage maximization problem (27) is that [38] found that the standard greedy [85] will be stuck at a local maximum subset $\{A_1, \ldots, A_n\}$ where $S = [n]$ and $f(S) = (1+\epsilon)n$. In contrast, when $\epsilon$ is very small, the optimal subset for the problem (27) is $\{A_{n+1}, \ldots, A_{2n}\}$ where $S = \{n+1, \ldots, 2n\}$ and $f(S) = 2n-k-1$. Note that $\lim_{n \to \infty} \lim_{\epsilon \to 0 \text{ and } k \to 0} \frac{(1+\epsilon)n}{2n-k-1} = \frac{1}{2}$. Motivated by this finding of [38], we also can show that the point $\mathbf{1}_{[n]} \triangleq (\underbrace{1, \ldots, 1}_{n}, 0, \ldots, 0)$ is a

local stationary point of the policy-based continuous extension of the set function $f$ in Equation (26). More specifically, we have the following theorem:

**Theorem 6.** *The point $\mathbf{1}_{[n]}$ is a stationary point of the policy-based continuous extension $F$ of the set function $f$ in Equation (26). Moreover, we can show $\frac{F(\mathbf{1}_{[n]})}{f(\{n+1,\ldots,2n\})} = \frac{(1+\epsilon)n}{2n-k-1} \to \frac{1}{2}$.*

**Remark 11.** *This theorem indicates that when the objective set function is submodular, namely, $c = \alpha = \gamma = \beta = 1$, the approximation guarantees established in Theorem 2 is **tight**.*

*Proof.* At first, for any $i \in [n]$, we assume $\boldsymbol{\pi}_i = (\pi_{i,1}, \pi_{i,2})$. Then, according to Definition 1, we have that the policy-based continuous extension $F$ of the set function $f$ in Equation (26) can be formulated as:

$$F(\boldsymbol{\pi}_1, \ldots, \boldsymbol{\pi}_n) \triangleq \sum_{i=1}^{n} \sum_{a_i \in \mathcal{V}_i \cup \{\emptyset\}} \left( f\left( \cup_{i=1}^n \{a_i\} \right) \prod_{i=1}^{n} p(a_i | \boldsymbol{\pi}_i) \right), \tag{28}$$

where $p(i|\boldsymbol{\pi}_i) = \pi_{i,1}$, $p(n+i|\boldsymbol{\pi}_i) = \pi_{i,2}$ and $p(\emptyset|\boldsymbol{\pi}_i) = 1 - \pi_{i,1} - \pi_{i,2}$.

From part **1)** of Theorem 1, we also can show that,

$$\frac{\partial F}{\partial \pi_{i,1}}(\mathbf{1}_{[n]}) = \epsilon \text{ for } 1 \leq i \leq n-1, \qquad \frac{\partial F}{\partial \pi_{n,1}}(\mathbf{1}_{[n]}) = n-1,$$
$$\frac{\partial F}{\partial \pi_{i,2}}(\mathbf{1}_{[n]}) = 0 \text{ for } 1 \leq i \leq n-1, \qquad \frac{\partial F}{\partial \pi_{n,2}}(\mathbf{1}_{[n]}) = n-k.$$

As a result, for any $(\pi_1, \ldots, \pi_n) \in \prod_{i=1}^n \Delta_2$, we have

$$\left\langle (\pi_1, \ldots, \pi_n) - \mathbf{1}_{[n]}, \nabla F(\mathbf{1}_{[n]}) \right\rangle$$

$$= \sum_{i=1}^n \left( \pi_{i,1} \frac{\partial F}{\partial \pi_{i,1}}(\mathbf{1}_{[n]}) + \pi_{i,2} \frac{\partial F}{\partial \pi_{i,2}}(\mathbf{1}_{[n]}) \right) - \sum_{i=1}^n \frac{\partial F}{\partial \pi_{i,1}}(\mathbf{1}_{[n]})$$

$$= \sum_{i=1}^n \left( \pi_{i,1} \frac{\partial F}{\partial \pi_{i,1}}(\mathbf{1}_{[n]}) + \pi_{i,2} \frac{\partial F}{\partial \pi_{i,2}}(\mathbf{1}_{[n]}) \right) - (1+\epsilon)(n-1)$$

$$= \epsilon \sum_{i=1}^{n-1} \pi_{i,1} + (n-1)\pi_{n,1} + (n-k)\pi_{n,2} - (1+\epsilon)(n-1)$$

$$= \epsilon \sum_{i=1}^{n-1} \pi_{i,1} + (n-1)(\pi_{n,1} + \pi_{n,2}) + \big((n-k) - (n-1)\big)\pi_{n,2} - (1+\epsilon)(n-1)$$

$$= \epsilon \sum_{i=1}^{n-1} (\pi_{i,1} - 1) + \big((n-k) - (n-1)\big)\pi_{i,2} - (n-1)(1 - \pi_{n,1} - \pi_{n,2})$$

$$= \epsilon \sum_{i=1}^{n-1} (\pi_{i,1} - 1) + (1-k)\pi_{i,2} - (n-1)(1 - \pi_{n,1} - \pi_{n,2}) \leq 0,$$

where the final inequality follows from $1 - k \leq 0$ and $1 - \pi_{n,1} - \pi_{n,2} \geq 0$. As a result, from Definition 2, we can know that the point $\mathbf{1}_{[n]}$ is a stationary point of the policy-based continuous extension $F$. Note that $\mathcal{F}(\mathbf{1}_{[n]}) = f([n]) = (1+\epsilon)(n-1)$ such that $\frac{F(\mathbf{1}_{[n]})}{f(\{n+1,\ldots,2n\})} = \frac{(1+\epsilon)n}{2n-k-1}$. Particularly when $k \to 0$, $\epsilon \to 0$ and $n \to \infty$, $\frac{F(\mathbf{1}_{[n]})}{f(\{n+1,\ldots,2n\})} = \frac{(1+\epsilon)n}{2n-k-1} \to \frac{1}{2}$. $\qquad \square$

# F  Proof of Theorem 3

In this section, we cut the proof of Theorem 3 in two distinct subsections. Specifically, Appendix F.1 provides the proof for parts **1)** and **2)**, while Appendix F.2 addresses the part **3)**.

## F.1  Proof of Part 1) and Part 2) in Theorem 3

Before verifying the parts **1)** and **2)**, we firstly prove the following theorem:

**Theorem 7.** *Given a monotone set function $f_t$, for its policy-based continuous extension $F_t$ introduced in Definition 1, if we consider a surrogate function $F_t^s : \prod_{i=1}^n \Delta_{\kappa_i} \to \mathbb{R}_+$ whose gradient at each point $\mathbf{x} \in \prod_{i=1}^n \Delta_{\kappa_i}$ is a weighted average of the gradient $\nabla F_t(z * \mathbf{x})$, namely, $\nabla F_t^s(\mathbf{x}) \triangleq \int_0^1 w(z) \nabla F_t(z * \boldsymbol{x}) \mathrm{d}z$ where $w(z)$ is a positive weight function over $[0,1]$, we have that:*

***i):*** *When $f_t$ is $\alpha$-weakly DR-submodular and $w(z) = e^{\alpha(z-1)}$, for any policy vector $(\pi_1, \ldots, \pi_n) \in \prod_{i=1}^n \Delta_{\kappa_i}$ and any subset $S$ within the constraint of problem (1), then the following inequality holds:*

$$\left\langle \mathbf{1}_S - (\pi_1, \ldots, \pi_n), \nabla F_t^s(\pi_1, \ldots, \pi_n) \right\rangle$$
$$= \left\langle \mathbf{1}_S - (\pi_1, \ldots, \pi_n), \int_0^1 e^{\alpha(z-1)} \nabla F_t(z * (\pi_1, \ldots, \pi_n)) \mathrm{d}z \right\rangle \qquad (29)$$
$$\geq \left(1 - e^{-\alpha}\right) f_t(S) - F_t(\pi_1, \ldots, \pi_n);$$

***ii):*** *When $f_t$ is $(\gamma, \beta)$-weakly submodular and $w(z) = e^{\phi(\gamma,\beta)(z-1)}$ where $\phi(\gamma, \beta) = \beta(1-\gamma) + \gamma^2$, for any policy vector $(\pi_1, \ldots, \pi_n) \in \prod_{i=1}^n \Delta_{\kappa_i}$ and any subset $S$ within the constraint of problem (1),*

*then the following inequality holds:*

$$\left\langle \mathbf{1}_S - (\pi_1, \ldots, \pi_n), \nabla F_t^s(\pi_1, \ldots, \pi_n) \right\rangle$$

$$= \left\langle \mathbf{1}_S - (\pi_1, \ldots, \pi_n), \int_0^1 e^{\phi(\gamma,\beta)(z-1)} \nabla F_t(z * (\pi_1, \ldots, \pi_n)) \, \mathrm{d}z \right\rangle \tag{30}$$

$$\geq \left( \frac{\gamma^2(1 - e^{-\phi(\gamma,\beta)})}{\phi(\gamma, \beta)} \right) f_t(S) - F_t(\pi_1, \ldots, \pi_n).$$

*Proof.* For **i**): At first, we verify the following relationship:

$$\left\langle (\pi_1, \ldots, \pi_n), \int_0^1 e^{\alpha(z-1)} \nabla F_t(z * (\pi_1, \ldots, \pi_n)) \, \mathrm{d}z \right\rangle$$

$$= \int_0^1 e^{\alpha(z-1)} \left\langle (\pi_1, \ldots, \pi_n), \nabla F_t(z * (\pi_1, \ldots, \pi_n)) \right\rangle \mathrm{d}z$$

$$= \int_0^1 e^{\alpha(z-1)} \frac{\mathrm{d}F_t(z * (\pi_1, \ldots, \pi_n))}{\mathrm{d}z} \mathrm{d}z$$

$$= \int_0^1 e^{\alpha(z-1)} \mathrm{d}F_t(z * (\pi_1, \ldots, \pi_n))$$

$$= e^{\alpha(z-1)} F_t(z * (\pi_1, \ldots, \pi_n)) \big|_{z=0}^{z=1} - \int_0^1 F_t(z * (\pi_1, \ldots, \pi_n)) \, \mathrm{d}\left(e^{\alpha(z-1)}\right)$$

$$= F_t(\pi_1, \ldots, \pi_n) - \alpha \int_0^1 e^{\alpha(z-1)} F_t(z * (\pi_1, \ldots, \pi_n)) \, \mathrm{d}z,$$

where the final equality follows from $F_t(\mathbf{0}) = 0$.

Then, we have the following inequality:

$$\left\langle \mathbf{1}_S, \int_0^1 e^{\alpha(z-1)} \nabla F_t(z * (\pi_1, \ldots, \pi_n)) \, \mathrm{d}z \right\rangle$$

$$= \int_0^1 e^{\alpha(z-1)} \left\langle \mathbf{1}_S, \nabla F_t(z * (\pi_1, \ldots, \pi_n)) \right\rangle \mathrm{d}z$$

$$\geq \alpha \int_0^1 e^{\alpha(z-1)} \left( f_t(S) - F_t(z * (\pi_1, \ldots, \pi_n)) \right) \mathrm{d}z$$

$$= \left( \alpha \int_0^1 e^{\alpha(z-1)} \mathrm{d}z \right) f_t(S) - \alpha \int_0^1 e^{\alpha(z-1)} F_t(z * (\pi_1, \ldots, \pi_n)) \, \mathrm{d}z$$

$$= \left( 1 - e^{-\alpha} \right) f_t(S) - \alpha \int_0^1 e^{\alpha(z-1)} F_t(z * (\pi_1, \ldots, \pi_n)) \, \mathrm{d}z,$$

where the first inequality follows from the part **4)** of Theorem 1.

As a result, we have

$$\left\langle \mathbf{1}_S - (\pi_1, \ldots, \pi_n), \int_0^1 e^{\alpha(z-1)} \nabla F_t(z * (\pi_1, \ldots, \pi_n)) \, \mathrm{d}z \right\rangle \geq \left( 1 - e^{-\alpha} \right) f_t(S) - F_t(\pi_1, \ldots, \pi_n).$$

As for **ii**): Similarly, when $f_t$ is monotone $(\gamma, \beta)$-weakly submodular and $w(z) = e^{\phi(\gamma,\beta)(z-1)}$ where $\phi(\gamma, \beta) = \beta(1 - \gamma) + \gamma^2$, we have

$$\left\langle (\boldsymbol{\pi}_1, \ldots, \boldsymbol{\pi}_n), \int_0^1 e^{\phi(\gamma,\beta)(z-1)} \nabla F_t (z * (\boldsymbol{\pi}_1, \ldots, \boldsymbol{\pi}_n)) \, \mathrm{d}z \right\rangle$$

$$= \int_0^1 e^{\phi(\gamma,\beta)(z-1)} \mathrm{d}F_t (z * (\boldsymbol{\pi}_1, \ldots, \boldsymbol{\pi}_n))$$

$$= e^{\phi(\gamma,\beta)(z-1)} F_t (z * (\boldsymbol{\pi}_1, \ldots, \boldsymbol{\pi}_n)) \mid_{z=0}^{z=1} - \phi(\gamma,\beta) \int_0^1 e^{\phi(\gamma,\beta)(z-1)} F_t (z * (\boldsymbol{\pi}_1, \ldots, \boldsymbol{\pi}_n)) \, \mathrm{d}z$$

$$= F_t(\boldsymbol{\pi}_1, \ldots, \boldsymbol{\pi}_n) - \phi(\gamma,\beta) \int_0^1 e^{\phi(\gamma,\beta)(z-1)} F_t (z * (\boldsymbol{\pi}_1, \ldots, \boldsymbol{\pi}_n)) \, \mathrm{d}z.$$

Then, we also have

$$\left\langle \mathbf{1}_S, \int_0^1 e^{\phi(\gamma,\beta)(z-1)} \nabla F_t (z * (\boldsymbol{\pi}_1, \ldots, \boldsymbol{\pi}_n)) \, \mathrm{d}z \right\rangle$$

$$= \int_0^1 e^{\phi(\gamma,\beta)(z-1)} \left\langle \mathbf{1}_S, \nabla F_t (z * (\boldsymbol{\pi}_1, \ldots, \boldsymbol{\pi}_n)) \right\rangle \mathrm{d}z$$

$$\geq \int_0^1 e^{\phi(\gamma,\beta)(z-1)} \left( \gamma^2 f_t(S) - \phi(\gamma,\beta) F_t \big( z * (\boldsymbol{\pi}_1, \ldots, \boldsymbol{\pi}_n) \big) \right) \mathrm{d}z$$

$$= \left( \gamma^2 \int_0^1 e^{\phi(\gamma,\beta)(z-1)} \mathrm{d}z \right) f_t(S) - \phi(\gamma,\beta) \int_0^1 e^{\phi(\gamma,\beta)(z-1)} F_t (z * (\boldsymbol{\pi}_1, \ldots, \boldsymbol{\pi}_n)) \, \mathrm{d}z$$

$$= \left( \frac{\gamma^2 (1 - e^{-\phi(\gamma,\beta)})}{\phi(\gamma,\beta)} \right) f_t(S) - \phi(\gamma,\beta) \int_0^1 e^{\phi(\gamma,\beta)(z-1)} F_t (z * (\boldsymbol{\pi}_1, \ldots, \boldsymbol{\pi}_n)) \, \mathrm{d}z,$$

where the first inequality follows from the part **4)** of Theorem 1.

As a result, we have

$$\left\langle \mathbf{1}_S - (\boldsymbol{\pi}_1, \ldots, \boldsymbol{\pi}_n), \int_0^1 e^{\phi(\gamma,\beta)(z-1)} \nabla F_t (z * (\boldsymbol{\pi}_1, \ldots, \boldsymbol{\pi}_n)) \, \mathrm{d}z \right\rangle \geq \left( \frac{\gamma^2 (1 - e^{-\phi(\gamma,\beta)})}{\phi(\gamma,\beta)} \right) f_t(S) - F_t(\boldsymbol{\pi}_1, \ldots, \boldsymbol{\pi}_n).$$

$\square$

From Definition 2, we can show that, when the policy vector $(\boldsymbol{\pi}_1^s, \ldots, \boldsymbol{\pi}_n^s)$ is the stationary point of the surrogate objective $F_t^s$ over domain $\prod_{i=1}^n \Delta_{\kappa_i}$ and $S$ satisfies the constraint of problem (1), due to $\mathbf{1}_S \in \prod_{i=1}^n \Delta_{\kappa_i}$, the following inequality holds:

$$\left\langle \mathbf{1}_S - (\boldsymbol{\pi}_1^s, \ldots, \boldsymbol{\pi}_n^s), \int_0^1 w(z) \nabla F_t (z * (\boldsymbol{\pi}_1^s, \ldots, \boldsymbol{\pi}_n^s)) \, \mathrm{d}z \right\rangle \leq 0,$$

where $w(z)$ is the corresponding weight function.

Then, following Theorem 7, we can show that when $f_t$ is monotone $\alpha$-weakly DR-submodular and $w(z) = e^{\alpha(z-1)}$, for any policy vector $(\boldsymbol{\pi}_1, \ldots, \boldsymbol{\pi}_n) \in \prod_{i=1}^n \Delta_{\kappa_i}$ and any subset $S$ within the constraint of problem (1),

$$F_t(\boldsymbol{\pi}_1^s, \ldots, \boldsymbol{\pi}_n^s) \geq (1 - e^{-\alpha}) f_t(S).$$

So we get the result of part **1)** of Theorem 3. Similarly, from **ii)** of Theorem 7, we also can achieve the part **2)** of Theorem 3.

### F.2 Proof of Part 3) in Theorem 3

In this subsection, we prove **3)** in Theorem 3.

At first, we define a modular function $g_t$ over $\mathcal{V}$, namely, for any subset $S \subseteq \mathcal{V}$, $g_t(S) \triangleq \sum_{a \in S} f_t(a | \mathcal{V} \setminus \{a\})$. Then, when $f_t$ is monotone submodular, we can show that the set function $(f_t - g_t)$ is also a monotone submodular function. That is,

**Theorem 8.** *For a monotone submodular function $f_t : \mathcal{V} \to \mathbb{R}_+$, we define a modular function $g_t(S) \triangleq \sum_{a \in S} f_t(a|\mathcal{V} \setminus \{a\}), \forall S \subseteq \mathcal{V}$, then we also can show that the set function $(f_t - g_t)$ is monotone submodular and non-negative.*

*Proof.* Because the set function $(f_t - g_t)$ is the sum of a submodular function $f_t$ and a modular function $-g_t$, $(f_t - g_t)$ must be submodular [105]. Then, we show the monotonicity of the set function $(f_t - g_t)$. At first, for any subset $S \subseteq \mathcal{V}$ and any action $a \in \mathcal{V} \setminus S$, we have that

$$(f_t - g_t)(a|S) = f_t(a|S) - g_t(a|S) = f_t(a|S) - f_t(a|\mathcal{V} \setminus \{a\}) \geq 0, \tag{31}$$

where the final inequality follows from the submodularity of $f_t$.

From Eq.(31), we can know that $(f_t - g_t)$ is monotone. Moreover, due to $f_t(\emptyset) = g_t(\emptyset) = 0$, $(f_t - g_t)$ is non-negative. $\qquad\square$

Next, we show the policy-based continuous extension of our aforementioned modular function $g_t$.

**Theorem 9.** *Given a monotone submodular function $f_t : \mathcal{V} \to \mathbb{R}_+$, we define its related modular function as $g_t(S) \triangleq \sum_{a \in S} f_t(a|\mathcal{V} \setminus \{a\}), \forall S \subseteq \mathcal{V}$. Then we can show that the policy-based continuous extension of $g_t$ is as follows: $G_t(\boldsymbol{\pi}_1, \ldots, \boldsymbol{\pi}_n) \triangleq \sum_{i=1}^n \sum_{m=1}^{\kappa_i} \big(f_t\big(v_{i,k}|\mathcal{V} - \{v_{i,m}\}\big)\big)\pi_{i,m}$ if $\boldsymbol{\pi}_i = (\pi_{i,1}, \ldots, \pi_{i,\kappa_i}) \in \Delta_{\kappa_i}, \forall i \in \mathcal{N}$.*

*Proof.* By the definition of $g_t$ and Definition 1, we can show that

$$G_t(\boldsymbol{\pi}_1, \ldots, \boldsymbol{\pi}_n) = \sum_{a_i \in \mathcal{V}_i \cup \{\emptyset\}, \forall i \in \mathcal{N}} \left( g_t\big(\cup_{i=1}^n \{a_i\}\big) \prod_{i=1}^n p(a_i|\boldsymbol{\pi}_i) \right)$$

$$= \sum_{a_i \in \mathcal{V}_i \cup \{\emptyset\}, \forall i \in \mathcal{N}} \left( \left( \sum_{i=1}^n f_t(a_i|\mathcal{V} \setminus \{a_i\}) \right) \prod_{i=1}^n p(a_i|\boldsymbol{\pi}_i) \right)$$

$$= \sum_{i=1}^n \sum_{a_i \in \mathcal{V}_i \cup \{\emptyset\}} f_t(a_i|\mathcal{V} \subseteq \{a_i\}) p(a_i|\boldsymbol{\pi}_i)$$

$$= \sum_{i=1}^n \sum_{m=1}^{\kappa_i} \big(f_t\big(v_{i,k}|\mathcal{V} - \{v_{i,m}\}\big)\big)\pi_{i,m},$$

where the final equality follows from $\mathcal{V}_i = \{v_{i,1}, \ldots, v_{i,\kappa_i}\}, \forall i \in \mathcal{N}$. $\qquad\square$

From Theorem 8, we know that, when $f_t$ is monotone submodular, so is $(f_t - g_t)$. Then, if we apply Theorem 7 to the monotone submodular function $(f_t - g_t)$, we can have the following relationship:

**Lemma 1.** *Given a monotone submodular set function $f_t$, we define a modular function $g_t(S) \triangleq \sum_{a \in S} f_t(a|\mathcal{V} \setminus \{a\}), \forall S \subseteq \mathcal{V}$. then, for any policy vector $(\boldsymbol{\pi}_1, \ldots, \boldsymbol{\pi}_n) \in \prod_{i=1}^n \Delta_{\kappa_i}$ and any subset $S$ within the constraint of problem (1), the following inequality holds:*

$$\left\langle \mathbf{1}_S - (\boldsymbol{\pi}_1, \ldots, \boldsymbol{\pi}_n), \int_0^1 e^{z-1} \nabla (F_t - G_t)(z * (\boldsymbol{\pi}_1, \ldots, \boldsymbol{\pi}_n)) \, \mathrm{d}z \right\rangle$$
$$\geq (1 - 1/e)(f_t - g_t)(S) - (F_t - G_t)(\boldsymbol{\pi}_1, \ldots, \boldsymbol{\pi}_n), \tag{32}$$

*where $F_t$ and $G_t$ is the corresponding policy-based continuous extension of $f_t$ and $g_t$, respectively.*

**Remark 12.** *Note that the submodular objective function is also 1-weakly DR-submodular fucntion. Thus, the results of Lemma 1 come from Theorem 7 and Theorem 8.*

Moreover, from the definition of $G_t$ in Theorem 9, we know that $G_t$ is a linear function over $\prod_{i=1} \Delta_{\kappa_i}$ such that

$$\left\langle \mathbf{1}_S - (\boldsymbol{\pi}_1, \ldots, \boldsymbol{\pi}_n), \nabla G_t(\boldsymbol{\pi}_1, \ldots, \boldsymbol{\pi}_n) \right\rangle = G_t(\mathbf{1}_S) - G_t(\boldsymbol{\pi}_1, \ldots, \boldsymbol{\pi}_n) = g_t(S) - G_t(\boldsymbol{\pi}_1, \ldots, \boldsymbol{\pi}_n). \tag{33}$$

Merging Eq.(32) and Eq.(33), we get the following theorem:

**Theorem 10.** *Given a monotone submodular set function $f_t$ with curvature $c$, for its policy-based continuous extension $F_t$ introduced in Definition 1, if we consider a surrogate function $F_t^s : \prod_{i=1}^n \Delta_{\kappa_i} \to \mathbb{R}_+$ whose gradient at each point $\mathbf{x} \in \prod_{i=1}^n \Delta_{\kappa_i}$ is a weighted average of the gradient $\nabla F_t(z * \mathbf{x})$, namely, $\nabla F_t^s(\mathbf{x}) \triangleq \int_0^1 e^{(z-1)} \nabla F_t(z * \boldsymbol{x}) \mathrm{d}z$, we have that, for any policy vector $(\boldsymbol{\pi}_1, \ldots, \boldsymbol{\pi}_n) \in \prod_{i=1}^n \Delta_{\kappa_i}$ and any subset $S$ within the constraint of problem (1), the following inequality holds:*

$$
\left\langle \mathbf{1}_S - (\boldsymbol{\pi}_1, \ldots, \boldsymbol{\pi}_n), \nabla \left( F_t^s + \frac{G_t}{e} \right) (\boldsymbol{\pi}_1, \ldots, \boldsymbol{\pi}_n) \right\rangle
$$
$$
= \left\langle \mathbf{1}_S - (\boldsymbol{\pi}_1, \ldots, \boldsymbol{\pi}_n), \frac{\nabla G_t(\boldsymbol{\pi}_1, \ldots, \boldsymbol{\pi}_n)}{e} + \int_0^1 e^{\alpha(z-1)} \nabla F_t (z * (\boldsymbol{\pi}_1, \ldots, \boldsymbol{\pi}_n)) \mathrm{d}z \right\rangle \quad (34)
$$
$$
\geq \left( 1 - \frac{c}{e} \right) f_t(S) - F_t(\boldsymbol{\pi}_1, \ldots, \boldsymbol{\pi}_n).
$$

*Proof.* According to Eq.(32) and Eq.(33), we have that

$$
\left\langle \mathbf{1}_S - (\boldsymbol{\pi}_1, \ldots, \boldsymbol{\pi}_n), \nabla G_t(\boldsymbol{\pi}_1, \ldots, \boldsymbol{\pi}_n) + \int_0^1 e^{z-1} \nabla (F_t - G_t) (z * (\boldsymbol{\pi}_1, \ldots, \boldsymbol{\pi}_n)) \mathrm{d}z \right\rangle
$$
$$
\geq (1 - 1/e) f_t(S) + \frac{g_t(S)}{e} - F_t((\boldsymbol{\pi}_1, \ldots, \boldsymbol{\pi}_n)) \quad (35)
$$
$$
\geq (1 - 1/e) f_t(S) + \frac{(1-c) f_t(S)}{e} - F_t((\boldsymbol{\pi}_1, \ldots, \boldsymbol{\pi}_n))
$$
$$
= (1 - c/e) f_t(S) - F_t((\boldsymbol{\pi}_1, \ldots, \boldsymbol{\pi}_n)),
$$

where the final inequality follows from the definition of curvature $c$ (See Lemma 2.1. in [105]).

Moreover, due to that $G_t$ is a linear function, we have

$$
\int_0^1 e^{z-1} \nabla (F_t - G_t) (z * (\boldsymbol{\pi}_1, \ldots, \boldsymbol{\pi}_n)) \mathrm{d}z
$$
$$
= \int_0^1 e^{z-1} \nabla F_t (z * (\boldsymbol{\pi}_1, \ldots, \boldsymbol{\pi}_n)) \mathrm{d}z - \int_0^1 e^{z-1} \mathrm{d}z \nabla G_t(\boldsymbol{\pi}_1, \ldots, \boldsymbol{\pi}_n) \quad (36)
$$
$$
= \int_0^1 e^{z-1} \nabla F_t (z * (\boldsymbol{\pi}_1, \ldots, \boldsymbol{\pi}_n)) \mathrm{d}z - (1 - 1/e) \nabla G_t(\boldsymbol{\pi}_1, \ldots, \boldsymbol{\pi}_n),
$$

where the first equality follows from $\nabla G_t (z * (\boldsymbol{\pi}_1, \ldots, \boldsymbol{\pi}_n)) = \nabla G_t(\boldsymbol{\pi}_1, \ldots, \boldsymbol{\pi}_n)$.

Merging Eq.(35) and Eq.(36), we get the Eq.(34). $\qquad \square$

From the result of Theorem 10, for any stationary point $(\boldsymbol{\pi}_1^s, \ldots, \boldsymbol{\pi}_n^s)$ of the objective $(F_t^s + \frac{G_t}{e})$ over $\prod_{i=1}^n \Delta_{\kappa_i}$ and any subset $S$ within the constraint of problem (1), then we have the following inequality:

$$
0 \geq \left\langle \mathbf{1}_S - (\boldsymbol{\pi}_1^s, \ldots, \boldsymbol{\pi}_n^s), \nabla \left( F_t^s + \frac{G_t}{e} \right) (\boldsymbol{\pi}_1^s, \ldots, \boldsymbol{\pi}_n^s) \right\rangle \geq \left( 1 - \frac{c}{e} \right) f_t(S) - F_t(\boldsymbol{\pi}_1^s, \ldots, \boldsymbol{\pi}_n^s).
$$

In other words, $F_t(\boldsymbol{\pi}_1^s, \ldots, \boldsymbol{\pi}_n^s) \geq \left( 1 - \frac{c}{e} \right) f_t(S^*)$ where $S^*$ is the optimal subset of problem (1).

# G  Convergence Analysis of `MA-SPL` Algorithm

In this section, we verify Theorem 4.

Before that, we firstly show that the surrogate gradient estimations in Line 16 of Algorithm 1, namely, $\mathbf{d}_i(t)$ is bounded and our proposed policy-based continuous extension $F_t$ is smooth.

**Lemma 2.** *If we set $M$ as the maximum marginal contribution of each monotone set function $f_t$, namely, $M \triangleq \max_{S \subseteq \mathcal{V}, e \in \mathcal{V} \setminus S, t \in [T]} (f_t(e|S))$, then we have $\|\mathbf{d}_i(t)\|_2 \leq \sqrt{\kappa_i} (\int_0^1 w(z) \mathrm{d}z + \frac{1}{e}) M$ where $\mathbf{d}_i(t)$ is the surrogate gradient estimations in Line 16 of Algorithm 1.*

*Proof.* From Lines 13-15, we know that $d_{i,m}(t) \triangleq (\int_{z=0}^1 w(z)\mathrm{d}z) f_t(v_{i,m}|S_i(t))$ or $d_{i,m}(t) \triangleq \left( (\int_{z=0}^1 w(z)\mathrm{d}z) f_t(v_{i,m}|S_i(t)) + e^{-1} f_t(v_{i,m}|\mathcal{V} - \{v_{i,m}\}) \right)$. Thus, we have $|d_{i,m}(t)| \leq (\int_0^1 w(z)\mathrm{d}z + \frac{1}{e})M$ for any $i \in \mathcal{N}$ and $m \in [\kappa_i]$ such that $\|\mathbf{d}_i(t)\|_2 = \sqrt{\sum_{m=1}^2 |d_{i,m}(t)|^2} \leq \sqrt{\kappa_i}(\int_0^1 w(z)\mathrm{d}z + \frac{1}{e})M$. $\qquad\square$

Before investigating the smoothness of our proposed policy-based continuous extension $F_t$, we firstly show its second-order partial derivative, i.e.,

**Lemma 3.** *For any vector* $(\pi_1, \ldots, \pi_n) \in \prod_{i=1}^n \Delta_{\kappa_i}$, *the second-order derivative of* $F_t$ *at variables* $\pi_{i_1,m_1}, \pi_{i_2,m_2}, \forall i_1 \in \mathcal{N}, \forall m_1 \in [\kappa_{i_1}], \forall i_2 \in \mathcal{N}, \forall m_2 \in [\kappa_{i_2}]$, *can be defined as:*

*i): When* $i_1 = i_2$, $\frac{\partial^2 F_t}{\partial \pi_{i_1,m_1} \partial \pi_{i_1,m_2}}(\pi_1, \ldots, \pi_n) = 0$;

*ii): When* $i_1 \neq i_2$,

$$
\frac{\partial^2 F_t}{\partial \pi_{i_1,m_1} \partial \pi_{i_2,m_2}}(\pi_{i,1}(t), \ldots, \pi_{i,n}(t))
$$
$$
= \mathbb{E}_{a_i \sim \pi_i, \forall i \in \mathcal{N}} \left( f_t\left(v_{i_1,m_1} \big| \left( \cup_{i \notin \{i_1,i_2\}} \{a_i\} \right) \cup \{v_{i_2,m_2}\} \right) - f_t\left(v_{i_1,m_1} \big| \cup_{i \notin \{i_1,i_2\}} \{a_i\} \right) \right).
$$

**Remark 13.** *From the Lemma 3, if we set* $M$ *as the maximum marginal contribution of each monotone set function* $f_t$, *namely,* $M \triangleq \max_{S \subseteq \mathcal{V}, e \in \mathcal{V} \backslash S, t \in [T]} (f_t(e|S))$, *then we can show that* $\left| \frac{\partial^2 F_t}{\partial \pi_{i_1,m_1} \partial \pi_{i_2,m_2}}(\pi_{i,1}(t), \ldots, \pi_{i,n}(t)) \right| \leq M$.

**Lemma 4.** *If we set* $M$ *as the maximum marginal contribution of each monotone set function* $f_t$, *namely,* $M \triangleq \max_{S \subseteq \mathcal{V}, e \in \mathcal{V} \backslash S, t \in [T]} (f_t(e|S))$, *then we can show that the Hessian matrix of the policy-based continuous extension* $F_t$ *of the previous monotone set function* $f_t$ *satisfies that,*

$$
\|\nabla^2 F_t(\pi_1, \ldots, \pi_n)\|_{2,\infty}^2 \leq M^2 \sum_{i=1}^n \kappa_i, \tag{37}
$$

*where* $(\pi_1, \ldots, \pi_n) \in \prod_{i=1}^n \Delta_{\kappa_i}$.

**Remark 14.** *For any matrix* $A \in \mathbb{R}^{n \times n}$, *the* $(2,\infty)$-*norm of* $A$ *is defined as* $\|A\|_{2,\infty} = \sup\{\|A\mathbf{x}\|_\infty : \mathbf{x} \in \mathbb{R}^n, \|\mathbf{x}\|_2 = 1\}$ *where* $\|\cdot\|_2$ *denotes the* $L_2$ *norm.*

*Proof.* From the definition of the norm $\|\cdot\|_{2,\infty}$, we can show that

$$
\|\nabla^2 F_t(\pi_1, \ldots, \pi_n)\|_{2,\infty}^2 = \max_{j \in \mathcal{N}} \|\nabla^2 F_t(\pi_1, \ldots, \pi_n)[j,:]\|_2^2 \leq M^2 \sum_{i=1}^n \kappa_i,
$$

where $\nabla^2 F_t(\pi_1, \ldots, \pi_n)[j,:]$ is the $j$-th line of the Hessian matrix $\nabla^2 F_t(\pi_1, \ldots, \pi_n)$ and the final inequality follows from Remark 13. $\qquad\square$

From Lemma 4, we can show that our proposed policy-based continuous extension $F_t$ is $(M \sum_{i=1}^n \kappa_i)$-smooth, namely,

**Lemma 5.** *If we set* $M$ *as the maximum marginal contribution of each monotone set function* $f_t$, *namely,* $M \triangleq \max_{S \subseteq \mathcal{V}, e \in \mathcal{V} \backslash S, t \in [T]} (f_t(e|S))$, *then we can show that our proposed policy-based continuous extension* $F_t$ *is* $(M \sum_{i=1}^n \kappa_i)$-*smooth.*

*Proof.* For any two point $(\boldsymbol{\pi}_1^a, \ldots, \boldsymbol{\pi}_n^a) \in \prod_{i=1}^n \Delta_{\kappa_i}$ and $(\boldsymbol{\pi}_1^b, \ldots, \boldsymbol{\pi}_n^b) \in \prod_{i=1}^n \Delta_{\kappa_i}$, if $\boldsymbol{\pi}_i^a \leq \boldsymbol{\pi}_i^b$ for any $i \in \mathcal{N}$, the following inequality holds:

$$\|\nabla F_t(\boldsymbol{\pi}_1^a, \ldots, \boldsymbol{\pi}_n^a) - \nabla F_t(\boldsymbol{\pi}_1^b, \ldots, \boldsymbol{\pi}_n^b)\|_2$$

$$= \|\int_0^1 \nabla^2 F_t(\boldsymbol{\pi}_1^a + \lambda \boldsymbol{\pi}_1^b, \ldots, \boldsymbol{\pi}_n^a + \lambda \boldsymbol{\pi}_n^b) \mathrm{d}\lambda (\boldsymbol{\pi}_1^a - \boldsymbol{\pi}_1^b, \ldots, \boldsymbol{\pi}_n^a - \boldsymbol{\pi}_n^b)\|_2$$

$$\leq \left(\sum_{i=1}^n \kappa_i\right)^{1/2} \|\int_0^1 \nabla^2 F_t(\boldsymbol{\pi}_1^a + \lambda \boldsymbol{\pi}_1^b, \ldots, \boldsymbol{\pi}_n^a + \lambda \boldsymbol{\pi}_n^b)(\boldsymbol{\pi}_1^a - \boldsymbol{\pi}_1^b, \ldots, \boldsymbol{\pi}_n^a - \boldsymbol{\pi}_n^b) \mathrm{d}\lambda\|_\infty$$

$$\leq M(\sum_{i=1}^n \kappa_i) \|(\boldsymbol{\pi}_1^a, \ldots, \boldsymbol{\pi}_n^a) - (\boldsymbol{\pi}_1^b, \ldots, \boldsymbol{\pi}_n^b)\|_2,$$

where the first inequality comes from $\|\mathbf{x}\|_2 \leq \sqrt{n}\|\mathbf{x}\|_\infty$ for any $n$-dimensional vector $\mathbf{x}$ and the final inequality follows from the Eq.(37). Thus, $F_t$ is $(M \sum_{i=1}^n \kappa_i)$-smooth. □

With Lemma 2 and Lemma 5, we next prove Theorem 4. At first, we show that each policy vector $(\boldsymbol{\pi}_{i,1}(t), \ldots, \boldsymbol{\pi}_{i,n}(t)), \forall i \in \mathcal{N}, \forall t \in [T]$ of Algorithm 1 is included in the constraint $\prod_{i=1}^n \Delta_{\kappa_i}$. That is, we have the following lemma:

**Lemma 6.** *In Algorithm 1, Under Assumption 1,* $(\boldsymbol{\pi}_{i,1}(t), \ldots, \boldsymbol{\pi}_{i,n}(t)) \in \prod_{i=1}^n \Delta_{\kappa_i}$.

*Proof.* We prove this theorem by induction. At first, from the Line 2 in Algorithm 1, we know that $\mathbf{x}_{i,j}(1) \in \Delta_{\kappa_j}$ for any $i, j \in \mathcal{N}$, so $(\boldsymbol{\pi}_{i,1}(1), \ldots, \boldsymbol{\pi}_{i,n}(1)) \in \prod_{i=1}^n \Delta_{\kappa_i}, \forall i \in \mathcal{N}$. Then, if for some $t \in [T]$, $(\boldsymbol{\pi}_{i,1}(t), \ldots, \boldsymbol{\pi}_{i,n}(t)) \in \prod_{i=1}^n \Delta_{\kappa_i}$. From Line 17, we know that $\boldsymbol{\pi}_{i,m}(t+1) := \sum_{j \in \mathcal{N}_i \cup \{i\}} w_{ij} \boldsymbol{\pi}_{j,m}(t)$ for any $m \neq i$. Then, due to Assumption 1, we have $\sum_{j \in \mathcal{N}_i \cup \{i\}} w_{ij} = 1$ and $w_{ij} \geq 0$ such that $\boldsymbol{\pi}_{i,m}(t+1) \in \Delta_{\kappa_m}, \forall m \neq i$. Moreover, from Line 19, we also know that $uppi_{i,i}(t+1) \in \Delta_{\kappa_i}$. Thus, $(\boldsymbol{\pi}_{i,1}(t+1), \ldots, \boldsymbol{\pi}_{i,n}(t+1)) \in \prod_{i=1}^n \Delta_{\kappa_i}$. □

Before going into the proof of Theorem 4, we define some commonly used symbols, namely,

$$\bar{\boldsymbol{\pi}}_{\cdot,j}(t) \triangleq \frac{\sum_{i=1}^n \boldsymbol{\pi}_{i,j}(t)}{n}, \quad \boldsymbol{\pi}_{\cdot,j}^{cate}(t) \triangleq [\boldsymbol{\pi}_{1,j}(t); \boldsymbol{\pi}_{2,j}(t); \ldots; \boldsymbol{\pi}_{n,j}(t)] \in \mathbb{R}^{n*\kappa_j}, \quad \forall j \in \mathcal{N};$$

$$\mathbf{z}_{i,m}(t) \triangleq \sum_{j \in \mathcal{N}_i \cup \{i\}} w_{ij} \boldsymbol{\pi}_{j,m}(t), \quad \forall i, m \in \mathcal{N};$$

$$\bar{\mathbf{z}}_{\cdot,j}(t) \triangleq \frac{\sum_{i=1}^n \mathbf{z}_{i,j}(t)}{n}, \quad \mathbf{z}_{\cdot,j}^{cate}(t) \triangleq [\mathbf{z}_{1,j}(t); \mathbf{z}_{2,j}(t); \ldots; \mathbf{z}_{n,j}(t)] \in \mathbb{R}^{n*\kappa_j}, \quad \forall j \in \mathcal{N};$$

$$\mathbf{r}_{i,j}(t) \triangleq \boldsymbol{\pi}_{i,j}(t+1) - \mathbf{z}_{i,j}(t), \quad \mathbf{r}_{\cdot,j}^{cate}(t) \triangleq [\mathbf{r}_{1,j}(t); \mathbf{r}_{2,j}(t); \ldots; \mathbf{r}_{n,j}(t)] \in \mathbb{R}^{n*\kappa_j}, \quad \forall j \in \mathcal{N}.$$

With these symbols, we can verify that

**Lemma 7.** $\mathbb{E}(\|\mathbf{r}_{i,j}(t)\|_2) = \mathbb{E}(\|\boldsymbol{\pi}_{i,j}(t+1) - \mathbf{z}_{i,j}(t)\|_2) \leq \sqrt{\kappa_i}(\int_0^1 w(z)\mathrm{d}z + \frac{1}{e})\eta_t M$ *where $M$ is the maximum marginal contribution of each monotone set function $f_t$.*

*Proof.* When $i \neq j$, from Line 17, we know that $\boldsymbol{\pi}_{i,j}(t+1) = \mathbf{z}_{i,j}(t)$ such that $\mathbf{r}_{i,j}(t) = \mathbf{0}_{\kappa_j}$ and $\mathbb{E}(\|\mathbf{r}_{i,j}(t)\|_2) = 0 \leq \sqrt{\kappa_i}(\int_0^1 w(z)\mathrm{d}z + \frac{1}{e})\eta_t M$. As for $i = j$, from Line 18 and Line 19, we have

$$\boldsymbol{\pi}_{i,i}(t+1) := \underset{\mathbf{b} \in \Delta_{\kappa_i}}{\arg\min} \|\mathbf{b} - (\mathbf{z}_{i,i}(t) + \eta_t \mathbf{d}_i(t))\|_2.$$

Note that $\mathbf{z}_{i,i}(t) \in \Delta_{\kappa_j}$. Thus, we have

$$\|\boldsymbol{\pi}_{i,i}(t+1) - \mathbf{z}_{i,i}(t)\|_2 \leq \|\mathbf{z}_{i,i}(t) - (\mathbf{z}_{i,i}(t) + \eta_t \mathbf{d}_i(t))\|_2 = \eta_t \|\mathbf{d}_i(t)\|_2 \leq \sqrt{\kappa_i}(\int_0^1 w(z)\mathrm{d}z + \frac{1}{e})\eta_t M,$$

$$(38)$$

where the final inequality follows from Lemma 2. □

**Lemma 8.** *Under Assumption 1, for any $t \in [T]$ and $i \in \mathcal{N}$, we have that*

$$\sum_{i \in \mathcal{N}} \mathbb{E}(\|\boldsymbol{\pi}_{i,j}(t+1) - \bar{\boldsymbol{\pi}}_{\cdot,j}(t+1)\|) \leq \sum_{m=1}^{t} \sqrt{n}\beta^{t-m}(\int_0^1 w(z)\mathrm{d}z + \frac{1}{e})\eta_m M,$$

$$\sum_{i \in \mathcal{N}} \mathbb{E}(\|\mathbf{z}_{i,j}(t+1) - \bar{\boldsymbol{\pi}}_{\cdot,j}(t+1)\|) \leq \sum_{m=1}^{t} \sqrt{n}\beta^{t-m}(\int_0^1 w(z)\mathrm{d}z + \frac{1}{e})\eta_m M,$$

*where $\tau \triangleq \max(|\lambda_2(\mathbf{W})|, |\lambda_n(\mathbf{W})|)$ is the second largest magnitude of the eigenvalues of the weight matrix $\mathbf{W}$.*

*Proof.* From the definition of $\mathbf{r}_{i,j}(t)$, we can show that

$$\mathbf{x}_{i,j}(t+1) = \mathbf{r}_{i,j}(t) + \mathbf{z}_{i,j}(t) = \mathbf{r}_{i,j}(t) + \sum_{k \in \mathcal{N}_i \cup \{i\}} w_{ik}\mathbf{x}_{k,j}. \tag{39}$$

As a result, from the Eq.(39), we also have that

$$\begin{aligned}
\boldsymbol{\pi}_{\cdot,j}^{cate}(t+1) &= \mathbf{r}_{\cdot,j}^{cate}(t) + (\mathbf{W} \otimes \mathbf{I}_{\kappa_j})\boldsymbol{\pi}_{\cdot,j}^{cate}(t) \\
&= \sum_{m=1}^{t} (\mathbf{W} \otimes \mathbf{I}_{\kappa_j})^{t-m}\mathbf{r}_{\cdot,j}^{cate}(m) \\
&= \sum_{m=1}^{t} (\mathbf{W}^{t-m} \otimes \mathbf{I}_{\kappa_j})\mathbf{r}_{\cdot,j}^{cate}(m),
\end{aligned} \tag{40}$$

where the symbol $\otimes$ denotes the Kronecker product.

If we also define $\bar{\boldsymbol{\pi}}_{\cdot,j}^{cate}(t) = [\bar{\boldsymbol{\pi}}_{\cdot,j}(t); \bar{\boldsymbol{\pi}}_{\cdot,j}(t); \ldots; \bar{\boldsymbol{\pi}}_{\cdot,j}(t)] \in \mathbb{R}^{n\kappa_j}$ and from the Eq.(40), we also have that

$$\begin{aligned}
\bar{\boldsymbol{\pi}}_{\cdot,j}^{cate}(t) &= \left(\frac{\mathbf{1}_n\mathbf{1}_n^T}{n} \otimes \mathbf{I}_{\kappa_j}\right)\boldsymbol{\pi}_{\cdot,j}^{cate}(t) \\
&= \sum_{m=1}^{t} (\frac{\mathbf{1}_n\mathbf{1}_n^T}{n} \otimes \mathbf{I}_{\kappa_j})\mathbf{r}_{\cdot,j}^{cate}(m).
\end{aligned} \tag{41}$$

Then, from the Eq.(40) and Eq.(41), we have that, for any $i \in \mathcal{N}$,

$$\boldsymbol{\pi}_{\cdot,j}^{cate}(t+1) - \bar{\boldsymbol{\pi}}_{\cdot,j}^{cate}(t) = \sum_{m=1}^{t} \sum_{j \in \mathcal{N}_i \cup \{i\}} \left([\mathbf{W}^{t-m}]_{ij} - \frac{1}{N}\right)\mathbf{r}_{\cdot,j}^{cate}(m). \tag{42}$$

Eq.(42) indicates that

$$\begin{aligned}
\mathbb{E}\left(\|\boldsymbol{\pi}_{\cdot,j}^{cate}(t+1) - \bar{\boldsymbol{\pi}}_{\cdot,j}^{cate}(t)\|\right) &= \mathbb{E}\left(\left\|\sum_{m=1}^{t} \sum_{j \in \mathcal{N}_i \cup \{i\}} \left([\mathbf{W}^{t-m}]_{ij} - \frac{1}{n}\right)\mathbf{r}_{\cdot,j}^{cate}(m)\right\|\right) \\
&\leq \mathbb{E}\left(\sum_{m=1}^{t} \sum_{j \in \mathcal{N}_i \cup \{i\}} \left|\left([\mathbf{W}^{t-m}]_{ij} - \frac{1}{n}\right)\right| * \|\mathbf{r}_{\cdot,j}^{cate}(m)\|_2\right) \\
&\leq \sum_{m=1}^{t} \sum_{j \in \mathcal{N}_i \cup \{i\}} \left|\left([\mathbf{W}^{t-m}]_{ij} - \frac{1}{n}\right)\right|\sqrt{\kappa_i}(\int_0^1 w(z)\mathrm{d}z + \frac{1}{e})\eta_m M \\
&\leq \sum_{m=1}^{t} \sqrt{n}\tau^{t-m}(\int_0^1 w(z)\mathrm{d}z + \frac{1}{e})\eta_m M,
\end{aligned}$$

where the second inequality comes from Lemma 7 and the final inequality follows from $\sum_{j \in \mathcal{N}_i \cup \{i\}} |[\mathbf{W}^{t-m}]_{ij} - \frac{1}{n}| \leq \sqrt{n}\tau^{t-m}$ (See Proposition 1 in [84]). Due to $\mathbf{z}_{i,j}(t+1) = \sum_{k \in \mathcal{N}_i \cup \{i\}} w_{ik}\boldsymbol{\pi}_{k,j}(t+1)$ we also can have $\mathbb{E}(\|z_{i,j}(t+1) - \bar{\boldsymbol{\pi}}_{\cdot,j}(t+1)\|_2) \leq \sum_{j \in \mathcal{N}_i \cup \{i\}} w_{ij}\mathbb{E}(\|\boldsymbol{\pi}_{i,j}(t+1) - \bar{\boldsymbol{\pi}}_{\cdot,j}(t+1)\|_2) \leq \sum_{m=1}^{t} \sqrt{n}\tau^{t-m}(\int_0^1 w(z)\mathrm{d}z + \frac{1}{e})\eta_m M.$ $\square$

**Lemma 9.** *When each objective function $f_t$ is monotone submodular with curvature $c$, $\alpha$-weakly DR-submodular or $(\gamma, \beta)$-weakly submodular, we set the ratio $\rho = \left(1 - \frac{c}{e}\right), (1 - e^{-\alpha}), \left(\frac{\gamma^2(1 - e^{-(\beta(1-\gamma)+\gamma^2)})}{\beta(1-\gamma)+\gamma^2}\right)$, respectively. Moreover, we use the symbol $M$ to denote the maximum marginal contribution of each monotone set function $f_t$, namely, $M \triangleq \max_{S \subseteq \mathcal{V}, e \in \mathcal{V} \setminus S, t \in [T]} (f_t(e|S))$. Then, under Assumption 1, if each set function $f_t$ is bounded $\forall t \in [T]$, the following inequality holds for Algorithm 1:*

$$\rho \sum_{t=1}^{T} f_t(\mathcal{A}_t^*) - \sum_{t=1}^{T} F_t(\bar{\boldsymbol{\pi}}_{\cdot,1}(t), \ldots, \bar{\boldsymbol{\pi}}_{\cdot,n}(t)) \leq 5M^2 n^{\frac{5}{2}} (\sum_{i=1}^{n} \kappa_i)(\int_0^1 w(z)\mathrm{d}z + \frac{1}{e})^2 \sum_{t=1}^{T} \sum_{m=1}^{t} \tau^{t-m} \eta_m$$

$$+ n\sqrt{\kappa_i}(\int_0^1 w(z)\mathrm{d}z + \frac{1}{e})M \sum_{t=1}^{T} \frac{\eta_t}{2} + \sum_{t=1}^{T} \frac{1}{2\eta_t} \sum_{i=1}^{n} \left(\|\mathbf{z}_{i,i}(t) - \mathbf{1}_{(\mathcal{A}_t^* \cap \mathcal{V}_i)}\|_2^2 - \|\boldsymbol{\pi}_{i,i}(t+1) - \mathbf{1}_{(\mathcal{A}_t^* \cap \mathcal{V}_i)}\|_2^2\right).$$

*where $\mathcal{A}_t^*$ is the optimal solution of problem (1) and $\tau \triangleq \max(|\lambda_2(\mathbf{W})|, |\lambda_n(\mathbf{W})|)$ is the second largest magnitude of the eigenvalues of the weight matrix $\mathbf{W}$.*

*Proof.* In order to unify the proofs in different setting, i.e., submodular objective with curvature $c$, $\alpha$-weakly DR-submodular objective and $(\gamma, \beta)$-weakly submodular objective, we define some auxiliary symbols. At first, when each objective function $f_t$ is monotone submodular with curvature $c$, $\alpha$-weakly DR-submodular or $(\gamma, \beta)$-weakly submodular, we set $\rho = \left(1 - \frac{c}{e}\right), (1 - e^{-\alpha}), \left(\frac{\gamma^2(1 - e^{-(\beta(1-\gamma)+\gamma^2)})}{\beta(1-\gamma)+\gamma^2}\right)$, respectively. Moreover, compared to weakly submodular scenarios, Theorem 3 designs a different auxiliary function for submodular scenarios. Therefore, when each $f_t$ is monotone submodular with curvature $c$ or weakly submodular, we define $F_t^a = F_t^s$ or $F_t^a = \left(F_t^s + \frac{G_t}{e}\right)$, respectively. Note that the $F_t^s$ is the surrogate function considered in Theorem 3 for different settings and $G_t$ is a linear function defined as $G_t(\boldsymbol{\pi}_1, \ldots, \boldsymbol{\pi}_n) \triangleq \sum_{i=1}^{n} \sum_{m=1}^{\kappa_i} \left(f_t(v_{i,k}|\mathcal{V} - \{v_{i,m}\})\right) \pi_{i,m}$.

From Theorem 7 and Theorem 10, we have that

$$\rho f_t(\mathcal{A}_t^*) - F_t(\bar{\boldsymbol{\pi}}_{\cdot,1}(t), \ldots, \bar{\boldsymbol{\pi}}_{\cdot,n}(t))$$

$$\leq \left\langle \nabla F_t^a(\bar{\boldsymbol{\pi}}_{\cdot,1}(t), \ldots, \bar{\boldsymbol{\pi}}_{\cdot,n}(t)), \mathbf{1}_{\mathcal{A}_t^*} - (\bar{\boldsymbol{\pi}}_{\cdot,1}(t), \ldots, \bar{\boldsymbol{\pi}}_{\cdot,n}(t)) \right\rangle$$

$$= \underbrace{\left\langle \nabla F_t^a(\bar{\boldsymbol{\pi}}_{\cdot,1}(t), \ldots, \bar{\boldsymbol{\pi}}_{\cdot,n}(t)) - \sum_{i \in \mathcal{N}} \left[\nabla F_t^a(\boldsymbol{\pi}_{i,1}(t), \ldots, \boldsymbol{\pi}_{i,1}(t))\right]_{\boldsymbol{\pi}_i}, \mathbf{1}_{\mathcal{A}_t^*} - (\bar{\boldsymbol{\pi}}_{\cdot,1}(t), \ldots, \bar{\boldsymbol{\pi}}_{\cdot,n}(t)) \right\rangle}_{①}$$

$$+ \underbrace{\left\langle \sum_{i \in \mathcal{N}} \left[\nabla F_t^a(\boldsymbol{\pi}_{i,1}(t), \ldots, \boldsymbol{\pi}_{i,1}(t))\right]_{\boldsymbol{\pi}_i}, \mathbf{1}_{\mathcal{A}_t^*} - (\boldsymbol{\pi}_{i,1}(t), \ldots, \boldsymbol{\pi}_{i,1}(t)) \right\rangle}_{②}$$

$$+ \underbrace{\left\langle \sum_{i \in \mathcal{N}} \left[\nabla F_t^a(\boldsymbol{\pi}_{i,1}(t), \ldots, \boldsymbol{\pi}_{i,1}(t))\right]_{\boldsymbol{\pi}_i}, (\boldsymbol{\pi}_{i,1}(t), \ldots, \boldsymbol{\pi}_{i,1}(t)) - (\bar{\boldsymbol{\pi}}_{\cdot,1}(t), \ldots, \bar{\boldsymbol{\pi}}_{\cdot,n}(t)) \right\rangle}_{③}$$

$$(43)$$

where the symbol $\left[\nabla F_t^a(\boldsymbol{\pi}_{i,1}(t), \ldots, \boldsymbol{\pi}_{i,1}(t))\right]_{\boldsymbol{\pi}_i}$ is the projection over the policy $\boldsymbol{\pi}_i$, i.e., $\left[\nabla F_t^a(\boldsymbol{\pi}_{i,1}(t), \ldots, \boldsymbol{\pi}_{i,1}(t))\right]_{\boldsymbol{\pi}_i}$ represents a $(\sum_{i=1}^{n} \kappa_i)$-dimensional vector that only keeps the first-order derivative at variable $\pi_{i,m}, \forall m \in [\kappa_i]$ and set other coordinates to 0, that is to say,

$$\left[\nabla F_t^a(\boldsymbol{\pi}_{i,1}(t), \ldots, \boldsymbol{\pi}_{i,1}(t))\right]_{\boldsymbol{\pi}_i} \triangleq \left(\ldots, 0, \ldots, \underbrace{\frac{\partial F_t}{\partial \pi_{i,1}}(\mathbf{x}), \ldots, \frac{\partial F_t}{\partial \pi_{i,\kappa_i}}(\mathbf{x})}_{\kappa_i}, \ldots, 0, \ldots\right),$$

where $\mathbf{x} \triangleq (\boldsymbol{\pi}_{i,1}(t), \ldots, \boldsymbol{\pi}_{i,1}(t))$.

For ①, we have

$$\left\langle \nabla F_t^a(\bar{\boldsymbol{\pi}}_{\cdot,1}(t), \ldots, \bar{\boldsymbol{\pi}}_{\cdot,n}(t)) - \sum_{i \in \mathcal{N}} \left[\nabla F_t^a(\boldsymbol{\pi}_{i,1}(t), \ldots, \boldsymbol{\pi}_{i,n}(t))\right]_{\pi_i}, \mathbf{1}_{\mathcal{A}_t^*} - (\bar{\boldsymbol{\pi}}_{\cdot,1}(t), \ldots, \bar{\boldsymbol{\pi}}_{\cdot,n}(t)) \right\rangle$$

$$\leq \left\| \nabla F_t^a(\bar{\boldsymbol{\pi}}_{\cdot,1}(t), \ldots, \bar{\boldsymbol{\pi}}_{\cdot,n}(t)) - \sum_{i \in \mathcal{N}} \left[\nabla F_t^a(\boldsymbol{\pi}_{i,1}(t), \ldots, \boldsymbol{\pi}_{i,n}(t))\right]_{\pi_i} \right\|_2 \left\| \mathbf{1}_{\mathcal{A}_t^*} - (\bar{\boldsymbol{\pi}}_{\cdot,1}(t), \ldots, \bar{\boldsymbol{\pi}}_{\cdot,n}(t)) \right\|_2$$

$$\leq \sum_{i \in \mathcal{N}} \left( \left\| \left[\nabla F_t^a(\bar{\boldsymbol{\pi}}_{\cdot,1}(t), \ldots, \bar{\boldsymbol{\pi}}_{\cdot,n}(t))\right]_{\pi_i} - \left[\nabla F_t^a(\boldsymbol{\pi}_{i,1}(t), \ldots, \boldsymbol{\pi}_{i,n}(t))\right]_{\pi_i} \right\|_2 \right) \left( \left\| \mathbf{1}_{\mathcal{A}_t^*} \right\|_2 + \left\| (\bar{\boldsymbol{\pi}}_{\cdot,1}(t), \ldots, \bar{\boldsymbol{\pi}}_{\cdot,n}(t)) \right\|_2 \right)$$

$$\leq 2n \sum_{i \in \mathcal{N}} \left( \left\| \nabla F_t^a(\bar{\boldsymbol{\pi}}_{\cdot,1}(t), \ldots, \bar{\boldsymbol{\pi}}_{\cdot,n}(t)) - \nabla F_t^a(\boldsymbol{\pi}_{i,1}(t), \ldots, \boldsymbol{\pi}_{i,n}(t)) \right\|_2 \right)$$

$$= 2n \sum_{i \in \mathcal{N}} \left( \left\| \int_{z=1}^1 w(z) \nabla F_t(z * \bar{\boldsymbol{\pi}}_{\cdot,1}(t), \ldots, z * \bar{\boldsymbol{\pi}}_{\cdot,n}(t)) \mathrm{d}z - \int_{z=1}^1 w(z) \nabla F_t(z * \boldsymbol{\pi}_{i,1}(t), \ldots, z * \boldsymbol{\pi}_{i,n}(t)) \mathrm{d}z \right\|_2 \right)$$

$$\leq 2Mn \left( \sum_{i=1}^n \kappa_i \right) \left( \int_0^1 w(z) z \mathrm{d}z \right) \| \sum_{i \in \mathcal{N}} \left( \left\| (\boldsymbol{\pi}_{i,1}(t), \ldots, \boldsymbol{\pi}_{i,n}(t)) - (\bar{\boldsymbol{\pi}}_{\cdot,1}(t), \ldots, \bar{\boldsymbol{\pi}}_{\cdot,n}(t)) \right\|_2 \right)$$

$$\leq 2Mn \left( \sum_{i=1}^n \kappa_i \right) \left( \int_0^1 w(z) z \mathrm{d}z \right) \sum_{j \in \mathcal{N}} \sum_{i \in \mathcal{N}} \| \boldsymbol{\pi}_{i,j}(t) - \bar{\boldsymbol{\pi}}_{\cdot,j}(t) \|_2$$

$$\leq 2M^2 n^{\frac{5}{2}} \left( \sum_{i=1}^n \kappa_i \right) \left( \int_0^1 w(z) z \mathrm{d}z \right) \left( \int_0^1 w(z) \mathrm{d}z + \frac{1}{e} \right) \sum_{m=1}^t \tau^{t-m} \eta_m,$$

(44)

where the fourth inequality follows Lemma 5 and the final inequality from Lemma 8.

For ③, under Lemma 2 and Lemma 8, we have,

$$\left\langle \sum_{i \in \mathcal{N}} \left[\nabla F_t^a(\boldsymbol{\pi}_{i,1}(t), \ldots, \boldsymbol{\pi}_{i,1}(t))\right]_{\pi_i}, (\boldsymbol{\pi}_{i,1}(t), \ldots, \boldsymbol{\pi}_{i,1}(t)) - (\bar{\boldsymbol{\pi}}_{\cdot,1}(t), \ldots, \bar{\boldsymbol{\pi}}_{\cdot,n}(t)) \right\rangle$$

$$\leq \sqrt{\kappa_i} \left( \int_0^1 w(z) \mathrm{d}z + \frac{1}{e} \right) M \sum_{i \in \mathcal{N}} \sum_{j \in \mathcal{N}} \| \boldsymbol{\pi}_{i,j}(t) - \bar{\boldsymbol{\pi}}_{\cdot,j}(t) \|_2$$

(45)

$$\leq \sqrt{\kappa_i} M^2 n^{3/2} \left( \int_0^1 w(z) \mathrm{d}z + \frac{1}{e} \right)^2 \sum_{m=1}^t \tau^{t-m} \eta_m.$$

As for ②, we have,

$$\mathbb{E}\left( \left\langle \sum_{i \in \mathcal{N}} \left[\nabla F_t^a(\boldsymbol{\pi}_{i,1}(t), \ldots, \boldsymbol{\pi}_{i,1}(t))\right]_{\pi_i}, \mathbf{1}_{\mathcal{A}_t^*} - (\boldsymbol{\pi}_{i,1}(t), \ldots, \boldsymbol{\pi}_{i,n}(t)) \right\rangle \right)$$

$$= \mathbb{E}\left( \mathbb{E}\left( \left\langle \sum_{i \in \mathcal{N}} \left[\nabla F_t^a(\boldsymbol{\pi}_{i,1}(t), \ldots, \boldsymbol{\pi}_{i,1}(t))\right]_{\pi_i}, \mathbf{1}_{\mathcal{A}_t^*} - (\boldsymbol{\pi}_{i,1}(t), \ldots, \boldsymbol{\pi}_{i,n}(t)) \right\rangle \Big| \boldsymbol{\pi}_{i,j}(t), \forall i, j \in \mathcal{N} \right) \right)$$

$$= \mathbb{E}\left( \left\langle \mathbb{E}\left( \sum_{i \in \mathcal{N}} \left[\nabla F_t^a(\boldsymbol{\pi}_{i,1}(t), \ldots, \boldsymbol{\pi}_{i,1}(t))\right]_{\pi_i} \Big| \boldsymbol{\pi}_{i,j}(t), \forall i, j \in \mathcal{N} \right), \mathbf{1}_{\mathcal{A}_t^*} - (\boldsymbol{\pi}_{i,1}(t), \ldots, \boldsymbol{\pi}_{i,n}(t)) \right\rangle \right)$$

$$= \mathbb{E}\left( \left\langle (\mathbf{d}_1(t), \ldots, \mathbf{d}_n(t)), \mathbf{1}_{\mathcal{A}_t^*} - (\boldsymbol{\pi}_{i,1}(t), \ldots, \boldsymbol{\pi}_{i,n}(t)) \right\rangle \right)$$

$$= \underbrace{\mathbb{E}\left( \left\langle (\mathbf{d}_1(t), \ldots, \mathbf{d}_n(t)), \mathbf{1}_{\mathcal{A}_t^*} - (\mathbf{z}_{i,1}(t), \ldots, \mathbf{z}_{i,1}(t)) \right\rangle \right)}_{④}$$

$$+ \underbrace{E\left( \left\langle (\mathbf{d}_1(t), \ldots, \mathbf{d}_n(t)), (\mathbf{z}_{i,1}(t), \ldots, \mathbf{z}_{i,n}(t)) - (\boldsymbol{\pi}_{i,1}(t), \ldots, \boldsymbol{\pi}_{i,n}(t)) \right\rangle \right)}_{⑤}.$$

(46)

For ⑤, we have that,

$$
⑤ \leq \mathbb{E}\Big( \|(\mathbf{d}_1(t), \ldots, \mathbf{d}_n(t))\|_2 \|(\mathbf{z}_{i,1}(t), \ldots, \mathbf{z}_{i,n}(t)) - (\boldsymbol{\pi}_{i,1}(t), \ldots, \boldsymbol{\pi}_{i,n}(t))\|_2 \Big)
$$

$$
\leq \sqrt{n\kappa_i} M(\int_0^1 w(z)\mathrm{d}z + \frac{1}{e})\mathbb{E}\Big( \|(\mathbf{z}_{i,1}(t), \ldots, \mathbf{z}_{i,n}(t)) - (\boldsymbol{\pi}_{i,1}(t), \ldots, \boldsymbol{\pi}_{i,n}(t))\|_2 \Big)
$$

$$
\leq \sqrt{n\kappa_i} M(\int_0^1 w(z)\mathrm{d}z + \frac{1}{e})\mathbb{E}\Big( \|(\mathbf{z}_{i,1}(t) - \bar{\boldsymbol{\pi}}_{\cdot,1}(t), \ldots, \mathbf{z}_{i,n}(t) - \bar{\boldsymbol{\pi}}_{\cdot,n}(t))\|_2 \Big)
$$

$$
+ \sqrt{n\kappa_i} M(\int_0^1 w(z)\mathrm{d}z + \frac{1}{e})\mathbb{E}\Big( \|(\boldsymbol{\pi}_{i,1}(t) - \bar{\boldsymbol{\pi}}_{\cdot,1}(t), \ldots, \boldsymbol{\pi}_{i,n}(t) - \bar{\boldsymbol{\pi}}_{\cdot,n}(t))\|_2 \Big) \tag{47}
$$

$$
\leq \sqrt{n\kappa_i} M(\int_0^1 w(z)\mathrm{d}z + \frac{1}{e}) \sum_{i \in \mathcal{N}} \sum_{j \in \mathcal{N}} \mathbb{E}\Big( \|\boldsymbol{\pi}_{i,j}(t) - \bar{\boldsymbol{\pi}}_{\cdot,j}(t)\|_2 + \|\mathbf{z}_{i,j}(t) - \bar{\boldsymbol{\pi}}_{\cdot,j}(t)\|_2 \Big)
$$

$$
\leq 2n^2 \sqrt{\kappa_i} M^2(\int_0^1 w(z)\mathrm{d}z + \frac{1}{e})^2 \sum_{m=1}^t \tau^{t-m}\eta_m
$$

For ④: At first, from the Line 19 of Algorithm 1, we know that, for any $\mathbf{x} \in \Delta_{\kappa_i}$, we have

$$
\|\boldsymbol{\pi}_{i,i}(t+1) - \mathbf{x}\|_2 \leq \|\mathbf{z}_{i,i}(t) + \eta_t \mathbf{d}_i(t) - \mathbf{x}\|_2.
$$

If we slightly abuse the notation $\mathbf{1}_{(\mathcal{A}_t^* \cap \mathcal{V}_i)}$ to denote $\kappa_i$-dimensional indicator vector over $(\mathcal{A}_t^* \cap \mathcal{V}_i)$. Note that $\mathbf{1}_{\mathcal{A}_t^*}$ is a $(\sum_{i=1}^n \kappa_i)$-dimensional indicator vector over $\mathcal{A}_t^*$. Then, we have

$$
\|\boldsymbol{\pi}_{i,i}(t+1) - \mathbf{1}_{(\mathcal{A}_t^* \cap \mathcal{V}_i)}\|_2^2
$$

$$
\leq \|\mathbf{z}_{i,i}(t) + \eta_t \mathbf{d}_i(t) - \mathbf{1}_{(\mathcal{A}_t^* \cap \mathcal{V}_i)}\|_2^2
$$

$$
= \|\mathbf{z}_{i,i}(t) - \mathbf{1}_{(\mathcal{A}_t^* \cap \mathcal{V}_i)}\|_2^2 + 2\eta_t \langle \mathbf{d}_i(t), \mathbf{z}_{i,i}(t) - \mathbf{1}_{(\mathcal{A}_t^* \cap \mathcal{V}_i)} \rangle + \eta_t^2 \|\mathbf{d}_i(t)\|_2^2.
$$

Therefore,

$$
\langle \mathbf{d}_i(t), \mathbf{1}_{(\mathcal{A}_t^* \cap \mathcal{V}_i)} - \mathbf{z}_{i,i}(t) \rangle
$$

$$
\leq \frac{\eta_t}{2} \|\mathbf{d}_i(t)\|_2^2 + \frac{1}{2\eta_t}\Big( \|\mathbf{z}_{i,i}(t) - \mathbf{1}_{(\mathcal{A}_t^* \cap \mathcal{V}_i)}\|_2^2 - \|\boldsymbol{\pi}_{i,i}(t+1) - \mathbf{1}_{(\mathcal{A}_t^* \cap \mathcal{V}_i)}\|_2^2 \Big)
$$

$$
\leq \frac{\eta_t}{2} \sqrt{\kappa_i}(\int_0^1 w(z)\mathrm{d}z + \frac{1}{e})M + \frac{1}{2\eta_t}\Big( \|\mathbf{z}_{i,i}(t) - \mathbf{1}_{(\mathcal{A}_t^* \cap \mathcal{V}_i)}\|_2^2 - \|\boldsymbol{\pi}_{i,i}(t+1) - \mathbf{1}_{(\mathcal{A}_t^* \cap \mathcal{V}_i)}\|_2^2 \Big),
$$

where the final inequality follows from Lemma 2.

Note that

$$
\mathbb{E}\Big( \big\langle (\mathbf{d}_1(t), \ldots, \mathbf{d}_n(t)), \mathbf{1}_{\mathcal{A}_t^*} - (\mathbf{z}_{i,1}(t), \ldots, \mathbf{z}_{i,1}(t)) \big\rangle \Big)
$$

$$
= \sum_{i=1}^n \mathbb{E}\left( \langle \mathbf{d}_i(t), \mathbf{1}_{(\mathcal{A}_t^* \cap \mathcal{V}_i)} - \mathbf{z}_{i,i}(t) \rangle \right)
$$

$$
\leq \frac{\eta_t}{2} n\sqrt{\kappa_i}(\int_0^1 w(z)\mathrm{d}z + \frac{1}{e})M + \frac{1}{2\eta_t} \sum_{i=1}^n \Big( \|\mathbf{z}_{i,i}(t) - \mathbf{1}_{(\mathcal{A}_t^* \cap \mathcal{V}_i)}\|_2^2 - \|\boldsymbol{\pi}_{i,i}(t+1) - \mathbf{1}_{(\mathcal{A}_t^* \cap \mathcal{V}_i)}\|_2^2 \Big).
$$

$$\tag{48}$$

Merging Eq.(48), Eq.(47), Eq.(46), Eq.(45), Eq.(44) into Eq.(43), we finally have

$$
\rho f_t(\mathcal{A}_t^*) - F_t(\bar{\boldsymbol{\pi}}_{\cdot,1}(t), \ldots, \bar{\boldsymbol{\pi}}_{\cdot,n}(t)) \leq 5M^2 n^{\frac{5}{2}}(\sum_{i=1}^n \kappa_i)(\int_0^1 w(z)\mathrm{d}z + \frac{1}{e})^2 \sum_{m=1}^t \beta^{t-m}\eta_m
$$

$$
+ \frac{\eta_t}{2} n\sqrt{\kappa_i}(\int_0^1 w(z)\mathrm{d}z + \frac{1}{e})M + \frac{1}{2\eta_t} \sum_{i=1}^n \Big( \|\mathbf{z}_{i,i}(t) - \mathbf{1}_{(\mathcal{A}_t^* \cap \mathcal{V}_i)}\|_2^2 - \|\boldsymbol{\pi}_{i,i}(t+1) - \mathbf{1}_{(\mathcal{A}_t^* \cap \mathcal{V}_i)}\|_2^2 \Big).
$$

Therefore, we get Lemma 9. $\qquad\qquad\square$

Next, we prove an upper bound of $\sum_{t=1}^T \frac{1}{2\eta_t} \sum_{i=1}^n \Big( \|\mathbf{z}_{i,i}(t) - \mathbf{1}_{(\mathcal{A}_t^* \cap \mathcal{V}_i)}\|_2^2 - \|\boldsymbol{\pi}_{i,i}(t+1) - \mathbf{1}_{(\mathcal{A}_t^* \cap \mathcal{V}_i)}\|_2^2 \Big)$, that is to say,

**Lemma 10.** *Under Assumption 1, we have that*

$$\sum_{t=1}^{T} \frac{1}{2\eta_t} \sum_{i=1}^{n} \left( \|\mathbf{z}_{i,i}(t) - \mathbf{1}_{(\mathcal{A}_t^* \cap \mathcal{V}_i)}\|_2^2 - \|\boldsymbol{\pi}_{i,i}(t+1) - \mathbf{1}_{(\mathcal{A}_t^* \cap \mathcal{V}_i)}\|_2^2 \right) \le \frac{4n}{\eta_{T+1}} + \sum_{t=1}^{T} \sum_{i=1}^{n} \frac{6}{\eta_{t+1}} \mathbb{E}\left( \|\mathbf{1}_{(\mathcal{A}_t^* \cap \mathcal{V}_i)} - \mathbf{1}_{(\mathcal{A}_{t+1}^* \cap \mathcal{V}_i)}\|_2 \right),$$

*where $\mathcal{A}_t^*$ is the optimal solution of problem* (1) *and we slightly abuse the notation $\mathbf{1}_{(\mathcal{A}_t^* \cap \mathcal{V}_i)}$ to denote $\kappa_i$-dimensional indicator vector over $(\mathcal{A}_t^* \cap \mathcal{V}_i)$. Note that $\mathbf{1}_{\mathcal{A}_t^*}$ is a $(\sum_{i=1}^{n} \kappa_i)$-dimensional indicator vector over $\mathcal{A}_t^*$.*

*Proof.*

$$\sum_{t=1}^{T} \frac{1}{2\eta_t} \sum_{i=1}^{n} \left( \|\mathbf{z}_{i,i}(t) - \mathbf{1}_{(\mathcal{A}_t^* \cap \mathcal{V}_i)}\|_2^2 - \|\boldsymbol{\pi}_{i,i}(t+1) - \mathbf{1}_{(\mathcal{A}_t^* \cap \mathcal{V}_i)}\|_2^2 \right)$$

$$= \underbrace{\sum_{t=1}^{T} \sum_{i=1}^{n} \left( \frac{1}{\eta_t} \mathbb{E}(\|\mathbf{z}_{i,i}(t) - \mathbf{1}_{(\mathcal{A}_t^* \cap \mathcal{V}_i)}\|_2^2) - \frac{1}{\eta_{t+1}} \mathbb{E}(\|\mathbf{z}_{i,i}(t+1) - \mathbf{1}_{(\mathcal{A}_{t+1}^* \cap \mathcal{V}_i)}\|_2^2) \right)}_{①}$$

$$+ \underbrace{\sum_{t=1}^{T} \sum_{i=1}^{n} \left( \frac{1}{\eta_{t+1}} \mathbb{E}\left( \|\mathbf{z}_{i,i}(t+1) - \mathbf{1}_{(\mathcal{A}_{t+1}^* \cap \mathcal{V}_i)}\|_2^2 - \|\mathbf{z}_{i,i}(t+1) - \mathbf{1}_{(\mathcal{A}_t^* \cap \mathcal{V}_i)}\|_2^2 \right) \right)}_{②}$$

$$+ \underbrace{\sum_{t=1}^{T} \sum_{i=1}^{n} \left( \frac{1}{\eta_{t+1}} \mathbb{E}\left( \|\mathbf{z}_{i,i}(t+1) - \mathbf{1}_{(\mathcal{A}_t^* \cap \mathcal{V}_i)}\|_2^2 - \|\boldsymbol{\pi}_{i,i}(t+1) - \mathbf{1}_{(\mathcal{A}_t^* \cap \mathcal{V}_i)}\|_2^2 \right) \right)}_{③}$$

$$+ \underbrace{\sum_{t=1}^{T} \sum_{i=1}^{n} \left( \frac{1}{\eta_{t+1}} - \frac{1}{\eta_t} \right) \mathbb{E}\left( \|\boldsymbol{\pi}_{i,i}(t+1) - \mathbf{1}_{(\mathcal{A}_t^* \cap \mathcal{V}_i)}\|_2^2 \right)}_{④}.$$

Firstly, we have $① \le \frac{\sum_{i=1}^{n} \mathbb{E}(\|\mathbf{z}_{i,i}(1) - \mathbf{1}_{(\mathcal{A}_1^* \cap \mathcal{V}_i)}\|_2^2)}{\eta_1} \le \frac{4n}{\eta_1}$.

Moreover, we have $④ \le 4n\left( \frac{1}{\eta_{T+1}} - \frac{1}{\eta_1} \right)$.

As for $②$, we have

$$\sum_{t=1}^{T} \sum_{i=1}^{n} \left( \frac{1}{\eta_{t+1}} \mathbb{E}\left( \|\mathbf{z}_{i,i}(t+1) - \mathbf{1}_{(\mathcal{A}_{t+1}^* \cap \mathcal{V}_i)}\|_2^2 - \|\mathbf{z}_{i,i}(t+1) - \mathbf{1}_{(\mathcal{A}_t^* \cap \mathcal{V}_i)}\|_2^2 \right) \right)$$

$$= \sum_{t=1}^{T} \sum_{i=1}^{n} \left( \frac{1}{\eta_{t+1}} \mathbb{E}\left( \|\mathbf{1}_{(\mathcal{A}_t^* \cap \mathcal{V}_i)} - \mathbf{1}_{(\mathcal{A}_{t+1}^* \cap \mathcal{V}_i)}\|_2^2 + 2 \left\langle \mathbf{z}_{i,i}(t+1) - \mathbf{1}_{(\mathcal{A}_t^* \cap \mathcal{V}_i)}, \mathbf{1}_{(\mathcal{A}_t^* \cap \mathcal{V}_i)} - \mathbf{1}_{(\mathcal{A}_{t+1}^* \cap \mathcal{V}_i)} \right\rangle \right) \right).$$

$$\le \sum_{t=1}^{T} \sum_{i=1}^{n} \frac{6}{\eta_{t+1}} \mathbb{E}\left( \|\mathbf{1}_{(\mathcal{A}_t^* \cap \mathcal{V}_i)} - \mathbf{1}_{(\mathcal{A}_{t+1}^* \cap \mathcal{V}_i)}\|_2 \right),$$

where the final inequality from $\left\langle \mathbf{z}_{i,i}(t+1) - \mathbf{1}_{(\mathcal{A}_t^* \cap \mathcal{V}_i)}, \mathbf{1}_{(\mathcal{A}_t^* \cap \mathcal{V}_i)} - \mathbf{1}_{(\mathcal{A}_{t+1}^* \cap \mathcal{V}_i)} \right\rangle \le \|\mathbf{z}_{i,i}(t+1) - \mathbf{1}_{(\mathcal{A}_t^* \cap \mathcal{V}_i)}\|_2 * \|\mathbf{1}_{(\mathcal{A}_t^* \cap \mathcal{V}_i)} - \mathbf{1}_{(\mathcal{A}_{t+1}^* \cap \mathcal{V}_i)}\|_2 \le 2\|\mathbf{1}_{(\mathcal{A}_t^* \cap \mathcal{V}_i)} - \mathbf{1}_{(\mathcal{A}_{t+1}^* \cap \mathcal{V}_i)}\|_2$.

Then, we have

$$\text{③} = \sum_{t=1}^{T}\sum_{i=1}^{n}\left(\frac{1}{\eta_{t+1}}\mathbb{E}\Big(\|\mathbf{z}_{i,i}(t+1)-\mathbf{1}_{(\mathcal{A}_t^*\cap\mathcal{V}_i)}\|_2^2-\|\boldsymbol{\pi}_{i,i}(t+1)-\mathbf{1}_{(\mathcal{A}_t^*\cap\mathcal{V}_i)}\|_2^2\Big)\right)$$

$$= \sum_{t=1}^{T}\sum_{i=1}^{n}\left(\frac{1}{\eta_{t+1}}\mathbb{E}\Big(\|\sum_{j\in\mathcal{N}_i\cup\{i\}}w_{ij}\boldsymbol{\pi}_{j,i}(t+1)-\mathbf{1}_{(\mathcal{A}_t^*\cap\mathcal{V}_i)}\|_2^2-\|\boldsymbol{\pi}_{i,i}(t+1)-\mathbf{1}_{(\mathcal{A}_t^*\cap\mathcal{V}_i)}\|_2^2\Big)\right)$$

$$\leq \sum_{t=1}^{T}\left(\frac{1}{\eta_{t+1}}\mathbb{E}\Big(\sum_{i\in\mathcal{N}}\sum_{j\in\mathcal{N}_i\cup\{i\}}\Big(w_{ij}\|\boldsymbol{\pi}_{j,i}(t+1)-\mathbf{1}_{(\mathcal{A}_t^*\cap\mathcal{V}_i)}\|_2^2\Big)-\sum_{i\in\mathcal{N}}\|\boldsymbol{\pi}_{i,i}(t+1)-\mathbf{1}_{(\mathcal{A}_t^*\cap\mathcal{V}_i)}\|_2^2\Big)\right)$$

$$= \sum_{t=1}^{T}\left(\frac{1}{\eta_{t+1}}\mathbb{E}\Big(\sum_{i\in\mathcal{N}}\Big((\sum_{j\in\mathcal{N}_i\cup\{i\}}w_{ji})\|\boldsymbol{\pi}_{j,i}(t+1)-\mathbf{1}_{(\mathcal{A}_t^*\cap\mathcal{V}_i)}\|_2^2\Big)-\sum_{i\in\mathcal{N}}\|\boldsymbol{\pi}_{i,i}(t+1)-\mathbf{1}_{(\mathcal{A}_t^*\cap\mathcal{V}_i)}\|_2^2\Big)\right)$$

$$= 0,$$

where the first inequality follows from the convexity of $\|\cdot\|_2^2$, the third equality is from $w_{ij} = w_{ji}$ and the final equality follows from Assumption 1.

We finally get

$$\sum_{t=1}^{T}\frac{1}{2\eta_t}\sum_{i=1}^{n}\left(\|\mathbf{z}_{i,i}(t)-\mathbf{1}_{(\mathcal{A}_t^*\cap\mathcal{V}_i)}\|_2^2-\|\boldsymbol{\pi}_{i,i}(t+1)-\mathbf{1}_{(\mathcal{A}_t^*\cap\mathcal{V}_i)}\|_2^2\right)\leq\frac{4n}{\eta_{T+1}}+\sum_{t=1}^{T}\sum_{i=1}^{n}\frac{6}{\eta_{t+1}}\mathbb{E}\Big(\|\mathbf{1}_{(\mathcal{A}_t^*\cap\mathcal{V}_i)}-\mathbf{1}_{(\mathcal{A}_{t+1}^*\cap\mathcal{V}_i)}\|_2\Big).$$

$\square$

As a result, we can show that

**Lemma 11.** *Under Assumption 1, when each set objective function $f_t$ is monotone submodular with curvature $c$, $\alpha$-weakly DR-submodular or $(\gamma,\beta)$-weakly submodular, if we set the weight function $w(z)$ according to Theorem 3, we have that*

$$\rho\sum_{t=1}^{T}f_t(\mathcal{A}_t^*)-\sum_{t=1}^{T}\mathbb{E}(f_t(\cup_{i=1}^{n}\{a_i(t)\}))\leq 6M^2n^{\frac{5}{2}}(\sum_{i=1}^{n}\kappa_i)(\int_0^1 w(z)\mathrm{d}z+\frac{1}{e})^2\sum_{t=1}^{T}\sum_{m=1}^{t}\beta^{t-m}\eta_m$$

$$+n\sqrt{\kappa_i}(\int_0^1 w(z)\mathrm{d}z+\frac{1}{e})M\sum_{t=1}^{T}\frac{\eta_t}{2}+\frac{4n}{\eta_{T+1}}+\sum_{t=1}^{T}\sum_{i=1}^{n}\frac{6}{\eta_{t+1}}\mathbb{E}\Big(\|\mathbf{1}_{(\mathcal{A}_t^*\cap\mathcal{V}_i)}-\mathbf{1}_{(\mathcal{A}_{t+1}^*\cap\mathcal{V}_i)}\|_2\Big).$$

*Proof.* Merging Lemma 10 into Lemma 9, we can get that

$$\rho\sum_{t=1}^{T}f_t(\mathcal{A}_t^*)-\sum_{t=1}^{T}F_t(\bar{\boldsymbol{\pi}}_{\cdot,1}(t),\ldots,\bar{\boldsymbol{\pi}}_{\cdot,n}(t))\leq 5M^2n^{\frac{5}{2}}(\sum_{i=1}^{n}\kappa_i)(\int_0^1 w(z)\mathrm{d}z+\frac{1}{e})^2\sum_{t=1}^{T}\sum_{m=1}^{t}\tau^{t-m}\eta_m$$

$$+n\sqrt{\kappa_i}(\int_0^1 w(z)\mathrm{d}z+\frac{1}{e})M\sum_{t=1}^{T}\frac{\eta_t}{2}+\frac{4n}{\eta_{T+1}}+\sum_{t=1}^{T}\sum_{i=1}^{n}\frac{6}{\eta_{t+1}}\mathbb{E}\Big(\|\mathbf{1}_{(\mathcal{A}_t^*\cap\mathcal{V}_i)}-\mathbf{1}_{(\mathcal{A}_{t+1}^*\cap\mathcal{V}_i)}\|_2\Big).$$

From Theorem 1, we also can show that $|F_t(\mathbf{x})-F_t(\mathbf{y})|\leq nM\|\mathbf{x}-\mathbf{y}\|_2$ for any $t\in[T]$ such that we have

$$|F_t(\bar{\boldsymbol{\pi}}_{\cdot,1}(t),\ldots,\bar{\boldsymbol{\pi}}_{\cdot,n}(t))-F_t(\boldsymbol{\pi}_{1,1}(t),\boldsymbol{\pi}_{2,2}(t),\ldots,\boldsymbol{\pi}_{n,n}(t))|$$

$$\leq nM\|(\bar{\boldsymbol{\pi}}_{\cdot,1}(t),\ldots,\bar{\boldsymbol{\pi}}_{\cdot,n}(t))-(\boldsymbol{\pi}_{1,1}(t),\ldots,\boldsymbol{\pi}_{n,n}(t))\|_2$$

$$= nM\sum_{i=1}^{n}\|\bar{\boldsymbol{\pi}}_{\cdot,i}(t)-\boldsymbol{\pi}_{i,i}(t)\|_2\leq nM\sum_{j=1}^{n}\sum_{i=1}^{n}\|\bar{\boldsymbol{\pi}}_{\cdot,i}(t)-\boldsymbol{\pi}_{j,i}(t)\|_2\leq n^{5/2}M^2(\int_0^1 w(z)\mathrm{d}z+\frac{1}{e})\sum_{m=1}^{t}\tau^{t-m}\eta_m,$$

where the final inequality follows from Lemma 8.

Moreover, from the monotonicity of $F_t$(See **2)** in Theorem 1) and Lines 5-8, we also can infer that

$$\mathbb{E}(f_t(\cup_{i=1}^{n}\{a_i(t)\}))=F_t(\mathbf{p}_1(t),\ldots,\mathbf{p}_n(t))\geq F_t(\boldsymbol{\pi}_{1,1}(t),\boldsymbol{\pi}_{2,2}(t),\ldots,\boldsymbol{\pi}_{n,n}(t)),$$

where the final inequality comes from $\mathbf{p}_i(t) := \frac{\pi_{i,i}(t)}{\|\pi_{i,i}(t)\|_1} \geq \pi_{i,i}(t)$. Therefore, we have that

$$\rho \sum_{t=1}^{T} f_t(\mathcal{A}_t^*) - \sum_{t=1}^{T} \mathbb{E}(f_t(\cup_{i=1}^n \{a_i(t)\})) \leq 6M^2 n^{\frac{5}{2}} (\sum_{i=1}^{n} \kappa_i)(\int_0^1 w(z)\mathrm{d}z + \frac{1}{e})^2 \sum_{t=1}^{T} \sum_{m=1}^{t} \tau^{t-m} \eta_m$$

$$+ n\sqrt{\kappa_i}(\int_0^1 w(z)\mathrm{d}z + \frac{1}{e})M \sum_{t=1}^{T} \frac{\eta_t}{2} + \frac{4n}{\eta_{T+1}} + \sum_{t=1}^{T} \sum_{i=1}^{n} \frac{6}{\eta_{t+1}} \mathbb{E}\Big(\|\mathbf{1}_{(\mathcal{A}_t^* \cap \mathcal{V}_i)} - \mathbf{1}_{(\mathcal{A}_{t+1}^* \cap \mathcal{V}_i)}\|_2\Big).$$

$\square$

From Lemma 11, if we set the step size $\eta_t = \mathcal{O}\big(\sqrt{\frac{(1-\tau)\mathcal{P}_T}{T}}\big)$ where $\mathcal{P}_T \triangleq \sum_{t=1}^{T} |\mathcal{A}_{t+1}^* \triangle \mathcal{A}_t^*|$ and $\triangle$ denotes the symmetric difference, we can show that

$$\rho \sum_{t=1}^{T} F_t(\mathbf{1}_{\mathcal{A}_t^*}) - \sum_{t=1}^{T} \mathbb{E}(f_t(\cup_{i=1}^n \{a_i(t)\}))$$

$$\leq \mathcal{O}(\sum_{t=1}^{T} \sum_{m=1}^{t} \tau^{t-m} \eta_m) + \mathcal{O}(\sum_{t=1}^{T} \eta_t) + \mathcal{O}(\frac{1}{\eta_{T+1}}) + \mathcal{O}\Big(\sum_{t=1}^{T} \sum_{i=1}^{n} \frac{1}{\eta_{t+1}} \mathbb{E}(\|\mathbf{1}_{(\mathcal{A}_t^* \cap \mathcal{V}_i)} - \mathbf{1}_{(\mathcal{A}_{t+1}^* \cap \mathcal{V}_i)}\|_2)\Big)$$

$$= \mathcal{O}\left(\sqrt{\frac{\mathcal{P}_T T}{1-\tau}}\right),$$

where the final equality follows from $\mathcal{P}_T \triangleq \sum_{t=1}^{T} \sum_{i=1}^{n} \|\mathbf{1}_{(\mathcal{A}_t^* \cap \mathcal{V}_i)} - \mathbf{1}_{(\mathcal{A}_{t+1}^* \cap \mathcal{V}_i)}\|_2 \triangleq \sum_{t=1}^{T} \sum_{i=1}^{n} \|\mathbf{1}_{(\mathcal{A}_t^* \cap \mathcal{V}_i)} - \mathbf{1}_{(\mathcal{A}_{t+1}^* \cap \mathcal{V}_i)}\|_1 \triangleq \sum_{t=1}^{T} |\mathcal{A}_{t+1}^* \triangle \mathcal{A}_t^*|$.

# H Convergence Analysis of `MA-MPL` Algorithm

In this section, we prove Theorem 5.

At first, we verify that each policy vector $(\pi_{i,1}^{(k)}(t), \dots, \pi_{i,n}^{(k)}(t)) \in \prod_{i=1} \Delta_{\kappa_i}$, namely, we have the following lemma:

**Lemma 12.** *When the communication graph $G(\mathcal{V}, \mathcal{E})$ is connected, we can show that $(\pi_{i,1}^{(k)}(t), \dots, \pi_{i,n}^{(k)}(t)) \in \prod_{i=1} \Delta_{\kappa_i}$ for any $i \in \mathcal{N}$ and $0 \leq k \leq K$ as well as $t \in [T]$.*

*Proof.* From the Line 1 of Algorithm 2, we can know that $(\pi_{i,1}^{(0)}(t), \dots, \pi_{i,n}^{(0)}(t)) := \mathbf{0}, \forall t \in [T]$ such that $(\pi_{i,1}^{(0)}(t), \dots, \pi_{i,n}^{(0)}(t)) \in \prod_{i=1} \Delta_{\kappa_i}$. Moreover, from Lines 8, for a fixed $t \in [T]$, we know that only agent $i \in \mathcal{N}$ can change the policy $\pi_{i,i}^{(k)}(t)$. Furthermore, from Line 10, we also can infer that, for other agent $j \neq i$, its policy $\pi_{j,i}^{(k)}(t)$ is copy of some past iteration of $\pi_{i,i}^{(k)}(t)$, namely, there exists a $k_1 \leq k$ such that $\pi_{j,i}^{(k)}(t) = \pi_{i,i}^{(k_1)}(t)$. According to the Line 8 and Line 10, we have $\pi_{i,i}^{(k)}(t) = \pi_{i,i}^{(k-1)}(t) + \frac{1}{K} \mathbf{v}_i^{(k)}(t)$, so we have $\pi_{i,i}^{(k)}(t) = \frac{1}{K} \sum_{m=1}^{k} \mathbf{v}_i^{(m)}(t) \in \Delta_{\kappa_i}$ for any $k \in [K]$ ,$t \in [T]$ and $i \in \mathcal{N}$. Therefore, we have $(\pi_{i,1}^{(k)}(t), \dots, \pi_{i,n}^{(k)}(t)) \in \prod_{i=1} \Delta_{\kappa_i}$ for any $i \in \mathcal{N}$ and $k \in [K]$ as well as $t \in [T]$. $\square$

Note that, from Lines 11-13, each agent $i \in \mathcal{N}$ only uses the policy $\pi_{i,i}^{(K)}(t)$ to make decision, so the real policy vector taken by all agents is $(\pi_{1,1}^{(K)}(t), \dots, \pi_{n,n}^{(K)}(t))$. Motivated by this finding, we next investigate some relationships between the policy vector $(\pi_{1,1}^{(k)}(t), \dots, \pi_{n,n}^{(k)}(t))$ and $(\pi_{i,1}^{(k)}(t), \dots, \pi_{i,n}^{(k)}(t))$. That is,

**Lemma 13.** *When the communication graph $G(\mathcal{V}, \mathcal{E})$ is connected, if we set $\kappa \triangleq \sum_{i=1}^{n} \kappa_i$ and utilize the symbol $\mathbf{1}_\kappa$ to denote the $\kappa$-dimensional vector whose all elements are 1, we can show that*

$$0 \leq \frac{1}{n} \left\langle \mathbf{1}_\kappa, (\pi_{1,1}^{(k)}(t), \dots, \pi_{n,n}^{(k)}(t)) - (\pi_{i,1}^{(k)}(t), \dots, \pi_{i,n}^{(k)}(t)) \right\rangle \leq \frac{d(G)}{K},$$

*where $d(G)$ is the diameter of a graph, that is to say, the length of the shortest path between the most distanced nodes.*

*Proof.* From Lines 6-10, we can show that, for fixed $k \in [K]$ and $t \in [T]$, each $\pi_{i,j}^{(k)}(t) = \pi_{j,j}^{(k_j)}(t)$ where $k_j \in [k - d(G), k]$. Moreover, according to the Line 8 and Line 10, we have $\pi_{j,j}^{(k)}(t) = \pi_{j,j}^{(k-1)}(t) + \frac{1}{K}\mathbf{v}_j^{(k)}(t)$, so we have $\pi_{j,j}^{(k)}(t) = \frac{1}{K}\sum_{m=1}^{k}\mathbf{v}_j^{(m)}(t)$ for any $k \in [K]$ ,$t \in [T]$ and $i \in \mathcal{N}$. Therefore, we have that $\pi_{j,j}^{(k)}(t) - \pi_{i,j}^{(k)}(t) = \frac{1}{K}\sum_{m=k_j}^{j}\mathbf{v}_j^{(m)}(t)$ such that $0 \le \left\langle \mathbf{1}_{\kappa_j}, \pi_{j,j}^{(k)}(t) - \pi_{i,j}^{(k)}(t) \right\rangle \le \frac{k-k_j}{K} \le \frac{d(G)}{K}$. So we have the result of Lemma 13. $\qquad\square$

Next, we investigate the relationships between $(\pi_{1,1}^{(k)}(t), \ldots, \pi_{n,n}^{(k)}(t))$ and $(\pi_{1,1}^{(k-1)}(t), \ldots, \pi_{n,n}^{(k-1)}(t))$.

**Lemma 14.** *When the communication graph $G(\mathcal{V}, \mathcal{E})$ is connected, we can show that*

$$(\pi_{1,1}^{(k)}(t), \ldots, \pi_{n,n}^{(k)}(t)) - (\pi_{1,1}^{(k-1)}(t), \ldots, \pi_{n,n}^{(k-1)}(t)) = \frac{1}{K}(\mathbf{v}_1^{(k)}(t), \ldots, (\mathbf{v}_n^{(k)}(t));$$

$$0 \le \frac{1}{n}\left\langle \mathbf{1}_{\kappa}, (\pi_{1,1}^{(k)}(t), \ldots, \pi_{n,n}^{(k)}(t)) - (\pi_{1,1}^{(k-1)}(t), \ldots, \pi_{n,n}^{(k-1)}(t)) \right\rangle \le \frac{1}{K},$$

*where $\kappa \triangleq \sum_{i=1}^{n}\kappa_i$ and $d(G)$ is the diameter of a graph, i.e., the length of the shortest path between the most distanced nodes.*

*Proof.* According to the Line 8 and Line 10, we have $\pi_{j,j}^{(k)}(t) = \pi_{j,j}^{(k-1)}(t) + \frac{1}{K}\mathbf{v}_j^{(k)}(t)$. Therefore, we get result of Lemma 14. $\qquad\square$

With Lemma 13 and Lemma 14, we next give an upper bound about the error between the gradient $\nabla F_t(\pi_{1,1}^{(k)}(t), \ldots, \pi_{n,n}^{(k)}(t))$ and $\nabla F_t(\pi_{i,1}^{(k)}(t), \ldots, \pi_{i,n}^{(k)}(t))$.

**Lemma 15.** *When the communication graph $G(\mathcal{V}, \mathcal{E})$ is connected, if we set $M$ as the maximum marginal contribution of each monotone set function $f_t$, namely, $M \triangleq \max_{S \subseteq \mathcal{V}, e \in \mathcal{V} \setminus S, t \in [T]}(f_t(e|S))$, we can show that*

$$\left\| \nabla F_t(\pi_{1,1}^{(k)}(t), \ldots, \pi_{n,n}^{(k)}(t)) - \nabla F_t(\pi_{i,1}^{(k)}(t), \ldots, \pi_{i,n}^{(k)}(t)) \right\|_2 \le \frac{\sqrt{\kappa}nMd(G)}{K}$$

*where $\kappa = \sum_{i=1}^{n}\kappa_i$ and $d(G)$ is the diameter of a graph, i.e., the length of the shortest path between the most distanced nodes.*

*Proof.* At first, we investigate the error between the first-order derivative $\frac{\partial F_t}{\partial \pi_{i,m}}(\pi_{1,1}^{(k)}(t), \ldots, \pi_{n,n}^{(k)}(t))$ and the first-order derivative $\frac{\partial F_t}{\partial \pi_{i,m}}(\pi_{i,1}^{(k)}(t), \ldots, \pi_{i,n}^{(k)}(t))$ for some $i \in \mathcal{N}$ and $m \in [\kappa_i]$.

$$\left| \frac{\partial F_t}{\partial \pi_{i,m}}(\pi_{1,1}^{(k)}(t), \ldots, \pi_{n,n}^{(k)}(t)) - \frac{\partial F_t}{\partial \pi_{i,m}}(\pi_{i,1}^{(k)}(t), \ldots, \pi_{i,n}^{(k)}(t)) \right|$$

$$= \left| \left\langle \left( \int_0^1 \frac{\partial^2 F_t}{\partial \pi_{i,m}\partial \pi_{1,1}}, \ldots, \frac{\partial^2 F_t}{\partial \pi_{i,m}\partial \pi_{n,\kappa_n}} \right)(\mathbf{x}), \left( \pi_{1,1}^{(k)}(t) - \pi_{i,1}^{(k)}(t), \ldots, \pi_{n,n}^{(k)}(t) - \pi_{i,n}^{(k)}(t) \right) \right\rangle \right|$$

$$\le M \left| \left\langle \mathbf{1}_{\kappa}, \left( \pi_{1,1}^{(k)}(t) - \pi_{i,1}^{(k)}(t), \ldots, \pi_{n,n}^{(k)}(t) - \pi_{i,n}^{(k)}(t) \right) \right\rangle \right| \le \frac{nMd(G)}{K},$$

where the first equality comes from $\mathbf{x} = \left( \lambda\pi_{1,1}^{(k)}(t) + (1 - \lambda)\pi_{i,1}^{(k)}(t), \ldots, \lambda\pi_{n,n}^{(k)}(t) + (1 - \lambda)\pi_{i,n}^{(k)}(t) \right)$ for some $\lambda \in [0, 1]$, the first inequality follows from Remark 13 and the final inequality comes from Lemma 13. Then, we have that

$$\left\| \nabla F_t(\pi_{1,1}^{(k)}(t), \ldots, \pi_{n,n}^{(k)}(t)) - \nabla F_t(\pi_{i,1}^{(k)}(t), \ldots, \pi_{i,n}^{(k)}(t)) \right\|_2$$

$$\le \sqrt{\kappa}\max_{\pi_{i,m}} \left| \frac{\partial F_t}{\partial \pi_{i,m}}(\pi_{1,1}^{(k)}(t), \ldots, \pi_{n,n}^{(k)}(t)) - \frac{\partial F_t}{\partial \pi_{i,m}}(\pi_{i,1}^{(k)}(t), \ldots, \pi_{i,n}^{(k)}(t)) \right| \le \frac{\sqrt{\kappa}nMd(G)}{K},$$

where $\kappa = \sum_{i=1}^{n}\kappa_i$ and $d(G)$ is the diameter of a graph, i.e., the length of the shortest path between the most distanced nodes. $\qquad\square$

Now, we prove Theorem 5.

**Lemma 16.** *When the communication graph $G(\mathcal{V}, \mathcal{E})$ is connected, if we set $M$ as the maximum marginal contribution of each monotone set function $f_t$, namely, $M \triangleq \max_{S \subseteq \mathcal{V}, e \in \mathcal{V} \setminus S, t \in [T]} (f_t(e|S))$, we can show that,*

*i):) if $f_t$ is a monotone $\alpha$-weakly submodular function:*

$$(1 - e^{-\alpha}) \sum_{t=1}^{T} f_t(\mathcal{A}_t^*) - \sum_{t=1}^{T} \mathbb{E}\left(F_t(\boldsymbol{\pi}_{1,1}^{(K)}(t), \dots, \boldsymbol{\pi}_{n,n}^{(k)}(t))\right)$$

$$\leq \left(\frac{nM(\sum_{i=1}^{n} \kappa_i)}{2\alpha} + \frac{2\sqrt{\kappa}n^2 M d(G)}{\alpha}\right) \frac{T}{K} + \frac{2nM\sqrt{\kappa}}{\alpha} \frac{T}{\sqrt{L}}$$

$$+ \frac{1}{K} \sum_{k=1}^{K} \sum_{i=1}^{n} \sum_{t=1}^{T} \left\langle \mathbf{d}_i^{(k)}(t), \mathbf{1}_{(\mathcal{A}_t^* \cap \mathcal{V}_i)} - \mathbf{v}_i^{(k)}(t) \right\rangle;$$

*ii):) if $f_t$ is a monotone $(\gamma, \beta)$-weakly submodular:*

$$\frac{\gamma^2(1 - e^{-\phi(\gamma,\beta)})}{\phi(\gamma, \beta)} \sum_{t=1}^{T} f_t(\mathcal{A}_t^*) - \sum_{t=1}^{T} \mathbb{E}\left(F_t(\boldsymbol{\pi}_{1,1}^{(K)}(t), \dots, \boldsymbol{\pi}_{n,n}^{(k)}(t))\right)$$

$$\leq \left(\frac{nM(\sum_{i=1}^{n} \kappa_i)}{2\phi(\gamma, \beta)} + \frac{2\sqrt{\kappa}n^2 M d(G)}{\phi(\gamma, \beta)}\right) \frac{T}{K} + \frac{2nM\sqrt{\kappa}}{\phi(\gamma, \beta)} \frac{T}{\sqrt{L}}$$

$$- \frac{1}{K} \sum_{k=1}^{K} \sum_{i=1}^{n} \sum_{t=1}^{T} \left\langle \mathbf{d}_i^{(k)}(t), \mathbf{1}_{(\mathcal{A}_t^* \cap \mathcal{V}_i)} - \mathbf{v}_i^{(k)}(t) \right\rangle,$$

*where where we slightly abuse the notation $\mathbf{1}_{(\mathcal{A}_t^* \cap \mathcal{V}_i)}$ to denote $\kappa_i$-dimensional indicator vector over $(\mathcal{A}_t^* \cap \mathcal{V}_i)$. Note that $\mathbf{1}_{\mathcal{A}_t^*}$ is a $(\sum_{i=1}^{n} \kappa_i)$-dimensional indicator vector over $\mathcal{A}_t^*$ and $\phi(\gamma, \beta) = \beta(1 - \gamma) + \gamma^2$. Note the $\sum_{i=1}^{n} \sum_{t=1}^{T} \left\langle \mathbf{d}_i^{(k)}(t), \mathbf{1}_{(\mathcal{A}_t^* \cap \mathcal{V}_i)} - \mathbf{v}_i^{(k)}(t) \right\rangle$ is the dynamic regret of linear maximization oracle $Q_i^{(k)}$ over the competitive sequence $\left(\mathbf{1}_{(\mathcal{A}_1^* \cap \mathcal{V}_i)}, \dots, \mathbf{1}_{(\mathcal{A}_T^* \cap \mathcal{V}_i)}\right)$. Therefore, we also rewrite the previous two results as:*

*i):) if $f_t$ is a monotone $\alpha$-weakly submodular function:*

$$(1 - e^{-\alpha}) \sum_{t=1}^{T} f_t(\mathcal{A}_t^*) - \sum_{t=1}^{T} \mathbb{E}\left(F_t(\boldsymbol{\pi}_{1,1}^{(K)}(t), \dots, \boldsymbol{\pi}_{n,n}^{(k)}(t))\right)$$

$$\leq \left(\frac{nM(\sum_{i=1}^{n} \kappa_i)}{2\alpha} + \frac{2\sqrt{\kappa}n^2 M d(G)}{\alpha}\right) \frac{T}{K} + \frac{2nM\sqrt{\kappa}}{\alpha} \frac{T}{\sqrt{L}}$$

$$+ \frac{1}{K} \sum_{k=1}^{K} \sum_{i=1}^{n} \mathcal{R}^{Q_i^{(k)}}\left(\mathbf{1}_{(\mathcal{A}_1^* \cap \mathcal{V}_i)}, \dots, \mathbf{1}_{(\mathcal{A}_T^* \cap \mathcal{V}_i)}\right);$$

*ii):) if $f_t$ is a monotone $(\gamma, \beta)$-weakly submodular:*

$$\frac{\gamma^2(1 - e^{-\phi(\gamma,\beta)})}{\phi(\gamma, \beta)} \sum_{t=1}^{T} f_t(\mathcal{A}_t^*) - \sum_{t=1}^{T} \mathbb{E}\left(F_t(\boldsymbol{\pi}_{1,1}^{(K)}(t), \dots, \boldsymbol{\pi}_{n,n}^{(k)}(t))\right)$$

$$\leq \left(\frac{nM(\sum_{i=1}^{n} \kappa_i)}{2\phi(\gamma, \beta)} + \frac{2\sqrt{\kappa}n^2 M d(G)}{\phi(\gamma, \beta)}\right) \frac{T}{K} + \frac{2nM\sqrt{\kappa}}{\phi(\gamma, \beta)} \frac{T}{\sqrt{L}}$$

$$- \frac{1}{K} \sum_{k=1}^{K} \mathcal{R}^{Q_i^{(k)}}\left(\mathbf{1}_{(\mathcal{A}_1^* \cap \mathcal{V}_i)}, \dots, \mathbf{1}_{(\mathcal{A}_T^* \cap \mathcal{V}_i)}\right),$$

*where $\mathcal{R}^{Q_i^{(k)}}\left(\mathbf{1}_{(\mathcal{A}_1^* \cap \mathcal{V}_i)}, \dots, \mathbf{1}_{(\mathcal{A}_T^* \cap \mathcal{V}_i)}\right)$ is the dynamic regret of linear maximization oracle $Q_i^{(k)}$ over the competitive sequence $\left(\mathbf{1}_{(\mathcal{A}_1^* \cap \mathcal{V}_i)}, \dots, \mathbf{1}_{(\mathcal{A}_T^* \cap \mathcal{V}_i)}\right)$.*

*Proof.*

$$F_t(\boldsymbol{\pi}_{1,1}^{(k)}(t),\ldots,\boldsymbol{\pi}_{n,n}^{(k)}(t)) - F_t(\boldsymbol{\pi}_{1,1}^{(k-1)}(t),\ldots,\boldsymbol{\pi}_{n,n}^{(k-1)}(t))$$

$$\geq \left\langle \nabla F_t(\boldsymbol{\pi}_{1,1}^{(k-1)}(t),\ldots,\boldsymbol{\pi}_{n,n}^{(k-1)}(t)), (\boldsymbol{\pi}_{1,1}^{(k)}(t) - \boldsymbol{\pi}_{1,1}^{(k-1)}(t)),\ldots,(\boldsymbol{\pi}_{1,1}^{(k)}(t) - \boldsymbol{\pi}_{n,n}^{(k-1)}(t)) \right\rangle$$

$$- \frac{M(\sum_{i=1}^{n}\kappa_i)}{2} \left\| \left( \boldsymbol{\pi}_{1,1}^{(k)}(t) - \boldsymbol{\pi}_{1,1}^{(k-1)}(t),\ldots,(\boldsymbol{\pi}_{1,1}^{(k)}(t) - \boldsymbol{\pi}_{n,n}^{(k-1)}(t)) \right) \right\|_2^2$$

$$\geq \frac{1}{K} \left\langle \nabla F_t(\boldsymbol{\pi}_{1,1}^{(k-1)}(t),\ldots,\boldsymbol{\pi}_{n,n}^{(k-1)}(t)), (\mathbf{v}_1^{(k)}(t),\ldots,\mathbf{v}_n^{(k)}(t)) \right\rangle - \frac{nM(\sum_{i=1}^{n}\kappa_i)}{2K^2}$$

$$= \frac{1}{K} \left\langle \nabla F_t(\boldsymbol{\pi}_{1,1}^{(k-1)}(t),\ldots,\boldsymbol{\pi}_{n,n}^{(k-1)}(t)), \mathbf{1}_{\mathcal{A}_t^*} \right\rangle - \frac{nM(\sum_{i=1}^{n}\kappa_i)}{2K^2}$$

$$+ \frac{1}{K} \left\langle \nabla F_t(\boldsymbol{\pi}_{1,1}^{(k-1)}(t),\ldots,\boldsymbol{\pi}_{n,n}^{(k-1)}(t)), (\mathbf{v}_1^{(k)}(t),\ldots,\mathbf{v}_n^{(k)}(t)) - \mathbf{1}_{\mathcal{A}_t^*} \right\rangle$$

$$= \frac{1}{K} \underbrace{\left\langle \nabla F_t(\boldsymbol{\pi}_{1,1}^{(k-1)}(t),\ldots,\boldsymbol{\pi}_{n,n}^{(k-1)}(t)), \mathbf{1}_{\mathcal{A}_t^*} \right\rangle}_{\text{①}} - \frac{nM(\sum_{i=1}^{n}\kappa_i)}{2K^2}$$

$$+ \frac{1}{K} \underbrace{\left\langle \nabla F_t(\boldsymbol{\pi}_{1,1}^{(k-1)}(t),\ldots,\boldsymbol{\pi}_{n,n}^{(k-1)}(t)) - \sum_{i\in\mathcal{N}} \left[ \nabla F_t(\boldsymbol{\pi}_{i,1}^{(k-1)}(t),\ldots,\boldsymbol{\pi}_{i,n}^{(k-1)}(t)) \right]_{\pi_i}, (\mathbf{v}_1^{(k)}(t),\ldots,\mathbf{v}_n^{(k)}(t)) - \mathbf{1}_{\mathcal{A}_t^*} \right\rangle}_{\text{②}}$$

$$+ \frac{1}{K} \underbrace{\left\langle \sum_{i\in\mathcal{N}} \left[ \nabla F_t(\boldsymbol{\pi}_{i,1}^{(k-1)}(t),\ldots,\boldsymbol{\pi}_{i,n}^{(k-1)}(t)) \right]_{\pi_i} - \left( \mathbf{d}_1^{(k)}(t),\ldots,\mathbf{d}_n^{(k)}(t) \right), (\mathbf{v}_1^{(k)}(t),\ldots,\mathbf{v}_n^{(k)}(t)) - \mathbf{1}_{\mathcal{A}_t^*} \right\rangle}_{\text{③}}$$

$$+ \frac{1}{K} \left\langle \left( \mathbf{d}_1^{(k)}(t),\ldots,\mathbf{d}_n^{(k)}(t) \right), (\mathbf{v}_1^{(k)}(t),\ldots,\mathbf{v}_n^{(k)}(t)) - \mathbf{1}_{\mathcal{A}_t^*} \right\rangle,$$

where the first inequality follows the Lemma 5, namely, our proposed policy-based continuous extension $F_t$ is $(M\sum_{i=1}^{n}\kappa_i)$-smooth andthe symbol $\left[ \nabla F_t^a(\boldsymbol{\pi}_{i,1}(t),\ldots,\boldsymbol{\pi}_{i,1}(t)) \right]_{\pi_i}$ is the projection over the policy $\pi_i$, namely, $\left[ \nabla F_t^a(\boldsymbol{\pi}_{i,1}(t),\ldots,\boldsymbol{\pi}_{i,1}(t)) \right]_{\pi_i}$ represents a $(\sum_{i=1}^{n}\kappa_i)$-dimensional vector that only keeps the first-order derivative at variable $\pi_{i,m}, \forall m \in [\kappa_i]$ and set other coordinates to 0, that is to say,

$$\left[ \nabla F_t^a(\boldsymbol{\pi}_{i,1}(t),\ldots,\boldsymbol{\pi}_{i,1}(t)) \right]_{\pi_i} \triangleq \left( \ldots, 0, \ldots, \underbrace{\frac{\partial F_t}{\partial \pi_{i,1}}(\mathbf{x}), \ldots, \frac{\partial F_t}{\partial \pi_{i,\kappa_i}}(\mathbf{x})}_{\kappa_i}, \ldots, 0, \ldots \right),$$

where $\mathbf{x} \triangleq (\boldsymbol{\pi}_{i,1}(t),\ldots,\boldsymbol{\pi}_{i,1}(t))$.

For ①, according to **4)** of Theorem 1, when $f_t$ is monotone $\alpha$-weakly DR-submodular,

$$\left\langle \nabla F_t(\boldsymbol{\pi}_{1,1}^{(k-1)}(t),\ldots,\boldsymbol{\pi}_{n,n}^{(k-1)}(t)), \mathbf{1}_{\mathcal{A}_t^*} \right\rangle \geq \alpha \left( f_t(\mathcal{A}_t^*) - F_t(\boldsymbol{\pi}_{1,1}^{(k-1)}(t),\ldots,\boldsymbol{\pi}_{n,n}^{(k-1)}(t)) \right);$$

As for the setting that $f_t$ is monotone $(\gamma,\beta)$-weakly submodular, we also have

$$\left\langle \nabla F_t(\boldsymbol{\pi}_{1,1}^{(k-1)}(t),\ldots,\boldsymbol{\pi}_{n,n}^{(k-1)}(t)), \mathbf{1}_{\mathcal{A}_t^*} \right\rangle \geq \gamma^2 f_t(\mathcal{A}_t^*) - (\beta(1-\gamma)+\gamma^2) F_t \left( \boldsymbol{\pi}_{1,1}^{(k-1)}(t),\ldots,\boldsymbol{\pi}_{n,n}^{(k-1)}(t) \right).$$

For ②, according to Lemma 15, we have

$$
\left| \left\langle \nabla F_t(\boldsymbol{\pi}_{1,1}^{(k-1)}(t), \ldots, \boldsymbol{\pi}_{n,n}^{(k-1)}(t)) - \sum_{i \in \mathcal{N}} \left[ \nabla F_t(\boldsymbol{\pi}_{i,1}^{(k-1)}(t), \ldots, \boldsymbol{\pi}_{i,n}^{(k-1)}(t)) \right]_{\pi_i}, (\mathbf{v}_1^{(k)}(t), \ldots, \mathbf{v}_n^{(k)}(t)) - \mathbf{1}_{\mathcal{A}_t^*} \right\rangle \right|
$$

$$
= \left| \left\langle \sum_{i \in \mathcal{N}} \left[ \nabla F_t(\boldsymbol{\pi}_{1,1}^{(k-1)}(t), \ldots, \boldsymbol{\pi}_{n,n}^{(k-1)}(t)) \right]_{\pi_i} - \sum_{i \in \mathcal{N}} \left[ \nabla F_t(\boldsymbol{\pi}_{i,1}^{(k-1)}(t), \ldots, \boldsymbol{\pi}_{i,n}^{(k-1)}(t)) \right]_{\pi_i}, (\mathbf{v}_1^{(k)}(t), \ldots, \mathbf{v}_n^{(k)}(t)) - \mathbf{1}_{\mathcal{A}_t^*} \right\rangle \right|
$$

$$
\leq \left\| \sum_{i \in \mathcal{N}} \left[ \nabla F_t(\boldsymbol{\pi}_{1,1}^{(k-1)}(t), \ldots, \boldsymbol{\pi}_{n,n}^{(k-1)}(t)) \right]_{\pi_i} - \sum_{i \in \mathcal{N}} \left[ \nabla F_t(\boldsymbol{\pi}_{i,1}^{(k-1)}(t), \ldots, \boldsymbol{\pi}_{i,n}^{(k-1)}(t)) \right]_{\pi_i} \right\|_2 \left\| (\mathbf{v}_1^{(k)}(t), \ldots, \mathbf{v}_n^{(k)}(t)) - \mathbf{1}_{\mathcal{A}_t^*} \right\|_2
$$

$$
\leq 2n \sum_{i \in \mathcal{N}} \left\| \left[ \nabla F_t(\boldsymbol{\pi}_{1,1}^{(k-1)}(t), \ldots, \boldsymbol{\pi}_{n,n}^{(k-1)}(t)) \right]_{\pi_i} - \left[ \nabla F_t(\boldsymbol{\pi}_{i,1}^{(k-1)}(t), \ldots, \boldsymbol{\pi}_{i,n}^{(k-1)}(t)) \right]_{\pi_i} \right\|_2
$$

$$
\leq 2n \sum_{i \in \mathcal{N}} \left\| \nabla F_t(\boldsymbol{\pi}_{1,1}^{(k-1)}(t), \ldots, \boldsymbol{\pi}_{n,n}^{(k-1)}(t)) - \nabla F_t(\boldsymbol{\pi}_{i,1}^{(k-1)}(t), \ldots, \boldsymbol{\pi}_{i,n}^{(k-1)}(t)) \right\|_2 \leq \frac{2\sqrt{\kappa} n^2 M d(G)}{K}.
$$

As for ③,

$$
\mathbb{E} \left| \left\langle \sum_{i \in \mathcal{N}} \left[ \nabla F_t(\boldsymbol{\pi}_{i,1}^{(k-1)}(t), \ldots, \boldsymbol{\pi}_{i,n}^{(k-1)}(t)) \right]_{\pi_i} - \left( \mathbf{d}_1^{(k)}(t), \ldots, \mathbf{d}_n^{(k)}(t) \right), (\mathbf{v}_1^{(k)}(t), \ldots, \mathbf{v}_n^{(k)}(t)) - \mathbf{1}_{\mathcal{A}_t^*} \right\rangle \right|
$$

$$
\leq \mathbb{E} \left( \left\| \sum_{i \in \mathcal{N}} \left[ \nabla F_t(\boldsymbol{\pi}_{i,1}^{(k-1)}(t), \ldots, \boldsymbol{\pi}_{i,n}^{(k-1)}(t)) \right]_{\pi_i} - \left( \mathbf{d}_1^{(k)}(t), \ldots, \mathbf{d}_n^{(k)}(t) \right) \right\|_2 \left\| (\mathbf{v}_1^{(k)}(t), \ldots, \mathbf{v}_n^{(k)}(t)) - \mathbf{1}_{\mathcal{A}_t^*} \right\|_2 \right)
$$

$$
\leq 2n \mathbb{E} \left( \left\| \sum_{i \in \mathcal{N}} \left[ \nabla F_t(\boldsymbol{\pi}_{i,1}^{(k-1)}(t), \ldots, \boldsymbol{\pi}_{i,n}^{(k-1)}(t)) \right]_{\pi_i} - \left( \mathbf{d}_1^{(k)}(t), \ldots, \mathbf{d}_n^{(k)}(t) \right) \right\|_2 \right)
$$

$$
\leq 2n \sqrt{ \mathbb{E} \left( \frac{1}{L} \left\| \left( \mathbf{g}_1^{(k)}(t), \ldots, \mathbf{g}_n^{(k)}(t) \right) \right\|_2 \right) }
$$

$$
\leq 2n \sqrt{ \frac{\kappa M^2}{L} },
$$

where the third inequality follows Lines 15-20, namely, $\left( \mathbf{d}_1^{(k)}(t), \ldots, \mathbf{d}_n^{(k)}(t) \right)$ is $L$-batch stochastic gradient for $\sum_{i \in \mathcal{N}} \left[ \nabla F_t(\boldsymbol{\pi}_{i,1}^{(k-1)}(t), \ldots, \boldsymbol{\pi}_{i,n}^{(k-1)}(t)) \right]_{\pi_i}$ and the final inequality follows from Remark 13 and $\kappa = \sum_{i=1}^n \kappa_i$.

Therefore, when $f_t$ is monotone $\alpha$-weakly DR-submodular, we have that

$$
\mathbb{E} \left( F_t(\boldsymbol{\pi}_{1,1}^{(k)}(t), \ldots, \boldsymbol{\pi}_{n,n}^{(k)}(t)) - F_t(\boldsymbol{\pi}_{1,1}^{(k-1)}(t), \ldots, \boldsymbol{\pi}_{n,n}^{(k-1)}(t)) \right)
$$

$$
\geq \frac{\alpha}{K} \mathbb{E} \left( f_t(\mathcal{A}_t^*) - F_t(\boldsymbol{\pi}_{1,1}^{(k-1)}(t), \ldots, \boldsymbol{\pi}_{n,n}^{(k-1)}(t)) \right) - \frac{n M (\sum_{i=1}^n \kappa_i)}{2 K^2} - \frac{2\sqrt{\kappa} n^2 M d(G)}{K^2} - \frac{2n M}{K} \sqrt{\frac{\kappa}{L}}
$$

$$
+ \frac{1}{K} \left\langle \left( \mathbf{d}_1^{(k)}(t), \ldots, \mathbf{d}_n^{(k)}(t) \right), (\mathbf{v}_1^{(k)}(t), \ldots, \mathbf{v}_n^{(k)}(t)) - \mathbf{1}_{\mathcal{A}_t^*} \right\rangle.
$$

Then, we can have that

$$
\mathbb{E} \left( f_t(\mathcal{A}_t^*) - F_t(\boldsymbol{\pi}_{1,1}^{(k)}(t), \ldots, \boldsymbol{\pi}_{n,n}^{(k)}(t)) \right)
$$

$$
\leq \left( 1 - \frac{\alpha}{K} \right) \mathbb{E} \left( f_t(\mathcal{A}_t^*) - F_t(\boldsymbol{\pi}_{1,1}^{(k-1)}(t), \ldots, \boldsymbol{\pi}_{n,n}^{(k-1)}(t)) \right) + \frac{n M (\sum_{i=1}^n \kappa_i)}{2 K^2} + \frac{2\sqrt{\kappa} n^2 M d(G)}{K^2} + \frac{2n M}{K} \sqrt{\frac{\kappa}{L}}
$$

$$
- \frac{1}{K} \left\langle \left( \mathbf{d}_1^{(k)}(t), \ldots, \mathbf{d}_n^{(k)}(t) \right), (\mathbf{v}_1^{(k)}(t), \ldots, \mathbf{v}_n^{(k)}(t)) - \mathbf{1}_{\mathcal{A}_t^*} \right\rangle.
$$

As a result,

$$\mathbb{E}\left(f_t(\mathcal{A}_t^*) - F_t(\boldsymbol{\pi}_{1,1}^{(K)}(t), \ldots, \boldsymbol{\pi}_{n,n}^{(k)}(t))\right)$$

$$\leq \left(1 - \frac{\alpha}{K}\right)^K \mathbb{E}\left(f_t(\mathcal{A}_t^*) - F_t(\boldsymbol{\pi}_{1,1}^{(0)}(t), \ldots, \boldsymbol{\pi}_{n,n}^{(k-1)}(t))\right)$$

$$- \frac{1}{K}\sum_{k=1}^K \left\langle \left(\mathbf{d}_1^{(k)}(t), \ldots, \mathbf{d}_n^{(k)}(t)\right), (\mathbf{v}_1^{(k)}(t), \ldots, \mathbf{v}_n^{(k)}(t)) - \mathbf{1}_{\mathcal{A}_t^*}\right\rangle$$

$$+ \sum_{m=1}^K \left(1 - \frac{\alpha}{K}\right)^{K-m} \left(\frac{nM(\sum_{i=1}^n \kappa_i)}{2K^2} + \frac{2\sqrt{\kappa}n^2 Md(G)}{K^2} + \frac{2nM}{K}\sqrt{\frac{\kappa}{L}}\right)$$

$$\leq \left(1 - \frac{\alpha}{K}\right)^K \mathbb{E}\left(f_t(\mathcal{A}_t^*)\right) + \frac{nM(\sum_{i=1}^n \kappa_i)}{2\alpha K} + \frac{2\sqrt{\kappa}n^2 Md(G)}{\alpha K} + \frac{2nM}{\alpha}\sqrt{\frac{\kappa}{L}}$$

$$- \frac{1}{K}\sum_{k=1}^K \left\langle \left(\mathbf{d}_1^{(k)}(t), \ldots, \mathbf{d}_n^{(k)}(t)\right), (\mathbf{v}_1^{(k)}(t), \ldots, \mathbf{v}_n^{(k)}(t)) - \mathbf{1}_{\mathcal{A}_t^*}\right\rangle$$

$$\leq e^{-\alpha}\mathbb{E}\left(f_t(\mathcal{A}_t^*)\right) + \frac{nM(\sum_{i=1}^n \kappa_i)}{2\alpha K} + \frac{2\sqrt{\kappa}n^2 Md(G)}{\alpha K} + \frac{2nM}{\alpha}\sqrt{\frac{\kappa}{L}}$$

$$- \frac{1}{K}\sum_{k=1}^K \left\langle \left(\mathbf{d}_1^{(k)}(t), \ldots, \mathbf{d}_n^{(k)}(t)\right), (\mathbf{v}_1^{(k)}(t), \ldots, \mathbf{v}_n^{(k)}(t)) - \mathbf{1}_{\mathcal{A}_t^*}\right\rangle,$$

where the second inequality follows from $\sum_{m=1}^K \left(1 - \frac{\alpha}{K}\right)^{K-m} \leq \frac{K}{\alpha}$, $F_t(\mathbf{0}) = 0$ and the final inequality from from $(1 - \frac{\alpha}{K})^K \leq e^{-\alpha}$ when $K \geq 3$.

So we can get the following result,

$$(1 - e^{-\alpha})\sum_{t=1}^T f_t(\mathcal{A}_t^*) - \sum_{t=1}^T \mathbb{E}\left(F_t(\boldsymbol{\pi}_{1,1}^{(K)}(t), \ldots, \boldsymbol{\pi}_{n,n}^{(k)}(t))\right)$$

$$\leq \left(\frac{nM(\sum_{i=1}^n \kappa_i)}{2\alpha} + \frac{2\sqrt{\kappa}n^2 Md(G)}{\alpha}\right)\frac{T}{K} + \frac{2nM\sqrt{\kappa}}{\alpha}\frac{T}{\sqrt{L}}$$

$$- \frac{1}{K}\sum_{k=1}^K \sum_{i=1}^n \sum_{t=1}^T \left\langle \mathbf{d}_i^{(k)}(t), \mathbf{v}_i^{(k)}(t) - \mathbf{1}_{(\mathcal{A}_t^* \cap \mathcal{V}_i)}\right\rangle$$

where we slightly abuse the notation $\mathbf{1}_{(\mathcal{A}_t^* \cap \mathcal{V}_i)}$ to denote $\kappa_i$-dimensional indicator vector over $(\mathcal{A}_t^* \cap \mathcal{V}_i)$. Note that $\mathbf{1}_{\mathcal{A}_t^*}$ is a $(\sum_{i=1}^n \kappa_i)$-dimensional indicator vector over $\mathcal{A}_t^*$.

Similarly, when $f_t$ is monotone $(\gamma, \beta)$-weakly submodular, we also have

$$\mathbb{E}\left(F_t(\boldsymbol{\pi}_{1,1}^{(k)}(t), \ldots, \boldsymbol{\pi}_{n,n}^{(k)}(t)) - F_t(\boldsymbol{\pi}_{1,1}^{(k-1)}(t), \ldots, \boldsymbol{\pi}_{n,n}^{(k-1)}(t))\right)$$

$$\geq \frac{1}{K}\mathbb{E}\left(\gamma^2 f_t(\mathcal{A}_t^*) - \phi(\gamma, \beta)F_t(\boldsymbol{\pi}_{1,1}^{(k-1)}(t), \ldots, \boldsymbol{\pi}_{n,n}^{(k-1)}(t))\right) - \frac{nM(\sum_{i=1}^n \kappa_i)}{2K^2} - \frac{2\sqrt{\kappa}n^2 Md(G)}{K^2} - \frac{2nM}{K}\sqrt{\frac{\kappa}{L}}$$

$$+ \frac{1}{K}\left\langle \left(\mathbf{d}_1^{(k)}(t), \ldots, \mathbf{d}_n^{(k)}(t)\right), (\mathbf{v}_1^{(k)}(t), \ldots, \mathbf{v}_n^{(k)}(t)) - \mathbf{1}_{\mathcal{A}_t^*}\right\rangle,$$

where $\phi(\gamma, \beta) = \beta(1 - \gamma) + \gamma^2$.

Then, we have

$$\mathbb{E}\left(\gamma^2 f_t(\mathcal{A}_t^*) - \phi(\gamma, \beta)F_t(\boldsymbol{\pi}_{1,1}^{(k)}(t), \ldots, \boldsymbol{\pi}_{n,n}^{(k)}(t))\right)$$

$$\leq \left(1 - \frac{\phi(\gamma, \beta)}{K}\right)\mathbb{E}\left(\gamma^2 f_t(\mathcal{A}_t^*) - \phi(\gamma, \beta)F_t(\boldsymbol{\pi}_{1,1}^{(k-1)}(t), \ldots, \boldsymbol{\pi}_{n,n}^{(k-1)}(t))\right)$$

$$+ \phi(\gamma, \beta)\left(\frac{nM(\sum_{i=1}^n \kappa_i)}{2K^2} + \frac{2\sqrt{\kappa}n^2 Md(G)}{K^2} + \frac{2nM}{K}\sqrt{\frac{\kappa}{L}}\right)$$

$$- \frac{\phi(\gamma, \beta)}{K}\left\langle \left(\mathbf{d}_1^{(k)}(t), \ldots, \mathbf{d}_n^{(k)}(t)\right), (\mathbf{v}_1^{(k)}(t), \ldots, \mathbf{v}_n^{(k)}(t)) - \mathbf{1}_{\mathcal{A}_t^*}\right\rangle.$$

As a result, we can get the following result,

$$\frac{\gamma^2(1-e^{-\phi(\gamma,\beta)})}{\phi(\gamma,\beta)}\sum_{t=1}^{T}f_t(\mathcal{A}_t^*)-\sum_{t=1}^{T}\mathbb{E}\left(F_t(\boldsymbol{\pi}_{1,1}^{(K)}(t),\ldots,\boldsymbol{\pi}_{n,n}^{(k)}(t))\right)$$

$$\leq\left(\frac{nM(\sum_{i=1}^{n}\kappa_i)}{2\phi(\gamma,\beta)}+\frac{2\sqrt{\kappa}n^2Md(G)}{\phi(\gamma,\beta)}\right)\frac{T}{K}+\frac{2nM\sqrt{\kappa}}{\phi(\gamma,\beta)}\frac{T}{\sqrt{L}}-\frac{1}{K}\sum_{k=1}^{K}\sum_{i=1}^{n}\sum_{t=1}^{T}\left\langle\mathbf{d}_i^{(k)}(t),\mathbf{v}_i^{(k)}(t)-\mathbf{1}_{(\mathcal{A}_t^*\cap\mathcal{V}_i)}\right\rangle,$$

$\square$

As a result, when $K=\sqrt{T}$ and $L=T$, if Assumption 2 holds, that is, each linear maximization oracle $Q_i^{(k)}$ can achieve the a regret bound of $\mathcal{O}(\sqrt{V_TT})$ where $V_T$ is the variation of any feasible path $(\mathbf{u}_1,\ldots,\mathbf{u}_T)$ where $\mathbf{u}_t\in\Delta_{\kappa_i},\forall t\in[T]$, that is, $V_T\triangleq\sum_{t=2}^{T}\|\mathbf{u}_t-\mathbf{u}_{t-1}\|_2$ for any path $(\mathbf{u}_1,\ldots,\mathbf{u}_T)\in\prod_{t=1}^{T}\Delta_{\kappa_i}$, then we can have the following results:

**i)**: when $f_t$ is a monotone $\alpha$-weakly submodular function, Algorithm 2 achieves:

$$(1-e^{-\alpha})\sum_{t=1}^{T}f_t(\mathcal{A}_t^*)-\sum_{t=1}^{T}\mathbb{E}\left(F_t(\boldsymbol{\pi}_{1,1}^{(K)}(t),\ldots,\boldsymbol{\pi}_{n,n}^{(k)}(t))\right)$$

$$\leq\left(\frac{nM(\sum_{i=1}^{n}\kappa_i)}{2\alpha}+\frac{2\sqrt{\kappa}n^2Md(G)}{\alpha}\right)\frac{T}{K}+\frac{2nM\sqrt{\kappa}}{\alpha}\frac{T}{\sqrt{L}}+\frac{1}{K}\sum_{k=1}^{K}\sum_{i=1}^{n}\mathcal{R}^{Q_i^{(k)}}\left(\mathbf{1}_{(\mathcal{A}_1^*\cap\mathcal{V}_i)},\ldots,\mathbf{1}_{(\mathcal{A}_T^*\cap\mathcal{V}_i)}\right)$$

$$\leq\frac{1}{K}\sum_{k=1}^{K}\sum_{i=1}^{n}\mathcal{O}\left(\sqrt{T\sum_{t=1}^{T-1}\left\|\mathbf{1}_{(\mathcal{A}_{t+1}^*\cap\mathcal{V}_i)}-\mathbf{1}_{(\mathcal{A}_t^*\cap\mathcal{V}_i)}\right\|_2}\right)+\mathcal{O}\left(d(G)\sqrt{T}\right)$$

$$\leq\sum_{i=1}^{n}\mathcal{O}\left(\sqrt{T\sum_{t=1}^{T-1}\left\|\mathbf{1}_{\mathcal{A}_{t+1}^*}-\mathbf{1}_{\mathcal{A}_t^*}\right\|_2}\right)+\mathcal{O}\left(d(G)\sqrt{T}\right)$$

$$\leq\mathcal{O}\left(d(G)\sqrt{T\sum_{t=1}^{T-1}\left\|\mathbf{1}_{\mathcal{A}_{t+1}^*}-\mathbf{1}_{\mathcal{A}_t^*}\right\|_2}\right)=\mathcal{O}\left(d(G)\sqrt{T\mathcal{P}_T}\right),$$

where the third inequality follows from the concavity of $\sqrt{\cdot}$ function and the final equality comes from $\mathcal{P}_T\triangleq\sum_{t=1}^{T}|\mathcal{A}_{t+1}^*\triangle\mathcal{A}_t^*|\triangleq\sum_{t=1}^{T-1}\|\mathbf{1}_{\mathcal{A}_t^*}-\mathbf{1}_{\mathcal{A}_{t+1}^*)}\|_1\triangleq\sum_{t=1}^{T-1}\|\mathbf{1}_{\mathcal{A}_t^*}-\mathbf{1}_{\mathcal{A}_{t+1}^*}\|_2$.

**ii)**: Similarly, when $f_t$ is a monotone $(\gamma,\beta)$-weakly submodular:

$$\frac{\gamma^2(1-e^{-\phi(\gamma,\beta)})}{\phi(\gamma,\beta)}\sum_{t=1}^{T}f_t(\mathcal{A}_t^*)-\sum_{t=1}^{T}\mathbb{E}\left(F_t(\boldsymbol{\pi}_{1,1}^{(K)}(t),\ldots,\boldsymbol{\pi}_{n,n}^{(k)}(t))\right)$$

$$\leq\mathcal{O}\left(d(G)\sqrt{T\sum_{t=1}^{T-1}\left\|\mathbf{1}_{\mathcal{A}_{t+1}^*}-\mathbf{1}_{\mathcal{A}_t^*}\right\|_2}\right)=\mathcal{O}\left(d(G)\sqrt{T\mathcal{P}_T}\right).$$

# I   Limitation and Broader Impact

In this work, in order to eliminate the dependence on the unknown DR ratio and submodularity ratio in our proposed `MA-SPL` algorithm, we introduce a *parameter-free* online algorithm named `MA-MPL` for the MA-OC problem. However, this new `MA-MPL` algorithm typically incurs greater communication complexity, as shown in Table 1. Notably, recent studies [74, 124] employed a blocking procedure from [121] to reduce the number of communication in the decentralized online Frank-Wolfe algorithm [133]. Since our `MA-SPL` algorithm also can be viewed a variant of decentralized online Frank-Wolfe algorithm [133], we believe that this blocking strategy is a promising technique to help reduce the communication complexity of our proposed `MA-MPL` algorithm. We plan to explore this in future work. Furthermore, this work focuses on theoretically exploring MA-OC problem with $(\gamma,\beta)$-weakly submodular and $\alpha$-weakly DR-submodular objectives. So we do not foresee any form of negative social impact induced by our work.

