# OpenReview forum: "Effective Policy Learning for Multi-Agent Online Coordination Beyond Submodular Objectives"
_NeurIPS.cc/2025/Conference — NeurIPS 2025 poster_

### Official Review · Reviewer_fXn2 · 2025-06-28

**Clarity:** 3
**Significance:** 2
**Originality:** 2
**Rating:** 4
**Confidence:** 3

**Summary:**

This paper deals with a multi-player cooperative online learning problem with monotone submodular or non-submodular objectives. The ground set is partitioned into $n$ groups, and each player $i \in [n]$ chooses one element from the $i$th group in each round. The goal is to upper-bound on approximate dynamic regret with approximation ratio $\rho$. The paper's contributions are three-fold. The first one is to introduce a generalization of the multilinear extension, which they call policy-based continuous extension. The second one is to apply the boosting technique to improve the approximation ratio (for example, to $1-c/e$ for monotone submodular functions with curvature $c$). The third one is to combine this offline technique with existing dynamic regret minimization algorithms to obtain dynamic regret upper bounds. The proposed algorithm are also empirically validated.

**Questions:**

Please see the weaknesses discussed above. If my understanding contains serious mistakes, I would be happy to reconsider my score. For example, the proposed multilinear extension is essentially different from existing ones, I would appreciate your clarifications.

**Ethical Concerns:**

["NO or VERY MINOR ethics concerns only"]

**Final Justification:**

I do not change the score from the initial review. I hope the points I raised were addressed in the revised version.

**Limitations:**

Yes

**Quality:**

3

**Strengths And Weaknesses:**

Strengths:
- Among this paper's contributions, the boosting technique is most interesting to me. This was originally proposed by Zhang, Deng, Chen, Hu, and Yang in ICML 2022 for continuous submodular maximization. This method designs an optimal weight and maximizes a surrogate function that is the weighted average of the objective function. This paper applies the boosting technique to curvature-bounded submodular functions, $\gamma$-weakly DR-submodular functions, and $(\gamma, \beta)$-weakly submodular functions to obtain good approximation ratio guarantees.
- The problem setting of multi-agent online coordination is interesting and opens the door to new applications of submodular maximization, although it was already proposed in Zhang, Wan, Yang, Shen, and Tao in ICLR 2025. If the authors provided more practical applications in which parameters $c$, $\gamma$, and $\beta$ can be bounded, the significance of this paper is clearer.

---
Weaknesses:
- I am not a big fan of the concept of approximate regret. Since the solution in each round can exceed the approximation ratio $\rho$, theoretical guarantees on approximate regret do not really represent the "regret" for not playing the optimal action in hindsight. It just guarantees convergence of the approximation ratio of the total reward in hindsight. This paper considers dynamic regret, and its interpretation is more complicated.
- The authors claim that the policy-based continuous extension is their contribution. This is a generalization of the multilinear extension to partition matroid constraints (or $k$-submodular functions in context). However, the same idea already appeared in existing papers. For example, Sahin, Bian, Buhmann, and Krause called it "generalized multilinear extension" and used it for submodular maximization on the integer lattice. Wang and Zhou used in ArXiv 2021 paper for $k$-submodular maximization.
- I understand that all the factors of this paper (multi-agent coordination, dynamics regret minimization, and the boosting technique) are independently interesting to some extent, but these combinations make the paper complicated and blur the paper's focus in my opinion.

---

> ### Author Rebuttal · Authors · 2025-07-27
>
> Thank you very much for your careful reading and constructive feedback. We are grateful for the time and effort you have dedicated to reviewing our manuscript. In what follows, we will address some of your concerns in `Weaknesses`.
>
> ---
> > `Weakness1`: I am not a big fan of the concept of approximate regret. Since the solution in each round can exceed the approximation ratio $\rho$, theoretical guarantees on approximate regret do not really represent the ``regret" for not playing the optimal action in hindsight. It just guarantees convergence of the approximation ratio of the total reward in hindsight. This paper considers dynamic regret, and its interpretation is more complicated.
> ---
> Thank you for your insightful comments. We fully agree with your viewpoint.
>
> Recently, after reading the work[Tajdini et al., 2024], we also realize that the $\rho$-Regret is not be a perfect metric for online (weakly) submodular maximization problem, as the rewards or solutions obtained in each round can indeed surpass the predefined approximation ratio $\rho$. The primary reason why we adopt the dynamic $\rho$-Regret in this paper is to maintain consistency with the previous studies [Xu et al., 2023b,a, Zhang et al., 2025].
>
> ---
> > `Weakness2`: The authors claim that the policy-based continuous extension is their contribution. This is a generalization of the multilinear extension to partition matroid constraints (or $k$-submodular functions in context). However, the same idea already appeared in existing papers. For example, Sahin, Bian, Buhmann, and Krause called it "generalized multilinear extension" and used it for submodular maximization on the integer lattice. Wang and Zhou used in ArXiv 2021 paper for $k$-submodular maximization.
> ---
> Thank you for your detailed feedback.
>
> We agree with your perspective. If each agent has an action set of the **same** size, that is, $V_{i}=(v_{i,1},\dots,v_{i,k})$ and we represent the decision of all $n$ agents as a lattice vector $\mathbf{s}=(s_{1},\dots,s_{n})$ where $s_{i}\in(0,1,\dots,k)$ stands for the $i$-th agent choosing action $v_{i,s_{i}}$ (with $s_{i}=0$ denoting the null action $\emptyset$), then the multi-agent coordination problem (1) in our paper can be transformed as a lattice function maximization problem: $\max_{\mathbf{s}\in(0,\dots,k)^n}c_{t}(\mathbf{s})$ where $c_{t}(\mathbf{s})=f_{t}(\cup_{i=1}^{n}\{v_{i,s_{i}}\})$ and each $v_{i,0}$ represents $\emptyset$. Under this reformulation, the continuous extension introduced in [Sahin et al., 2020, Zhou et al., 2025] is indeed equivalent to the policy-based continuous extension in this paper.
>
> Since our proposed policy-based continuous extension was primarily inspired by concepts from multi-agent cooperative game [Albrecht et al., 2024, Zhang et al., 2021], we are grateful for your highlighting of these relevant works in another field regarding lattice submodular or k-submodular maximization that we had previously overlooked. We will incorporate them into the final version of our manuscript. Furthermore, we also want to emphasize that there exist notable **differences** between our work and the prior studies[Sahin et al., 2020, Zhou et al., 2025] about lattice function or $k$-submodular maximization.
>
> **a)** The lattice formulation typically requires an **order relationship** among different actions of the same agent.  For instance, in the facility location problem, each action  $v_{i,j}$ usually represents a sensor with a coverage radius of $j$ placed at location  $i$ and thus we know $v_{i,k}\ge v_{i,k-1}\dots\ge v_{i,1}$. In other words, the sensor with a coverage radius of $j$ is better than that with radius of $k$ when $k<j$. However, in our multi-agent coordination problem, we do not impose any specific **order relationship** on the decisions space. Without this predefined order, we even cannot ensure that the aforementioned $c_{t}(\mathbf{s})$ is a lattice submodular or $k$-submodular function over the domain $(0,\dots,k)^n$ when $f_{t}$ is submodular.  It is worth noting that the definition of lattice submodular or $k$-submodular function typically requires an order relationship for coordinate-wise maximum and minimum operations.  As a result, we think the results and algorithms of [Sahin et al., 2020, Zhou et al., 2025]  are not directly applicable to our scenario, despite the equivalence in the underlying continuous extension.
>
> **b)** In addition to submodularity, our paper also considers the unexplored weak submodularity and allows for varying sizes of action sets among agents.
>
> **(We apologize for not having located the Wang and Zhou paper from 2021 on ArXiv regarding $k$-submodular maximization. However, we have found a similar work [Zhou et al., 2025]. But we are unsure if this is the paper Reviewer fXn2 mentioned.)**
>
> ---
> > `Weakness3`: I understand that all the factors of this paper (multi-agent coordination, dynamics regret minimization, and the boosting technique) are independently interesting to some extent, but these combinations make the paper complicated and blur the paper's focus in my opinion.
>
> ---
> Thanks for your constructive suggestions.  We agree that the combination of multiple factors complicates our paper. To better highlight the core of multi-agent coordination, we plan to provide more specific details about multi-target tracking in the numerical experiments in the final version. Meanwhile, we will offer more in-depth explanations and motivations regarding our proposed  **MA-SPL** and **MA-MPL** algorithms in Section 5.2.
>
> ---
> **References**
>
> - S. V. Albrecht, F. Christianos, and L. Schafer. Multi-Agent Reinforcement Learning: Foundations and Modern Approaches. MIT Press, 2024.
>
> - A. Sahin, Y. Bian, J. Buhmann, and A. Krause. From sets to multisets: provable variational inference for probabilistic integer submodular models. ICML 2020
>
> - A. Tajdini, L. Jain, and K. G. Jamieson. Nearly minimax optimal submodular maximization with bandit feedback. NeurIPS, 2024
>
> - Z. Xu, X. Lin, and V. Tzoumas. Bandit submodular maximization for multi-robot coordination in unpredictable and partially observable environments. Robotics: Science and Systems(RSS), 2023a.
>
> - Z. Xu, H. Zhou, and V. Tzoumas. Online submodular coordination with bounded tracking regret: Theory, algorithm, and applications to multi-robot coordination. IEEE Robotics and Automation Letters(RAL),  2023b
>
> - K. Zhang, Z. Yang, and T. Basar. Multi-agent reinforcement learning: A selective overview of theories and algorithms. Handbook of reinforcement learning and control, pages 321–384, 2021.
>
> - Q. Zhang, Z. Wan, Y. Yang, L. Shen, and D. Tao. Near-optimal online learning for multi-agent submodular coordination: Tight approximation and communication efficiency. ICLR 2025
>
> - **H. Zhou, L. Huang, and B. Wang. Improved approximation algorithms for $k$-submodular maximization via multilinear extension.  ICLR 2025**

---

> > ### Comment · Reviewer_fXn2 · 2025-08-05
> > **Response to Author Rebuttal**
> >
> > Thank you for the clarifications. I hope the points I raised were helpful and I appreciate your efforts to address them in the revised version.

---

### Official Review · Reviewer_EsTn · 2025-06-28

**Clarity:** 4
**Significance:** 3
**Originality:** 3
**Rating:** 5
**Confidence:** 3

**Summary:**

This paper addresses the multi-agent online coordination (MA-OC) problem where multiple agents jointly select actions to maximize a time-varying, submodular or weakly submodular reward function under combinatorial constraints. The authors introduce a new relaxation called the policy-based continuous extension and develop two algorithms:

- MA-SPL: An online learning algorithm with tight approximation ratios for monotone submodular, weakly submodular, and weakly DR-submodular objectives.

- MA-MPL: A parameter-free variant that does not require knowledge of problem-specific constants (like curvature or weak submodularity parameters), yet still achieves similar performance.

The algorithms come with theoretical dynamic $\rho$-regret bounds and approximation guarantees. Empirical evaluations on multi-target tracking scenarios demonstrate that both methods outperform existing baselines.

**Questions:**

- Regarding the first point in weakness about the assumption on local feedback oracles, could the authors explain and discuss how the oracle can exactly and effectively model the limitation of UAV's perception range (Lines 106--110) and how such oracles might be realized and learned in practice?

- Could the authors elaborate on the practical implications of the increased communication complexity of MA-MPL compared to MA-SPL? Are there specific real-world scenarios where this trade-off would make MA-MPL less desirable despite its parameter-free nature?

- The authors claim that the approximation ratio achieved by the algorithms MA-SPL and MA-MPL is tight.
    Can the authors comment on the tightness of the regret bound achieved by MA-SPL and MA-MPL as well?

- The authors claim that ``new techniques are required to verify Theorem 3'' (Line 284).
    Can the authors briefly summarize the new techniques required in the proof for Theorem 3?

- In Figures 1(d)--1(f), the x-axis "Time Step" goes up to $500$, and the average utility does not seem to converge, nor the curves tend to become flat.
    By contrast, in Figure 1(a)-1(c), the time step goes up to 1200, and the average utility converges to a steady value.
    Is there a specific reason why the authors did not use a larger time scale and let the average utility converge for Figures 1(d)--1(f)?

**Ethical Concerns:**

["NO or VERY MINOR ethics concerns only"]

**Final Justification:**

1. This paper introduces new algorithms for multiagent online coordination, with nice theoretical results.
2. The techniques used meaningfully extend prior works on submodular optimization, and while I had some minor concerns about the practicality of their algorithms, the rebuttals have convinced me that the contributions are significant.
3. Most of the other reviewers also shared similar sentiments, and I think the only major 'unresolved' issue is that of the performance measure (dynamic regret) analyzed in this paper. However, I think finding a more suitable performance measure for the online coordination problem probably warrants another paper on its own, and I am thus happy to maintain my score for the current paper.

**Limitations:**

Yes.

**Paper Formatting Concerns:**

No major formatting issues.

**Quality:**

3

**Strengths And Weaknesses:**

Strengths
- The introduction of the novel ``policy-based continuous extension'' is a significant contribution. Its ability to provide a lossless rounding scheme for weakly submodular functions (not just submodular ones) is a clear advantage over traditional existing multi-linear extensions and addresses a critical limitation in the field.
- Theoretical guarantees are strong and tight. The paper provides rigorous theoretical analysis for both MA-SPL and MA-MPL algorithms, demonstrating near-optimal approximation ratios and dynamic regret bounds.
- The parameter-free MA-MPL algorithm is novel. The development of the MA-MPL algorithm to eliminate the dependence on possibly unknown parameters (such as $\alpha$, $\beta$, $\gamma$) is also practical.  In real-world scenarios, these parameters are often difficult to estimate, making the parameter-free approach a valuable direction to study.
- Overall, the manuscript is overall nicely written and easy to follow. Moreover, the theoretical analysis is rigorous, and the proofs look correct and sound, built upon existing results and tools on submodular optimization and online optimization.

Weaknesses

- The most significant weakness and largest concern with this paper lies in the assumption on access to a local feedback oracle. The paper assumes access to local marginal feedback oracles $ \mathcal{Q}_{i}^{t} $ for marginal gains which are further to be used for calculating surrogate gradients (Lines 111--115). While the authors claim that such a local marginal feedback oracle is ``common'' (Lines 116--118), and it is used as a core assumption in [111] cited by the authors, which is the work of this manuscript built upon, more discussions on how the oracles are connected to real-world scenarios and used to model them (for example, how the limitation of UAV's perception range can be captured by and modeled as local marginal feedback oracles, and possibly with one more concrete example beyond UAVs) and how such oracles might be realized and learned in practice (instead of merely citing [23]) would be helpful and could enhance clarity for a broader audience.
- Although the novelty of the manuscript is acknowledged, it seems to me that the novelty of the parameter-dependent algorithm MA-SPL is rather limited. The algorithm mostly follows the structure of the algorithm MA-OSEA in [111] and inherits the results from it, such as action sampling, information exchange, information aggregation, surrogate gradient estimation and policy update. The only difference is the use of policy-based continuous extension instead of multi-linear extensions. Hence, the framework of the algorithm MA-SPL overall isn't entirely novel, a fact that should be clearly acknowledged in the manuscript.
- While the algorithm MA-MPL is parameter-free, the paper acknowledges that it ``typically incurs greater communication complexity'' (Line 1505 and Table I) in the order of $\mathcal{O}(T^{5/2})$. Although it states future work will explore reducing this, a more in-depth discussion or initial analysis of this complexity compared to MA-SPL or other methods would strengthen the paper. It is also worthwhile to highlight the trade-off between a parameter-free algorithm and a greater running time in the manuscript.
- The experimental validation is limited to simulated environments for UAV target tracking.
    Evaluation on real-world systems or physical hardware (e.g., robotic coordination) would strengthen the empirical claims and showcase practical viability. Moreover, for weakly submodular objectives (Figure 1d-1f, Appendix A.2), the comparison is primarily against a random baseline, which seems to be a weak and low benchmark. While the authors state that no previous work explores MA-OC with weakly submodular objectives (Line 836), it would be beneficial to discuss if there are any offline weakly submodular maximization algorithms that could conceptually be adapted or contrasted (even if not directly online) to provide more context for the performance of the proposed algorithm.

---

> ### Author Rebuttal · Authors · 2025-07-28
>
> Thank you very much for your careful reading and insightful suggestions. We are grateful for the time and effort you have dedicated to reviewing our manuscript. In what follows, we will address some of your concerns in  `Questions`.
>
> ---
> >`Question1`: Regarding the first point in weakness about the assumption on local feedback oracles, could the authors explain and discuss how the oracle can exactly and effectively model the limitation of UAV's perception range (Lines 106--110) and how such oracles might be realized and learned in practice?
> ---
> Thank you very much for your question.
>
> In practical scenarios, beyond the online decision-making process discussed in this paper, each agent $i\in\mathcal{N}$ generally utilizes its own observations and those of neighboring agents, denoted as $\Theta_{t}(i)$, to estimate the set objective function $f_{t}(S)$. However, due to the limitations of local information $\Theta_{t}(i)$, the obtained set function estimation $f_{t}(\cdot|\Theta_{t}(i))$ often fails to accurately predict the marginal contributions of agents  $j\in\mathcal{N}$ that are far from agent $i$. To address this issue and simultaneously reduce the accumulation of learning errors, each agent $i$ is often permitted to use $f_{t}(\cdot|\Theta_{t}(i))$ for approximating its own actions' marginal contributions. This is why we adopt a local feedback assumption for each agent  $i$.
>
> It is worth noting that, in the local feedback model, we overlook the approximation errors of $f_{t}(\cdot|\Theta_{t}(i))$ and suppose that each $f_{t}(\cdot|\Theta_{t}(i))$ can unbiasedly or accurately estimate the marginal contributions of agent $i$. Similar to [23], we also incorporate prediction errors into our regret bound analysis. We leave this for future research.
>
>  Below, we provide a practical example of how to achieve these local feedback oracles. Consider the scenario in [83] where multiple satellites monitor a set of locations on Earth. Suppose there are $M$ locations labeled as $[M]$ and $n$ satellites denoted as $[n]$. At each time step $t\in[T]$, each satellite $i$ selects $k$ locations from its feasible locations $L_{i}\subseteq[M]$ to monitor, where the monitoring quality of each location $j$ is determined by the current video resolution $o^t_j(i)$
> provided by satellite $i$. Additionally, we assume that the monitoring effectiveness of each location depends only on the highest resolution of satellites monitoring This leads to the following submodular set objective function: $f_t(S)=\sum_{j=1}^{M}\max_{i\in U_j(S)}o^t_j(i)$ where $S\subseteq[n]\times[M]$ represents the pairing of satellites and locations and $U_{j}(S)$ denotes the set of satellites monitoring location $j$ under $S$. It is important to note that each decision made by satellite $i$ only affects location $j\in L_{i}$. In other words, if the communication graph allows satellites with overlapping monitoring ranges to exchange their decisions and resolution values $o^{t}_{j}(i)$ (see Fig. 1 in [83]), then each satellite $i$ can precisely calculate the marginal contribution of its own actions throughout simple arithmetic and maximization operations.  It is worth noting that, in this case, satellite $i$ cannot estimate the contributions of satellites without overlapping monitoring ranges.
>
> ---
> > `Question2`: Could the authors elaborate on the practical implications of the increased communication complexity of MA-MPL compared to MA-SPL? Are there specific real-world scenarios where this trade-off would make MA-MPL less desirable despite its parameter-free nature?
>
> ---
> Thanks for your question.
>
> It is worth noting that, in  Line 9 of **MA-SPL** algorithm, we require prior knowledge of the ratios $\alpha,\beta,\gamma$ to set the weight function $w(z)$. However, according to the definitions of weak submodularity and weak DR-submodularity, we know that accurately computing these parameters generally  involves exponential computations.
>
> Thus, the core of our proposed **MA-MPL** algorithm aims to overcome scenarios where obtaining the structural parameters $\alpha,\beta,\gamma$ is very difficult. Specifically, **MA-MPL** algorithm uses an additional $O(\sqrt{T})=O(\frac{T^{3/2}}{T})$ times communication complexity and $O(T^{3/2})=O(\frac{kT^{5/2}}{kT})$ times  query complexity to avoid the exponential computations required for the ratios $\alpha,\beta,\gamma$ inherent in **MA-SPL** algorithm.
>
> From our previous description, we know that when the computational cost of the ratios $\alpha,\beta,\gamma$ is significantly lower than the additional $O(\sqrt{T})$ times communication overhead and $O(T^{3/2})$ times function evaluations, **MA-MPL** becomes less desirable than **MA-SPL**. For instance, in Figure 1(a)-1(c) where we know $\alpha=\beta=\gamma=1$, parameter-dependent **MA-SPL** algorithm can achieve a better average reward with only $1/15$ of the function evaluations of parameter-free MA-MPL algorithm. (Note that this ratio $1/15$ is derived from the setups in Appendix A.3, where **MA-SPL** uses $10$ batches of stochastic estimation, while **MA-MPL** employs 10 batches of stochastic estimation for each of $15$ online linear maximization oracles.)
>
> ---
> >`Question3`: The authors claim that the approximation ratio achieved by the algorithms **MA-SPL** and **MA-MPL** is tight. Can the authors comment on the tightness of the regret bound achieved by **MA-SPL** and **MA-MPL** as well?
> ---
> Thank you for your question.
>
> It is important to note that when our objective function  $f_{t}$ is submodular and its curvature $c\rightarrow0$, the $f_{t}$ will degenerate into a modular function, that is, $f_{t}(\cup_{i=1}^{n}a_{i})=\sum_{i=1}^{n}f_{t}(a_{i})$. According to Definition 1, in this case, our proposed policy-based continuous extension $F_{t}$ will become a continuous **linear** function. From the classic theory of online linear optimization(OLO),  it is well-known that the $\mathcal{O}(\sqrt{T})$-regret bound  is optimal for general OLO problems, which also implies the optimality of the regret bounds of our proposed **MA-SPL** and **MA-MPL** algorithms with respect to the dependence on the time horizon $T$.
>
> In addition, we observe that the regret bounds of the **MA-SPL** and **MA-MPL** algorithms are also highly dependent on the topology of the communication graph $G$, such as the spectral gap $\tau$ and the diameter $d(G)$. Since this paper predetermines a communication network $G$, the design of a more efficient graph $G$ is beyond our current scope. At the same time, we are aware that many studies have investigated how to identify a more efficient communication architecture for multi-agent coordination problems. For further exploration, please refer to [45,40].
>
> ---
> > `Question4`: The authors claim that **new techniques are required to verify Theorem 3** (Line 284). Can the authors briefly summarize the new techniques required in the proof for Theorem 3?
> ---
> Thank you for your question. In the original work [109,111], the authors required the following inequality to prove their key conclusions(See Eq. (7) in [109] or Eq. (11) in [111]), that is to say,
> \begin{equation*}
>     \langle\textbf{y},\nabla G(\textbf{x})\rangle\ge\langle\textbf{y}\vee\textbf{x}-\textbf{x},\nabla G(\textbf{x})\rangle\ge G(\textbf{y}\vee\textbf{x})-G(\textbf{x}),
> \end{equation*} where $G$ is multi-linear extension of some submodular function $f$ and $\vee$  denotes coordinate-wise maximum operation. It is worth noting that this inequality necessitates that $\textbf{y}\vee\textbf{x}$ remains within the domain of $G$ for any two feasible $\textbf{y}$ and $\textbf{x}$. However, the domain $\prod_{i=1}^{n}\Delta_{k_{i}}$ of our proposed policy-based continuous extension $F_t$ does not meet this requirement.
>
> To overcome this limitation, we have developed two new inequalities for our proposed policy-based continuous extension  $F_t$  in our paper, that is,
>
> **a)**: when $f_{t}$ is monotone $\alpha$-weakly DR-submodular, the following inequality holds:
> \begin{equation*}
> 	\alpha\Big(f_{t}(S)-F_{t}(\uppi_{1},\dots,\uppi_{n})\Big)\le\langle1_{S},\nabla F_{t}(\uppi_{1},\dots,\uppi_{n})\rangle,
> \end{equation*} where  $S$ satisfies the constraints of problem (1) and the symbol $1_{S}$ represents the indicator function over the subset $S\subseteq V$, namely, for any action $v_{i,m}\in S$, the vector $1_{S}$ sets the corresponding probability for this action $v_{i,m}$ as $1$;
>
> **b)**: when $f_{t}$ is monotone $(\gamma,\beta)$-weakly submodular, we can show that
> \begin{equation*}
> 	\Big(\gamma^{2}f_{t}(S)-(\beta(1-\gamma)+\gamma^{2})F_{t}(\uppi_{1},\dots,\uppi_{n})\Big)\le\langle 1_{S},\nabla F_{t}(\uppi_{1},\dots,\uppi_{n})\rangle.
> \end{equation*}
>
> ---
> >`Question5`: In Figures 1(d)-1(f), the x-axis ``Time Step" goes up to 500, and the average utility does not seem to converge, nor the curves tend to become flat. By contrast, in Figure 1(a)-1(c), the time step goes up to 1200, and the average utility converges to a steady value. Is there a specific reason why the authors did not use a larger time scale and let the average utility converge for Figures 1(d)-1(f)?
> ---
> Thanks for your question.
>
>  We sincerely apologize for any confusion caused by the different time step scales used in Figures 1(a)-1(c) and Figures 1(d)-1(f). In Figures 1(d)-1(f), due to the unknown diminishing-return(DR) ratio $\alpha$, we employed a brute-force $0.1$-network search to identify the optimal parameter setups. Consequently, in sharp contrast to Figures 1(a)-1(c), we needed to conduct additional $50$ experiments (each repeated five times). To save total computational time, we therefore adopted a smaller time step of 500 in Figures 1(d)-1(f).
>
> We deeply appreciate you pointing out this limitation in our experimental design. To ensure that the curves in Figures 1(d)-1(f) become flat or converge, we will extend the time steps of these experiments in our final manuscript version.

---

> > ### Comment · Reviewer_EsTn · 2025-08-01
> > **Response to Author Rebuttal**
> >
> > Thank you for your answers and clarifications. I find it very interesting that local oracles (a standard theoretical tool) can be interpreted in a way which is consistent with the UAV application. Moreover, the connections between the regret bounds and classical optimal bounds from OLO are also an interesting comment. I agree with the comments of Reviewer 4FMc that this problem essentially boils down to an identical interest game, albeit one which admits submodular set-utility functions. Your results could point to deeper (and broader) connections to multiplayer games with submodular utility functions. Overall, I remain positive on the paper, and will maintain my score.

---

### Official Review · Reviewer_fLMy · 2025-07-01

**Clarity:** 3
**Significance:** 3
**Originality:** 3
**Rating:** 4
**Confidence:** 1

**Summary:**

The authors propose a novel continuous-relaxation technique to achieve multi-agent cooperation under submodular and weakly submodular objectives, and they theoretically prove the optimality of the algorithm while analyzing performance metrics such as communication cost and regret. The paper is well-structured and clearly written. However, further discussion is needed regarding problem modeling, assumptions, and the practical feasibility and deployability of the proposed method.

**Questions:**

1. How do you justify the requirement that each agent's action set must be disjoint, especially in scenarios where agents (such as identical UAVs or UGVs) would naturally share similar or identical action sets?
2. Could you discuss the practical feasibility of your approach in terms of communication requirements, asynchronous operation, and onboard computational constraints to better demonstrate the deployability of your algorithm?

**Ethical Concerns:**

["NO or VERY MINOR ethics concerns only"]

**Final Justification:**

I remain positive on the paper, including the writing, the method and the theoretical results. The submission is not in my area, so I have lowest confidence about my score.

**Limitations:**

yes

**Quality:**

3

**Strengths And Weaknesses:**

**Strength**

The authors have provided thorough modeling, algorithm design, and theoretical analysis for multi-agent cooperation problems with submodular and weakly submodular objective functions. The paper is well-written, the algorithms are comprehensive, and the proofs are convincing.

**Weaknesses**

1. I have some concerns regarding the modeling of the problem. The author requires the action sets of each agent to be disjoint, which is rather puzzling, since identical UAVs or UGVs would typically have the same action set. Furthermore, the stated problem (1) appears to be independent of the system state, whereas, in my view, the UAVs’ tracking tasks are highly dependent on target locations and related state information. In addition, if there are two equivalent decisions at time t (for instance, agent a moving right or agent b moving left), it is unclear which will be selected. Has the heterogeneity in the decision outcomes of different agents been fully considered?
2. Although the paper focuses primarily on the theoretical aspects, it would be beneficial to include some discussion on practical considerations, such as the feasibility of communication, the possibility of asynchronous operations, and computational feasibility under constrained onboard resources, in order to strengthen the persuasiveness of the algorithm’s deploy ability.

---

> ### Author Rebuttal · Authors · 2025-07-29
>
> Thank you very much for your careful reading and constructive feedback. We are grateful for the time and effort you have dedicated to reviewing our manuscript. In what follows, we will address some of your concerns in `Weaknesses` and `Questions`.
>
> ---
> >`Question1`: How do you justify the requirement that each agent's action set must be disjoint, especially in scenarios where agents (such as identical UAVs or UGVs) would naturally share similar or identical action sets?
> ---
> Thanks for your feedback.
>
> This question certainly highlights an aspect of our paper that deserves further clarification. In the context of UAVs’ tracking task, each UVA indeed shares the same movement space. However, it is crucial to emphasize that the **movement space** is distinct from the **action set** we consider in our paper. Broadly speaking, the objective function of the UAVs’ tracking task is not solely dependent on the identical **movement space**.  The key factor in shaping the objective function is **which UAV/agent carries out which particular movement decision**. To reflect the different role of each agent within the tracking system, we  commonly assign unique identity tags to each movement.
>
> Specifically, in our experiments, we consider the movement decision space $M=((\theta,s):s\in\{5,10,15\}\text{units/s},\theta\in(\frac{\pi}{4},\frac{\pi}{2},\frac{3\pi}{4},\pi,\dots,2\pi))$ where $\theta$ and $s$ represent the movement angle and the speed, respectively. To identify the executor of the movement, we will label the movement space $M$ with identity tags. Consequently, for each agent $i\in\{1,2,\dots,20\}$, its action space $V_{i}$ is constructed as follows:
> \begin{equation*}
>    V_{i}=M\times i=((\theta,s,i):s\in\{5,10,15\}\text{units/s},\theta\in(\frac{\pi}{4},\frac{\pi}{2},\frac{3\pi}{4},\pi,\dots,2\pi)),
> \end{equation*}
> where the action $(\theta,s,i)$ indicates that agent $i$ moves towards the direction of $\theta$ at a speed of $s$.
>
> With identification information included, these action sets are mutually disjoint, namely, $V_{i}\cap V_{j}=\emptyset$ for any $i\neq j\in(1,2,\dots,20)$. Please note that this approach with identity tagging is widely adopted in multi-agent systems, for example, multi-round product marketing [Sun et al., 2018, Huang et al.,2024] and online multi-advertisement slot allocation [Streeter et al., 2009].
>
> ---
> > `Weakness1`: I have some concerns regarding the modeling of the problem. The author requires the action sets of each agent to be disjoint, which is rather puzzling, since identical UAVs or UGVs would typically have the same action set. Furthermore, the stated problem (1) appears to be independent of the system state, whereas, in my view, the UAVs’ tracking tasks are highly dependent on target locations and related state information. In addition, if there are two equivalent decisions at time $t$ (for instance, agent a moving right or agent b moving left), it is unclear which will be selected. Has the heterogeneity in the decision outcomes of different agents been fully considered?
> ---
> Thank you very much for your feedback. Firstly, regarding the assumption that the action sets of each agent are mutually disjoint, please refer to our response to `Question1`.
>
> Secondly, in our models, every specific sequence of system states $(s_{1},s_{2},\dots,s_{T})$ in fact corresponds to a unique sequence of objective set functions $(f_{1},\dots,f_{T})$, where each set function $f_{t}$ is determined by the system states $(s_{1},\dots,s_{t-1})$ for any $1<t\le T$. Especially in the context of UAVs tracking tasks, these system states  primarily depend on the real-time locations of targets and agents. Unlike most control-related papers that assume the system environment follows a specific equation, our modeling considers any sequence of state sequences $(s_{1},s_{2},\dots,s_{T})$ that can ensure the (weak) submodularity of every $f_{t}$ for any $t\in[T]$. It is worth noting that when the system states $(s_{1},\dots,s_{t-1})$ are fixed, the submodularity or weak submodularity of each objective set function $f_{t}$ has been extensively verified in the literature[49, 50, 53, 55, 104, 115]. Thus, the goal of our paper aims to design an effective online algorithm that can achieve a tight approximation guarantee with a $\mathcal{O}(\sqrt{T})$-regret bound under the worst-case states sequence $(s_{1},s_{2},\dots,s_{T})$ that can ensure (weak) submodularity. Overall, our model focus on the worst-case states sequence(adversarial environment) rather than a specific one, which might lead you to mistakenly think our paper is unrelated to system states.
>
> Finally, we apologize that our current model and algorithms seemingly cannot distinguish two different decisions that have the same effect on the objective set function $f_{t}$. Incorporating the heterogeneity regarding the decision outcomes of different agents into our model is indeed an interesting topic for future research. Before that, in order to ensure the diversity and fairness of decisions, many studies, such as [Chen et al., 2020, Mitra et al., 2021, Zuo et al., 2023], have introduced an additional penalty term into the original objective function. Specifically, instead of optimizing  $f_{t}$, they consider a new objective function $f_{t}+\lambda\cdot h_{t}$, where $h_{t}$ measures the fairness or diversity of different agents and $\lambda$ is a positive constant. We believe this approach is also applicable for addressing heterogeneity.
>
> ---
> >`Weakness2`:Although the paper focuses primarily on the theoretical aspects, it would be beneficial to include some discussion on practical considerations, such as the feasibility of communication, the possibility of asynchronous operations, and computational feasibility under constrained onboard resources, in order to strengthen the persuasiveness of the algorithm’s deploy ability.
>
> >`Question2`: Could you discuss the practical feasibility of your approach in terms of communication requirements, asynchronous operation, and onboard computational constraints to better demonstrate the deployability of your algorithm?
> ---
> Thanks for your constructive suggestions.
>
> From a high-level viewpoint, we think the main bottlenecks affecting the deployability of our proposed algorithms stem from **a)** the time-varying communication graph caused by agents' position changes and **b)**  the non-negligible energy constraints of agents.
>
> **a)** : Generally speaking, due to limited perceptual range, each agent/UAV only can exchange information with others within its sensing radius.  Furthermore, in the online decision-making process, the positions of agents/UAVs will evolve over time, which in turn causes the communication graph topology to continually change. However, in our MA-SPL and MA-MPL algorithms, we assume a fixed communication graph, which may not align well with the real-word time-varying communication topology. Fortunately, time-varying network topologies have been extensively studied in distributed optimization [Nedic et al., 2017, Zhu et al., 2021]. We believe these previous insights from distributed optimization can be effectively integrated into our algorithms to enhance their real-world applicability.
>
>
> **b)** : Broadly speaking, each movement decision of UAVs will consume a certain amount of power resources. However, our current model only focuses on the utility of decisions without accounting for these associated energy costs. This may lead our algorithms to ignore decisions with high ROI(return-on-investment) but lower utility. In reality, such decisions might extend the UAVs' operational duration and boost cumulative utility by allowing more decision rounds. Fortunately, we find that the energy constraints of UAVs are well-matched with the trending long-term constrained online learning framework [Sadeghi and Fazel, 2020, Nie et al., 2024]. Specifically, at each time step $t\in[T]$,  a power consumption function $c_{t}^{i}$ is assigned to each agent $i$. We aim to ensure that the total power consumption of every agent $i$ does not exceed its predefined budget $B_{T}^{i}$, that is, $\sum_{t=1}^{T}c_{t}^{i}(a_{i}(t))\le B_{T}^{i}$, where $a_{i}(t)$ is the action taken by agent $i$ at time $t$. We believe that incorporating this concept of long-term constraints into our model can extend our algorithms to accommodate energy cost constraints, thereby improving their real-world deployability.
>
> ---
> **References**
>
> - W. Chen, W. Zhang, and H. Zhao. Gradient method for continuous influence maximization with budget-saving considerations. AAAI 2020
>
> - Y. Huang, S. Zhang, L. V. S. Lakshmanan, W. Lin, X. Xiao, and B. Tang. Efficient and effective algorithms for a family of influence maximization problems with a matroid constraint. Very Large Data Bases(VLDB) 2024
>
> - S. Mitra, M. Feldman, and A. Karbasi. Submodular+ concave. NeurIPS, 2021
>
> - A. Nedic, A. Olshevsky, and W. Shi. Achieving geometric convergence for distributed optimization over time-varying graphs. SIAM Journal on Optimization, 2017
>
> - G. Nie, V. Aggarwal, and C. Quinn. Gradient methods for online dr-submodular maximization with stochastic long-term constraints. NeurIPS, 2024
>
> - O. Sadeghi and M. Fazel. Online continuous dr-submodular maximization with long-term budget constraints. AISTATS, 2020
>
> - M. Streeter, D. Golovin, and A. Krause. Online learning of assignments.NeurIPS, 2009.
>
> - L. Sun, W. Huang, P. S. Yu, and W. Chen. Multi-round influence maximization. KDD, 2018
>
> - J. Zhu, Q. Wu, M. Zhang, R. Zheng, and K. Li. Projection-free decentralized online learning for submodular maximization over time-varying networks. Journal of Machine Learning Research, 2021
>
> - P. Zuo, Y. Wang, and S. Tang. Regularized online dr-submodular optimization. UAI, 2023

---

> > ### Comment · Reviewer_fLMy · 2025-08-05
> >
> > Thank you for your comprehensive response to my comments. At this stage, I will retain my current score. I hope that the revised version will provide a thorough clarification of the concerns outlined above.

---

### Official Review · Reviewer_4FMc · 2025-07-03

**Clarity:** 4
**Significance:** 3
**Originality:** 3
**Rating:** 5
**Confidence:** 4

**Summary:**

This paper studies the multi-agent online coordination problem, where multiple agent choose their actions in a finite action set to maximize a set function. This is an optimization problem with discrete support. The authors propose to learn stochastic polices for the agents so that the problem is converted to a continuous optimization problem, and this framework is named policy-based continuous-relaxation. Under this framework, the authors characterizes the suboptimality of stationary points of the relaxed continuous function (Theorem 2). Moreover, a surrogate objective function is proposed to improve the suboptimality gap (Theorem 3). Effectiveness of the algorithm is verified by simulations.

**Questions:**

Questions are included in ``Strengths and Weaknesses''.

**Ethical Concerns:**

["NO or VERY MINOR ethics concerns only"]

**Final Justification:**

The authors addressed most of my concerns in the rebuttal. I will maintain my original rating towards accept.

**Limitations:**

Yes.

**Paper Formatting Concerns:**

No major formatting concerns.

**Quality:**

4

**Strengths And Weaknesses:**

**This is a nice submission with the following strengths:**
- A new continuous relaxation is proposed for set function optimization, which enables provably efficient optimization for weakly submodular functions. As a comparison, previous relaxation approaches relies heavily on the ``strong'' submodular assumption.
- The authors theoretically characterize suboptimality of stationary points and proposes a method to improve suboptimality.
- An end-to-end optimization algorithm is proposed and analyzed.
- Empirical evaluation verify the effectiveness of the algorithm.

**That said, the reviewer do have the following concerns/question:**
- After policy-based relaxation, the problem is essentially reduced to a multi-agent identical interest game (a subset of potential games (PG)). There exists a huge amount of literature that designs provably efficient algorithms that learn nash equilibrium (a subset of stationary points) and provide price of anarchy lower bounds (or suboptimality in the context of this paper) for PGs. Are results for PGs also applicable to this setting? If so, how are the results in this paper compare to results for potential games?
- Consider the surrogate function In theorem 3. Are stationary points (denoted by $x^s$) of the surrogate function also stationary points of the policy-based relaxed objective $F$? If not, what will happen if we start from $x^s$ and do gradient descent using gradients of function $F$? Will this strategy increase or decrease the sub-optimality gap? In case this is hard to characterize theoretically, I would like to see at least come empirical evaluations and discussions.
- In line 5 of Algorithm 1, the authors normalize the policy and sample actions. Is it necessary for the empirical performance or theoretical analysis? As far as I am concerned, function $F^s$ seems to be supported on the entire probability simplex and gradient descent seems to work without this projection step. If this step is indeed necessary either theoretically or empirically, ablation simulation studies and discussions would be helpful.

**Minor comments**:
- The current abstract has involved too many specific terms (e.g. submodularity ratio, weakly submodularity ratio) for the general audience to understand. It would be helpful to talk about them more high-levelly.
- It would be helpful to give a high-level introduction about how previous extension, e.g. multi-linear extension, works.

---

> ### Author Rebuttal · Authors · 2025-07-30
>
> Thank you very much for your careful reading and constructive feedback. We are grateful for the time and effort you have dedicated to reviewing our manuscript. In what follows, we will address some of your concerns in `Weaknesses`.
>
> ---
> `Weakness1`: After policy-based relaxation, the problem is essentially reduced to a multi-agent identical interest game (a subset of potential games (PG)). There exists a huge amount of literature that designs provably efficient algorithms that learn nash equilibrium (a subset of stationary points) and provide price of anarchy lower bounds (or suboptimality in the context of this paper) for PGs. Are results for PGs also applicable to this setting? If so, how are the results in this paper compare to results for potential games?
>
> ---
>
> Thank you for your insightful comments.
>
> At first, we are so sorry that neither of us is a specialist in computational game theory or Nash equilibrium. However, based on our limited knowledge of potential games, we think that if we ignore the changing environment and return to the offline setting, previous algorithms for learning or approximating Nash equilibrium could be applicable to our proposed policy-based continuous extension $F_{t}$. Nonetheless, according to our Theorem 2 and the approximation result for the equilibrium solutions of submodular utility functions (See Proposition 1 in [Du et al., 2022]), we can infer that these existing algorithms for approximating Nash equilibrium of $F_{t}$, in theory, also can guarantee a **sub-optimal approximation ratio** of $(\frac{1}{1+c})$, $(\frac{\alpha^{2}}{1+\alpha^{2}})$ or $(\frac{\gamma^{2}}{\beta+\beta(1-\gamma)+\gamma^{2}})$ to the offline multi-agent coordination problem (1), when the original set function $f_{t}$ is monotone  submodular with curvature $c$, $\alpha$-weakly DR-submodular or $(\gamma,\beta)$-weakly submodular.
>
> Secondly, in order to obtain tight approximation guarantees, we indeed could apply algorithms for approximating Nash equilibrium to the surrogate function $F^s_t$ proposed in our Theorem 3. However, similar to our proposed **MA-SPL** algorithm, these aforementioned equilibrium algorithms on $F^{s}_{t}$ also require prior knowledge of the difficult-to-obtain diminishing-return(DR) ratio $\alpha$ and submodularity ratios $\beta,\gamma$ to set weight function $w(z)$. In comparison, our **MA-MPL** algorithm not only achieves **tight** approximation ratios but also is **parameter-free**.
>
> ---
> >`Weakness2`: Consider the surrogate function In theorem 3. Are stationary points (denoted by $x^{s}$) of the surrogate function also stationary points of the policy-based relaxed objective $F$? If not, what will happen if we start from $x^{s}$ and do gradient descent using gradients of function $F$? Will this strategy increase or decrease the sub-optimality gap? In case this is hard to characterize theoretically, I would like to see at least come empirical evaluations and discussions.
>
> ---
> Thank you for your constructive feedback.
>
> At first, we are so sorry that we cannot confirm whether the stationary point $x^{s}$ of surrogate function $F^s_t$ is also a stationary point of the original policy-based continuous relaxation $F_{t}$. It is worth noting that, if $x^{s}$ is a stationary point of $F^s_t$, for any  $y\in\prod_{i=1}^{n}\Delta_{k_{i}}$, it satisfies $\langle y-x^s,\int_0^1 w(z)\nabla F_{t}(z*x^s)dz\rangle\le 0 $.
>
> However, when $y\in\prod_{i=1}^{n}\Delta_{k_{i}}$,  we seemingly cannot derive $\langle y-x^{s},\nabla F_{t}(x^{s})\rangle\le0$ from the above condition $\langle y-x^{s},\int_{0}^{1} w(z)\nabla F_{t}(z*x^{s})\mathrm{d}z\rangle\le0$.
>
> Secondly, previous studies [Sviridenko et al., 2017, Harshaw et al., 2019]  have indicated that for any $\epsilon>0$, it cannot be approximated in polynomial time within a ratio of $(1-\frac{c}{e}+\epsilon)$ or $(1-e^{-\alpha}+\epsilon)$  for maximizing a monotone submodular function with curvature $c$ or $\alpha$-weakly DR-submodular function, unless **RP**=**NP**. Furthermore, from Theorem 3, we can know that $x^{s}$ can attain an approximation ratio of $(1-\frac{c}{e})$ or $(1-e^{-\alpha})$ to the submodular or $\alpha$-weakly DR-submodular maximization problem (1), respectively. Therefore, based on these established inapproximability results  [Sviridenko et al., 2017, Harshaw et al., 2019] , we can ensure that it is theoretically impossible to improve the approximation ratios for the entire family of submodular or weakly submodular functions by starting from a stationary point $x^{s}$ of $F^{s}_{t}$ and then performing gradient ascent using the gradients of the original $F_t$,  as $x^{s}$ has already achieved the **tight** approximation ratio of $(1-\frac{c}{e})$ or $(1-e^{-\alpha})$.
>
> ---
> `Weakness3`: In line 5 of Algorithm 1, the authors normalize the policy and sample actions. Is it necessary for the empirical performance or theoretical analysis? As far as I am concerned, function $F^{s}$ seems to be supported on the entire probability simplex and gradient descent seems to work without this projection step. If this step is indeed necessary either theoretically or empirically, ablation simulation studies and discussions would be helpful.
>
> ---
> Thank you very much for your insightful question. It certainly highlights an aspect of our paper that deserves further clarification.
>
> At first, it is worth noting that, in this paper, we mainly explore the **monotone** objective set function $f_{t}$. Then, according to part **2)** of Theorem 1, we also can know that when the original objective set function $f_{t}$ is monotone, its policy-based continuous extension $F_{t}$ is also monotone. More specifically, for any two  policy vectors $(\pi_1,\dots,\pi_n)$ and $(\hat{\pi}_1,\dots,\hat{\pi}_n)$,  if  $\hat{\pi}_i\ge\pi_i$  and $f_t$  is **monotone**,
> we have $F_t (\hat{\pi}_1,\dots,\hat{\pi}_n)\ge F_t (\pi_1,\dots,\pi_n)$.
>
>
>  Motivated by this result, we also can show that, for any policy vector $(\pi_{1},\dots,\pi_{n})\in\prod_{i=1}^{n}\Delta_{k_{i}}$, when $f_{t}$ is **monotone**,
>
> we have $F_t (\frac{\pi_1}{|| \pi_1 ||_1},..., \frac{\pi_n}{|| \pi_n ||_1})\ge F_t (\pi_1,\dots,\pi_n)$  due to  $\frac{\pi_i}{|| \pi_i ||_1}\ge \pi_i $.
>
> As a result, in line 5 of Algorithm 1, we utilize the normalized policy to select actions. We truly appreciate you pointing out the deficiency in the description of our Algorithm 1. We will add an extra remark to the line 5 of Algorithm 1 in the final version.
>
> ---
> >`Minor Weakness1`: The current abstract has involved too many specific terms (e.g. submodularity ratio, weakly submodularity ratio) for the general audience to understand. It would be helpful to talk about them more high-level.
>
> ---
> Thanks for your suggestion.
> Indeed, the abstract currently contains too many specific terms like diminishing-return(DR) ratio $\alpha$
> and submodularity ratios $\beta,\gamma$, which are not friendly to general audiences. To make the abstract more high-level, we will unify **$\alpha$-weakly DR-submodular and $(\gamma,\beta)$-weakly submodular scenarios** as **weakly submodular scenarios** in our final version.
>
> ---
> >`Minor Weakness2`:  It would be helpful to give a high-level introduction about how previous extension, e.g. multi-linear extension, works.
>
> ---
> Thanks for your helpful suggestion.
>
> Before that, we have considered adding some basic content about multi-linear extension and its comparison with our proposed policy-based continuous extension in **Related Work** section or the end of **Section 4.2**. However, due to space limitations, we moved it to Appendix C.4. In order to help readers interested in multi-linear extension quickly locate Appendix C.4, we will add some relevant details and a reference to it after Remark 5 in our final version.
>
> ---
> **References**
> - B. Du, K. Qian, C. Claudel, and D. Sun. Jacobi-style iteration for distributed submodular maximization. IEEE transactions on automatic control(TAC), 2022
>
> - C. Harshaw, M. Feldman, J. Ward, and A. Karbasi. Submodular maximization beyond non-negativity: Guarantees, fast algorithms, and applications. ICML, 2019.
>
> -  M. Sviridenko, J. Vondr ´ak, and J. Ward. Optimal approximation for submodular and supermodular optimization with bounded curvature. Mathematics of Operations Research(MOR), 2017

---

> > ### Comment · Reviewer_4FMc · 2025-08-08
> >
> > Thank you for your comprehensive rebuttal, my concerns are addressed well. As a final suggestion, I would appreciate it if the authors could connect this paper to the game theory community by incorporating related discussions in the paper.

---

### Decision · Program_Chairs · 2025-09-17

**Decision:**

Accept (poster)

**Comment:**

This paper introduces two algorithms for multi-agent online coordination based on a new policy-based continuous extension. The work is supported by strong theoretical results, rigorous proofs, and the practical value of a parameter-free algorithm (MA-MPL). Reviewers highlighted these strengths while also pointing to some concerns about assumptions (disjoint action sets, oracle access), limited novelty for MA-SPL, higher communication cost of MA-MPL, and the lack of real-world experiments. During rebuttal, the authors clarified these points, and reviewers acknowledged the responses and provided final justifications for the reviews. Overall, the review process was thorough, and there is consensus that the contributions are solid and impactful.

The only remaining issue after the panel discussion is the use of dynamic \(\rho\)-regret as a suitable performance measure in this section. To address this, the panel would like to request from the authors to add a short discussion in the camera-ready version of their paper highlighting the points that were raised during the rebuttal discussion and, in particular, with Reviewer fXn2; namely that \(\rho\)-Regret may not be a perfect metric for online (weakly) submodular maximization problem, as the rewards or solutions obtained in each round can indeed surpass the predefined approximation ratio, explaining also the reasons why this is used in the current paper.